# Laboratory characterisations and intercomparison sounding test of dual thermistor radiosondes for radiation correction

Sang-Wook Lee[1,2], Sunghun Kim[1], Young-Suk Lee[1], Jae-Keun Yoo[1], Sungjun Lee[1], Suyong Kwon[1,2], Byung Il Choi[1], Jaewon So[3], Yong-Gyoo Kim[1,*]

[1]Division of Physical Metrology, Korea Research Institute of Standards and Science, Daejeon 34113, Republic of Korea
[2]Department of Science of Measurement, University of Science and Technology, Daejeon 34113, Republic of Korea
[3]Weathex, Gunpo 15880, Republic of Korea

*Correspondence to*: Y.-G. Kim (dragon@kriss.re.kr)

**Abstract.** A dual thermistor radiosonde (DTR) comprising two (aluminium-coated and black) sensors with different emissivities was developed to correct the effects of solar radiation on temperature probes based on *in-situ* radiation measurements. Herein, the DTR performance is characterised in terms of the uncertainty via a series of ground-based facilities and an intercomparison radiosounding test. The DTR characterisation procedure using laboratory facilities is as follows: individually calibrate the temperature of the thermistors in a climate chamber from −70 to 30 ℃ to evaluate the uncertainty of raw temperature measurement before radiation correction; test the effect of temperature on the resistance reading using radiosonde boards in the climate chamber from −70 to 20 ℃ to identify a potential source of errors owing to the boards, especially at cold temperatures; individually perform radiation tests on thermistors at room temperature to investigate the degree of heating of aluminium-coated and black sensors (the average ratio = 1:2.4) and use the result for obtaining unit-specific radiation correction formulas; and perform parameterisation of the radiation measurement and correction formulas with five representative pairs of sensors in terms of temperature, pressure, ventilation speed, and irradiance using an upper air simulator. These results are combined and applied to the DTR sounding test conducted in July, 2021. Thereafter, the effective irradiance is measured using the temperature difference between the aluminium-coated and black sensors of the DTR. The measured irradiance is then used for the radiation correction of the DTR aluminium-coated sensor. The radiation-corrected temperature of the DTR is mostly consistent with that of a commercial radiosonde (Vaisala, RS41) within the expanded uncertainty (~0.35 ℃) of the DTR at the coverage factor $k = 2$. Furthermore, the components contributing to the uncertainty of the radiation measurement and correction are analysed. The DTR methodology can improve the accuracy of temperature measurement in the upper air within the framework of the traceability to the International System of Units.

# 1 Introduction

Measurement of essential climate variables such as temperature and water vapour (i.e. humidity) is important as they are essential input data for weather and climate prediction models (Bojinski et al., 2014). The temperature and humidity in upper air are frequently and widely measured using radiosondes. Radiosonde is a telemetry device comprising various sensors that measure meteorological parameters and transmit the collected measurement data via radio frequency while being flown by a weather balloon up to about 35 km in altitude. Radiosonde observations are often co-located with global navigation satellite

system radio occultation and used as reference for validating their one-dimensional interpolation which follows the flight trajectories of balloon soundings. The measurement accuracy of radiosondes needs to be improved in terms of uncertainty within the framework of the traceability to the International System of Units (SI).

Joint research programs between the metrology and the meteorology and climate communities, such as the 'MeteoMet– Metrology for Meteorology' project, were initiated (Merlone et al., 2015; Merlone et al., 2018) to acquire high quality

observation data on meteorological variables. Reference facilities have been developed through the project for calibrating the meteorological observation instruments to be used in the meteorological community. Additionally, low-temperature and low-pressure humidity chambers have been developed for calibrating radiosonde humidity sensors in the environments imitating the upper troposphere/lower stratosphere (Sairanen et al., 2015; Cuccaro et al., 2018). To investigate the climate change, a certain level of measurement uncertainty in radiosoundings should be secured. Hence, the Global Climate Observing System

(GCOS) Reference Upper Air Network (GRUAN) was founded to establish a dataset of traceable measurements with quantified uncertainties (Gcos, 2007). The required measurement accuracy of temperature specified by GRUAN is 0.2 °C in the stratosphere (Gcos, 2007).

A difficulty in improving the measurement accuracy of radiosondes is the correction of the solar-radiation-induced heating of sensors during the daytime. The radiative heating of sensors is also affected by environmental conditions, such as

temperature, air pressure and air ventilation, that are involved with convective cooling (Lee et al., 2018b; Lee et al., 2020). All these parameters should be considered together to precisely evaluate the radiation correction of radiosonde temperature sensors. Although most radiosonde manufacturers apply radiation corrections to their products (Nash et al., 2011), they do not disclose the detailed methodologies, including reference systems and correction algorithms. To independently evaluate the radiosondes, GRUAN has built a ground-based calibration facility and established a correction algorithm for the

GRUAN data processing (GDP) of the Vaisala RS92 radiosonde (Dirksen et al., 2014). The uncertainty of the GDP of RS92 during daytime was gradually increased from 0.2 °C at the surface to 0.6 °C at 30 km with the coverage factor $k = 2$ (Dirksen et al., 2014). Recently, the same group built a new simulator to investigate the solar temperature error of radiosondes (SISTER) and derived a new GDP algorithm for the Vaisala RS41 radiosonde (Von Rohden et al., 2022). The setup can control the irradiance, air pressure, ventilation, sensor rotation and tilting of the light incident angle. Using the setup, the

uncertainty of the GDP of RS41 is evaluated to be about 0.3 °C ($k = 2$) at 35 km. It is also found that the daytime GRUAN profile is 0.35 °C warmer than the manufacturer's at 35 km (Von Rohden et al., 2022). However, the surrounding

temperature, which also affects the radiation correction, cannot be changed. Furthermore, an upper-air simulator (UAS) was developed by the Korea research institute of standards and science (KRISS) to similarly evaluate radiosondes (Lee et al., 2020). The UAS at KRISS can simultaneously control the temperature, pressure, ventilation and irradiance, and the UAS

was recently supplemented with sensor tilting and rotation functions. Using this setup, a radiation-correction formula of the RS41 radiosonde is presented (Lee et al., 2021).

However, the radiation correction processes by GRUAN and KRISS assume that the solar irradiance is known. In fact, the solar irradiance is dependent on various parameters such as cloud conditions, solar elevation angle, season and location. Till date, the direct *in-situ* measurements of solar irradiance are difficult without using additional pyranometers measuring on the

70 same payload of radiosondes (Philipona et al., 2013). An alternative approach comprises the simulation of solar irradiance with appropriate cloud scenarios, surface albedo and solar angle (Key and Schweiger, 1998). From the perspective of the radiation correction uncertainty, the SI traceability of the simulated irradiance is incomplete when the sky is clear or cloudy because the simulated irradiance is constructed from the average of clear and cloudy sky cases (Von Rohden et al., 2022). This results in the increase of the radiation correction uncertainty in troposphere.

To resolve this issue, the concept of a dual thermistor radiosonde (DTR) comprising two temperature sensors with different emissivities was introduced to measure the effective irradiance using the temperature difference between them (Lee et al., 2018a; Lee et al., 2018b). The DTR operation principle was demonstrated by investigating the effects of air ventilation, and temperature and pressure using a wind tunnel and a climate chamber system, respectively. The temperature difference between the dual thermistors showed to be linearly proportional to the effective irradiance, and the radiation-induced heating

of the sensors was corrected according to the measured effective irradiance. Only the slope of the linear function of the radiation measurements and correction formulas changed with the environmental parameters, and the linearity itself was not altered. However, these DTR formulas were obtained using two separate setups that cannot be combined, and thus, the correction formula was incomplete in terms of the SI traceability.

Herein, the combined effect of temperature, pressure, ventilation and irradiance on DTR is investigated using the UAS at

85 KRISS for the parameterisation of the radiation measurement and correction. The obtained formulas are used in an intercomparison sounding test performed in July, 2021. Furthermore, a series of laboratory characterisations of DTR is conducted, including individual calibration of thermistors, test of temperature effect on resistance reading by radiosonde boards and individual radiation test on thermistors. The uncertainties due to parameterisation of the radiation correction formula using UAS and other characterisations are also evaluated. Then, the uncertainty components and their combined

budget for the measured irradiance and corrected temperature in the sounding test are presented. Finally, the corrected temperatures of the DTR and the RS41 from parallel soundings are compared and the difference between them is discussed in terms of the uncertainty.

## 2 Introduction to DTR

### 2.1 Dual thermistors with different emissivity

Figure 1(a) shows a DTR comprising two temperature sensors that are chip-in-glass type negative temperature coefficient (NTC) thermistors (Shibaura electronics, Model: PB7-41E). The glass bead encapsulating the sensing element is ellipsoidal shape with $0.55 \pm 0.1$ mm diameter and $1.1 \pm 0.3$ mm length. The two thermistors are attached to the sensor boom via soldering and followed by epoxy for electrical insulation. The thermistors and sensor boom are coated with aluminium (Al) via thermal evaporation. One sensor is additionally coated with a black epoxy (Loctite, Model: STYCAST 2850 FT) to differentiate the emissivity (absorptivity) between them (inset of Fig. 1(a)). For convenience, herein, the sensor coated with only Al is referred to as the white sensor, while the other sensor is referred to as the black sensor.

### 2.2 DTR operation principle

Previously, a pioneer work using multiple thermistors with different spectral responses (emissivity and absorptivity) was conducted for the radiation correction. In the work, however, complete knowledge on material properties of air and sensors and sensor geometry is required to solve multiple heat balance equations (Schmidlin et al., 1986). DTR utilises the purely experimental temperature difference between the white and black sensors to measure the effective irradiance and correct its effect on the white sensor (Fig. 1(b)). The temperature increase of each sensor due to solar irradiation is linearly proportional to the effective irradiance, as previously investigated by various theoretical and experimental studies (Lee et al., 2018b; Lee et al., 2018a; Luers, 1990; Mcmillin et al., 1992). Additionally, the temperature of the black sensor ($T_{B\_raw}$) is higher than that of the white sensor ($T_{W\_raw}$) due to its high light absorptivity. Thus, the temperature difference between them ($T_{B\_raw} - T_{W\_raw}$) is also linearly proportional to the effective irradiance. Although other environmental parameters, e.g. air pressure and ventilation, affect the degree of heating of sensors via convective cooling; they only change the slope of the linear function and do not affect the linearity itself. The effect of other environmental parameters on the temperature difference and the temperature increase of the white sensor are investigated using the UAS developed at KRISS (Lee et al., 2021). Experimental results of the UAS are used to determine a formula to measure the effective irradiance based on the temperature difference between the two thermistors. Section 4 describes the procedure to obtain these formulas for the measurement of irradiance and the correction of the white sensor using the UAS in more detail.

## 3 DTR characterisation

### 3.1 Characterisation procedure

The DTR characterisation procedure is summarised in Figures 2(a)–(e). The characterisation process is categorised into laboratory experiments and sounding test. First, the thermistors on the sensor boom are individually calibrated using a climate chamber from −70 to 30 °C to evaluate the uncertainty of raw temperature measurement before radiation correction

(Fig. 2(a)). Then, the temperature effect on the resistance reading by radiosonde boards is tested in the climate chamber from −70 to 20 °C to identify a potential source of errors owing to the boards, especially at cold temperatures (Fig. 2(b)). The temperature increase of all thermistors due to irradiation is individually recorded at room temperature (Fig. 2(c)) to include the differences in the sensitivities of the individual thermistors in the radiation correction. The radiation measurement and correction formulas of DTR are obtained in terms of temperature, pressure, ventilation speed and irradiance using the UAS (Fig. 2(d)). The laboratory experimental results are combined and applied to the DTR sounding system. Then, the sounding results of DTR are compared with those of a commercial radiosonde through dual soundings (Fig. 2(e)). Each characterization procedure will be discussed in detail in the following sections.

## 3.2 Individual calibration of thermistors in a climate chamber

All thermistors are individually calibrated in a climate chamber (Kambic, Model: KK-190 CHULT). Figure 3(a) displays the calibration setup showing the sensors on the booms in the climate chamber, a digital multimeter (Keysight, Model: 34980A) to record the sensor resistances and a data acquisition computer. The setup can calibrate 35 pairs ($7 \times 5$) of dual thermistors that are located on the same rectangular plane (230 mm × 190 mm). Five platinum resistance thermometers (PRT) with a nominal resistance of 100 Ω (PT100) are used as reference thermometers. They were calibrated at KRISS with an uncertainty of 0.05 °C at a coverage factor $k = 2$ and installed at the centre and four corners in the same rectangular plane as the thermistors. The average of the temperatures measured by the five reference PRTs ($T_{Ref\_aver}$) is used as the reference temperature for calibration. Six calibration points are selected from −70 to 30 °C (Fig. 3(b)), of which y-axis denotes the spatial temperature deviations ($T_{Ref\_devi}$) represented by the maximum deviation from $T_{Ref\_aver}$. Although the calibration range should be extended to −90 °C to cover temperatures over tropic and polar regions, it is not feasible using the climate chamber because the typical lowest temperature limit is more or less −80 °C.

Generally, the Steinhart–Hart equation is used for the calibration of NTC thermistors (White, 2017). However, the application of a third-order polynomial equation, i.e. the inclusion of a quadratic term, which is not present in the Steinhart-Hart equation, yields smaller fitting residuals than that of a second-order equation (Yang et al., 2021). Therefore, the Steinhart-Hart equation is modified for the calibration as follows:

$$\frac{1}{T_s} = a_0 + a_1 \ln(R_s) + a_2 [\ln(R_s)]^2 + a_3 [\ln(R_s)]^3, \tag{1}$$

where $T_S$ is the sensor temperature obtained based on the sensor resistance $R_S$ and $a_0$, $a_1$, $a_2$ and $a_3$ are the fitting coefficients. The distributions of the fitting residuals of the white and black sensors are shown in Figs. 3(c) and (d), respectively. Totally, 696 data points are obtained, collected from six calibration points of 116 thermistors of the same colour. No essential difference is observed between the white and black sensors in the distributions of the residuals, implying that the emissivity difference plays a negligible role in the sensor calibration process.

Furthermore, the uncertainty of radiosonde sensors (thermistors) due to calibration $U(T_{S\_cal})$ at $k = 2$ is calculated, as shown in Fig. 3(e). The contributing uncertainty factors are temperature deviations $U(T_{Ref\_devi})$, stability $U(T_{Ref\_stab})$ and calibration

$U(T_{\text{Ref\_cal}})$ of the reference PRTs and the temperature stability $U(T_{\text{S\_stab}})$ and fitting residuals $U(T_{\text{S\_fit\_resid}})$ of the sensors (thermistors). Consequently, the uncertainty of thermistors due to calibration is about 0.1–0.3 ℃ ($k = 2$) between 30 and −70 ℃. The uncertainty due to spatial temperature deviations $U(T_{\text{Ref\_devi}})$ in the chamber dominates the calibration uncertainty. The deviations are due to the temperature difference between the front door side and the rear fan side of the chamber. One of the practical ways to reduce the calibration uncertainty is to conduct another round of calibration with the

thermistor set (35 pairs) rotated 180° in the chamber and average out the effect of temperature deviations. Another way is to find other locations with smaller temperature deviations. The temperature deviations can be affected by the thermal insulation of the door and the aisles for data cables as well as the ventilation by the fan in the chamber.

### 3.3 Test of temperature effect on resistance reading by radiosonde boards

   To properly measure the temperature using the thermistors via Eq. (1), the effect of the temperature of the radiosonde

electronics board on the thermistor resistance measurement should be investigated in the same temperature range of the thermistor calibration. Thus, ten radiosonde prototypes covered with expanded polystyrene foam are installed in the climate chamber with varying temperatures. The radiosonde boards are wired to external reference resistors (Cropico, Model: 008-B) instead of thermistors, as shown in Fig. 4(a). The resistance measured by radiosonde boards is collected by a computer via wired communication.

Figure 4(b) shows the difference between the reference resistance and radiosonde reading as a function of the reference resistance. The reference resistance is changed according to the environmental temperature of the radiosonde boards, which is varied from −70 to 20 ℃. For example, a reference resistance of 700 kΩ is chosen to imitate the sensor resistance of −70 ℃ when the temperature of the climate chamber measured by reference PRTs ($T_{\text{Ref\_aver}}$) is −70 ℃. Thereafter, both the reference resistance and resistance reading by radiosonde boards are converted into temperatures using a calibration curve based on Eq.

(1). The resultant temperature error by radiosonde boards with varying temperature is shown in Fig. 4(c). Assuming that the probability distribution is a normal distribution function, the standard deviation (SD) of all data points (0.04 ℃) is used for standard uncertainty due to the influence of the temperature of radiosonde electronics boards on the resistance (or temperature) measurement.

### 3.4 Individual radiation test on radiosonde thermistors

The purpose of the calibration of thermistors and the investigation of the temperature effect on radiosonde electronics boards is to assess the accuracy (or uncertainty) of raw temperature measurement before radiation correction. The next step is to investigate the sensitivity of individual thermistors to irradiation because the amount of radiation correction varies for individual radiosondes, presumably related to the production process of the thermistors. This can be attributed to the irregularity in the thermistor glass bead sizes, the black epoxy coating and the sensor connection to the boom via soldering

and epoxy. Effective irradiance to thermistors and the cooling by convection can be changed based on the glass bead sizes

and the Al and black epoxy coatings. The connection between the sensor leads and the boom may be irregular because the soldering and the coating of epoxy resin were conducted manually. Radiative heating of glass beads, leads, and connection parts between the sensor leads and the boom should be affected by their size as previously reported (De Podesta et al., 2018). However, obtaining radiation correction formulas of radiosondes individually using UAS with varying temperature, pressure, air ventilation speed and irradiance is time-consuming and economically unfavourable. Therefore, a radiation test is performed on all thermistors in a vacuum chamber at room temperature (~25 °C), and the results are correlated to the UAS experiments to acquire sensor-specific radiation correction formulas that reflect the unit difference. The RRT irradiance at the sensor position is 800 W·m$^{-2}$ with 0.8% standard deviation for each irradiation. The ventilation and the pressure in the chamber are not measured. Since they depend on the performance of the vacuum pump and the sealing of the chamber lid using an O-ring, there can be slight variations in the ventilation and the pressure.

Figure 5(a) shows an individual radiation test setup comprising a solar simulator, vacuum pump, vacuum chamber, digital multimeter and computer. A pair of dual thermistors is illuminated through a window in the lid of the vacuum chamber. The diameter ($D$) of the beam spot on the sensor is 45 mm and the distance between the sensor bead and the beam boundary is 25 mm. The rotation of 12 pairs of sensors and the light irradiation of the solar simulator are automatically controlled using a computer program. When a pair of dual thermistors arrives and stops beneath the window during rotation, the window is screened by a shutter to block the light irradiation. At this time, the irradiance is measured using a calibrated pyranometer on the shutter. Then, the shutter is opened and closed for 180 s each and this process is repeated three times for the illumination on each pair of thermistors. The temperatures of the white ($T_W$) and black ($T_B$) sensors are recorded (Fig. 5(b)), and 107 pairs of dual thermistors are tested in total. The temperature rise by the irradiation is determined by the difference of the average temperature for the last 30 seconds (30 data) before the shutter is opened and closed. The mean temperature rise of the three repeated measurements is assigned as the RRT value for each pair of thermistors. The average ratio of the radiative heating of aluminium-coated and black sensors is 1:2.4 in the RRT experiment. Figures 5(c) and (d) show the temperature difference distributions between a pair of thermistors ($T_{B\_on} - T_{W\_on}$) and the temperature increase of the white sensor ($T_{W\_on} - T_{W\_off}$), respectively. The subscript on and off indicate when the light irradiation is turned on and off, respectively. These values are used as parameters for the sensor-specific radiation correction formulas obtained by UAS experiments. Although the irradiance is constant for each sensor, the cooling efficiency of the sensors may vary depending on the bead size of thermistors, air flow, and the pressure. Slight variations of air flow and/or pressure in the RRT chamber (not monitored) may partly be responsible for the observed distributions of radiative heating of the sensors in Figs. 5(c) and (d). Five representative pairs of thermistors are selected for the radiation correction experiments using UAS, as indicated by black arrows in Figs. 5(c) and (d).

## 4 Parameterisation for radiation measurement and correction by DTR using UAS

### 4.1 Radiation measurement by DTR

The DTR is installed upside down in the test chamber of the UAS with the thermistors and the sensor boom in parallel with the air flow but perpendicular to the irradiation. Figures 6(a)–(e) show the UAS measurements for five representative
temperature differences $(T_{B\_on} - T_{W\_on})$ between a pair of dual thermistors selected from the rotational radiation test (RRT) in Fig. 5(c). Previous studies reported that UAS has the capability of simultaneously varying environmental parameters for the radiation correction of commercial radiosondes (Lee et al., 2020; Lee et al., 2021). In Fig. 6, air pressure $(P)$ is varied from 5 to 500 hPa and temperature $(T_{W\_off})$ is varied from −68 to 20 °C with a fixed irradiance $(S_0 = 960$ W·m$^{-2})$ and ventilation speed $(v_0 = 5$ m·s$^{-1})$. As expected, the level of $(T_{B\_on} - T_{W\_on})_{RRT}$ is positively correlated with the degree of $(T_{B\_on} - T_{W\_on})_{UAS}$,
exhibiting a gradual decrease of $(T_{B\_on} - T_{W\_on})_{UAS}$ with decreasing $(T_{B\_on} - T_{W\_on})_{RRT}$ (Figs. 6(a)–(e)).

To parameterise the radiation measurement formula, $(T_{B\_on} - T_{W\_on})$ of the UAS is fitted with empirical equations as follows:

$$(T_{B\_on} - T_{W\_on})_{UAS} = T_0(T_{W\_on}) + A_0(T_{W\_on})\cdot\exp(-P\cdot P_0(T_{W\_on})^{-1}) + A_1(T_{W\_on})\cdot\exp(-P\cdot P_1(T_{W\_on})^{-1}), \quad (2)$$

where $T_0(T_{W\_on})$, $A_0(T_{W\_on})$, $P_0(T_{W\_on})$, $A_1(T_{W\_on})$ and $P_1(T_{W\_on})$ are the fitting coefficients being functions of $T_{W\_on}$ and units of °C, °C, hPa, °C and hPa, respectively. The dashed lines in Figs. 6(a)–(e) represent the fittings.

Interestingly, the level of $(T_{B\_on} - T_{W\_on})$ gradually increases as the temperature decreases especially for low pressures. A similar phenomenon was previously observed in a chamber with no apparent air ventilation (Lee et al., 2018a). The observed effect of temperature on $(T_{B\_on} - T_{W\_on})$ is because the convective heat transfer between the sensor and air is reduced as the thermal conductivity of the air is decreased at cold temperatures (Lee et al., 2021). To incorporate the effect of temperature $(T_{W\_on})$ in Eq. (2), its coefficients of $T_0(T_{W\_on})$, $A_0(T_{W\_on})$, $P_0(T_{W\_on})$, $A_1(T_{W\_on})$ and $P_1(T_{W\_on})$ are fitted with linear functions of
$T_{W\_on}$ as follows:

$$T_0(T_{W\_on}) = a_0\cdot T_{W\_on} + a_1 , \quad (3)$$

$$A_0(T_{W\_on}) = b_0\cdot T_{W\_on} + b_1 , \quad (4)$$

$$P_0(T_{W\_on}) = c_0\cdot T_{W\_on} + c_1 , \quad (5)$$

$$A_1(T_{W\_on}) = d_0\cdot T_{W\_on} + d_1 , \quad (6)$$

$$P_1(T_{W\_on}) = e_0\cdot T_{W\_on} + e_1 , \quad (7)$$

where $a_0$, $a_1$, $b_0$, $b_1$, $c_0$, $c_1$ $d_0$, $d_1$, $e_0$ and $e_1$ are the fitting coefficients. These coefficients are collected from five pairs of thermistors and each coefficient is again functionalised with $(T_{B\_on} - T_{W\_on})_{RRT}$ to incorporate the individuality of thermistors observed in RRT into Eq. (2) as follows:

$$Coefficient_{Rad\_meas} = Slope_{Rad\_meas}\cdot (T_{B\_on} - T_{W\_on})_{RRT} + Intercept_{Rad\_meas} , \quad (8)$$

where $Coefficient_{Rad\_meas}$ represents $a_0$, $a_1$, $b_0$, $b_1$, $c_0$, $c_1$ $d_0$, $d_1$, $e_0$ and $e_1$ from the five pairs of dual thermistors, and $Slope_{Rad\_meas}$ and $Intercept_{Rad\_meas}$ are the corresponding fitting coefficients. **Table 1** presents the $Slope_{Rad\_meas}$ and $Intercept_{Rad\_meas}$ values.

**Table 1**. $Slope_{Rad\_meas}$ and $Intercept_{Rad\_meas}$ of $a_0$, $a_1$, $b_0$, $b_1$, $c_0$, $c_1$ $d_0$, $d_1$, $e_0$ and $e_1$.

| Coefficient$_{Rad\_meas}$ | Unit | $Slope_{Rad\_meas}$ | $Intercept_{Rad\_meas}$ |
|---|---|---|---|
| $a_0$ | | 0 | $2.7 \times 10^{-1}$ |
| $a_1$ | °C | 0 | $3.3 \times 10^{-1}$ |
| $b_0$ | | $-8.8 \times 10^{-4}$ | $4.8 \times 10^{-1}$ |
| $b_1$ | °C | $-1.0 \times 10^{-3}$ | $6.3 \times 10^{-2}$ |
| $c_0$ | hPa·°C$^{-1}$ | $1.4 \times 10^{-2}$ | $-3.1 \times 10^{-1}$ |
| $c_1$ | hPa | $6.6 \times 10^{-3}$ | 17.8 |
| $d_0$ | | $-3.5 \times 10^{-4}$ | $4.3 \times 10^{-1}$ |
| $d_1$ | °C | $-1.5 \times 10^{-3}$ | $7.8 \times 10^{-2}$ |
| $e_0$ | hPa·°C$^{-1}$ | $-2.8 \times 10^{-1}$ | $-6.4$ |
| $e_1$ | hPa | $2.1 \times 10^{-1}$ | 235.8 |

During soundings, the irradiance ($S$) is unknown but can be found using ($T_B - T_W$) of DTR. Hence, Eq. (2) is employed to measure the *in-situ* irradiance using ($T_{B\_raw} - T_{W\_raw}$), where $T_{B\_raw}$ and $T_{W\_raw}$ are raw temperatures of the black and white sensors, respectively, based on the fact that the temperature difference between two sensors is linearly proportional to $S$. (Lee et al., 2018a; Lee et al., 2018b):

$$S = S_0 \times (T_{B\_raw} - T_{W\_raw}) \cdot (T_{B\_on} - T_{W\_on})_{UAS}^{-1}, \tag{9}$$

where the result of individual radiation test ($T_{B\_on} - T_{W\_on}$)$_{RRT}$ is incorporated using Eqs. (2)–(8). Consequently, Fig. 6(f) shows the fitting residual. Although the temperature difference of the five pairs of thermistors is different by nearly a factor of three, as shown in Figs. 6(a)–(e), the residuals are within ±20% due to the parameterisation of the RRT value into Eq. (9). In Eq. (9), the air ventilation speed ($v$) imitating the ascent speed of radiosondes is fixed at $v_0 = 5$ m·s$^{-1}$ and thus the effect of air ventilation cannot be identified in Fig. 6(f). As determined by a separate pair of thermistors, ($T_{B\_on} - T_{W\_on}$)$_{UAS}$ decreases
by 0.08 °C on average when $v$ increases by 1 m·s$^{-1}$ due to the convective cooling in the range of $v = 4$–6.5 m·s$^{-1}$ and $P = 7$–100 hPa (data not shown). Thus, Eq. (9) can be revised to include the effect of air ventilation speed as follows:

$$S = S_0 \times (T_{B\_raw} - T_{W\_raw}) \cdot [(T_{B\_on} - T_{W\_on})_{UAS} - 0.08 \cdot (v - v_0)]^{-1}, \tag{10}.$$

The standard deviation of the residual for a pair of thermistors is 4.1% with Eq. (9) while it is reduced to 3.4% with Eq. (10) when the air ventilation is actually changed (4–6.5 m·s$^{-1}$). The absolute value of the sensitivity coefficient (-0.08 °C / (m·s$^{-1}$))
against the ventilation speed will be significantly bigger when $v$ is lower than 4 m·s$^{-1}$ while it will be a bit smaller when $P$ is higher than 100 hPa. Note that Eq. (10) is used for the intercomparison sounding test, as described later.

## 4.2 Radiation correction by DTR

Figures 7(a)–(e) show the UAS measurements for obtaining the radiation correction values $(T_{W\_on} - T_{W\_off})$ of the white sensors that are selected from the RRT in Fig. 5(d). The experimental condition of Fig. 7 is identical to that of Fig. 6, wherein $P$ varies from 5 to 500 hPa and $T_{W\_off}$ varies from −68 to 20 °C with a fixed $S_0 = 960$ W·m$^{-2}$ and $v_0 = 5$ m·s$^{-1}$. Since the level of $(T_{W\_on} - T_{W\_off})_{RRT}$ is positively correlated with the degree of $(T_{W\_on} - T_{W\_off})_{UAS}$, $(T_{W\_on} - T_{W\_off})_{RRT}$ can be parameterised into a radiation correction formula based on $(T_{W\_on} - T_{W\_off})_{UAS}$ to neutralise the difference among units.

To obtain the radiation correction formula of DTR, $(T_{W\_on} - T_{W\_off})_{UAS}$ is fitted with empirical equations (dashed lines in Fig. 7) as follows:

$$(T_{W\_on} - T_{W\_off})_{UAS} = T_1(T_{W\_on}) + A_2(T_{W\_on}) \cdot \exp(-P \cdot P_2(T_{W\_on})^{-1}) + A_3(T_{W\_on}) \cdot \exp(-P \cdot P_3(T_{W\_on})^{-1}), \tag{11}$$

where $T_1(T_{W\_on})$, $A_2(T_{W\_on})$, $P_2(T_{W\_on})$, $A_3(T_{W\_on})$ and $P_3(T_{W\_on})$ are the fitting coefficients as a function of $T_{W\_on}$ with units of °C, °C, hPa, °C and hPa, respectively.

$(T_{W\_on} - T_{W\_off})_{UAS}$ shows a temperature dependency. $(T_{W\_on} - T_{W\_off})_{UAS}$ at −68 °C is 118.9 ± 3.5% (mean ± SD of five units) of that at 20 °C, when $P = 5$ hPa. In the previous study, the ratio for RS41 investigated by the same manner is 119% (Lee et al., 2021). This is attributed to the decrease of the thermal conductivity of air at cold temperatures, which reduces the heat transfer from the sensor to air despite the constant irradiation.

To incorporate the effect of temperature $(T_{W\_on})$ in Eq. (11), the coefficients of $T_1(T_{W\_on})$, $A_2(T_{W\_on})$, $P_2(T_{W\_on})$, $A_3(T_{W\_on})$ and $P_3(T_{W\_on})$ are fitted with linear functions of $T_{W\_on}$ as follows:

$$T_1(T_{W\_on}) = f_0 \cdot T_{W\_on} + f_1 , \tag{12}$$

$$A_2(T_{W\_on}) = g_0 \cdot T_{W\_on} + g_1 , \tag{13}$$

$$P_2(T_{W\_on}) = h_0 \cdot T_{W\_on} + h_1 , \tag{14}$$

$$A_3(T_{W\_on}) = i_0 \cdot T_{W\_on} + i_1 , \tag{15}$$

$$P_3(T_{W\_on}) = j_0 \cdot T_{W\_on} + j_1 , \tag{16}$$

where $f_0$, $f_1$, $g_0$, $g_1$, $h_0$, $h_1$ $i_0$, $i_1$, $j_0$ and $j_1$ are the fitting coefficients. Then, these coefficients are collected from five pairs of thermistors and each coefficient is converted into a function of $(T_{W\_on} - T_{W\_off})_{RRT}$ to incorporate the individuality of thermistors into Eq. (11):

$$Coefficient_{\_Rad\_cor} = Slope_{\_Rad\_cor} \cdot (T_{W\_on} - T_{W\_off})_{RRT} + Intercept_{\_Rad\_cor} , \tag{17}$$

where $Coefficient_{\_Rad\_cor}$ represents $f_0$, $f_1$, $g_0$, $g_1$, $h_0$, $h_1$ $i_0$, $i_1$, $j_0$ and $j_1$, and $Slope_{\_Rad\_cor}$ and $Intercept_{\_Rad\_cor}$ represent the corresponding fitting coefficients. **Table 2** presents the $Slope_{\_Rad\_cor}$ and the $Intercept_{\_Rad\_cor}$ values.

**Table 2**. $Slope_{\_Rad\_cor}$ and $Intercept_{\_Rad\_cor}$ of $f_0$, $f_1$, $g_0$, $g_1$, $h_0$, $h_1$ $i_0$, $i_1$, $j_0$ and $j_1$.

| Coefficient_Rad_cor | Unit | Slope_Rad_cor | Intercept_Rad_cor |
|---|---|---|---|
| $f_0$ | | 0 | $3.0 \times 10^{-1}$ |
| $f_1$ | °C | 0 | $-1.3 \times 10^{-1}$ |

| | | | |
|---|---|---|---|
| $g_0$ | | $-2.2 \times 10^{-3}$ | $4.7 \times 10^{-1}$ |
| $g_1$ | °C | $1.5 \times 10^{-3}$ | $9.0 \times 10^{-2}$ |
| $h_0$ | hPa·°C$^{-1}$ | $-1.9 \times 10^{-2}$ | $-1.7 \times 10^{-2}$ |
| $h_1$ | hPa | $3.1 \times 10^{-2}$ | 9.5 |
| $i_0$ | | $-7.7 \times 10^{-4}$ | $4.0 \times 10^{-1}$ |
| $i_1$ | °C | $-3.7 \times 10^{-4}$ | $-6.4 \times 10^{-2}$ |
| $j_0$ | hPa·°C$^{-1}$ | $-3.1 \times 10^{-1}$ | $-7.0$ |
| $j_1$ | hPa | $6.2 \times 10^{-1}$ | 135.6 |

Although the irradiance ($S_0$) is fixed as 960 W·m$^{-2}$ herein, ($T_{W\_on} - T_{W\_off}$)$_{UAS}$ is linearly proportional to $S$, as experimentally and theoretically studied in previous studies (Mcmillin et al., 1992; Lee et al., 2018b). To include the irradiance ($S$) obtained using Eq. (10) into the radiation correction formula, Eq. (11) is revised as follows:

$$(T_{W\_raw} - T_{W\_cor}) = (S \cdot S_0^{-1}) \times (T_{W\_on} - T_{W\_off})_{UAS}, \tag{18}$$

where $T_{W\_raw}$ and $T_{W\_cor}$ are the raw temperature and radiation-corrected temperature of the white sensor. The result of individual radiation test ($T_{W\_on} - T_{W\_off}$)$_{RRT}$ is incorporated using Eqs. (11)–(17).

The fitting residual obtained using Eq. (18) is shown in Fig. 7(f). Although the radiation correction values of the five pairs of thermistors differ by more than a factor of two, as shown in Figs. 5(a)–(e), the residuals are within ±0.2 °C, because the RRT

results are incorporated into Eq. (18).

The air ventilation speed ($v$) imitating the ascent speed of the radiosondes is fixed at $v_0 = 5$ m·s$^{-1}$ in Eq. (18) and thus the effect of air ventilation cannot be identified in Fig. 7(f). As determined by a separate pair of thermistors, ($T_{W\_on} - T_{W\_off}$)$_{UAS}$ decreases by 0.1 °C on average when $v$ increases by 1 m·s$^{-1}$ due to the convective cooling in the range of $v = 4$–6.5 m·s$^{-1}$ and $P = 7$–100 hPa (data not shown). Thus, Eq. (18) can be modified to include the effect of $v$ as follows:

$$(T_{W\_raw} - T_{W\_cor}) = (S \cdot S_0^{-1}) \times [(T_{W\_on} - T_{W\_off})_{UAS} - 0.1 \cdot (v - v_0)], \tag{19}.$$

When the air ventilation is actually changed (4–6.5 m·s$^{-1}$), the standard deviation of the residual for a pair of thermistors is 0.10 °C with Eq. (18) while it is reduced to 0.04 °C with Eq. (19). The absolute value of the sensitivity coefficient (-0.1 °C / (m·s$^{-1}$)) against the ventilation speed will be significantly bigger when $v$ is lower than 4 m·s$^{-1}$ while it will be a bit smaller when $P$ is higher than 100 hPa. Note that Eq. (19) is applied to the DTR radiation correction in the intercomparison sounding

test, as described later.

## 5 Sounding test of DTR

### 5.1 Daytime

The radiation measurement and correction formulas of the DTR obtained via laboratory characterisations were applied to the sounding test performed during July, 2021 in Jeju Island, South Korea. One, two, or three DTRs were tested in parallel with a RS41 in a single flight. The number of comparison ($N$) was $N = 12$ at daytime and $N = 6$ at nighttime from 7 and 3 soundings, respectively. The daytime sounding was performed from 11:00 am to 5 pm local time while the nighttime sounding was from 12:00 am to 4 am. The sky was normally cloudy. Figure 8(a) shows an example of the temperature difference ($T_{B\_raw} - T_{W\_raw}$) between the two sensors during sounding in the daytime. Note that ($T_{B\_raw} - T_{W\_raw}$) in the sounding data corresponds to the ($T_{B\_on} - T_{W\_on}$)$_{UAS}$ of the UAS experiment. Figure 8(b) displays the irradiance measured by the DTR based on the temperature difference between the dual thermistors and environmental parameters, including $T_{W\_raw}$, $P$ and $v$. The irradiance measured by the DTR is the net effective irradiance to/from the thermistors including the components of direct solar irradiation, its reflection and scattering, the long-wave radiation from the earth, and the long-wave radiation from the thermistors. However, these components cannot be distinguished through DTR measurements. The radiation correction formula of the DTR is obtained based on the portion of the long-wave and the short-wave radiation from the solar simulator used as a radiation source in the UAS experiments. The emissivity and absorptivity are dependent on the wavelength. In this regard, the radiative heating of the DTR in soundings can be affected by the actual ratio of the long-wave and the short-wave radiation. For aluminium coating, the reflectance was 0.8−0.9 below 1000 nm and 0.9 above 1000 nm in wavelength. This means that the influence of the ratio between the long- and short-wave radiation would be a few percent of the radiative heating of the DTR even when the portion below 1000 nm is drastically different between the laboratory experiments and soundings. Then, using the effective irradiance ($S$), the radiation correction value ($T_{W\_raw} - T_{W\_cor}$) of the white sensor is obtained using Eq. (19), as shown in Fig. 8(c). The correction value of the white sensor tends to gradually increase from the ground to stratosphere with some fluctuations in the troposphere due to clouds.

### 5.2 Nighttime

The radiation measurement and correction formulas of the DTR were also applied during the nighttime sounding test. Figure 8(d) shows a typical example of the temperature difference ($T_{B\_raw} - T_{W\_raw}$) between dual thermistors during nighttime. Interestingly, at an altitude of 30 km, the temperature of the black sensor is lower than that of the white sensor by about 0.5 °C. This phenomenon occurs due to the high emissivity of the black sensor, which facilitates a long-wave radiation from the black sensor more than that from the white sensor. The ($T_{B\_raw} - T_{W\_raw}$) converted into the effective irradiance at nighttime using Eq. (10) is negative (Fig. 8(e)). Additionally, the radiation correction value ($T_{W\_raw} - T_{W\_cor}$) of the white sensor obtained using Eq. (19) is negative (Fig. 8(f)). This implies that the raw temperature of the white sensor is lower than the air temperature, and thus, the absolute correction value should be added to $T_{W\_raw}$ for the correction. The negative net irradiance at nighttime was also observed in the previous work for the radiation correction of radiosondes based on the

measurement of radiative flux profiles using two pyranometers for measuring downward and upward solar short-wave radiation, and two pyrgeometers for measuring upward and downward thermal long-wave radiation (Philipona et al., 2013).

The long-wave radiation balance (LRB) of the sensor defined by the sum of the absorbed and emitted fluxes corresponds to the effective irradiance at nighttime in this work. Both the LRB in the work of Philipona *et al.* and the effective irradiance at nighttime in this work are negative in the lower troposphere, and then become positive and again negative further up in the stratosphere. This means that temperature sensors of radiosondes are cooled in lower troposphere, warmed in higher up, and again cooled further up in the stratosphere and thus should be corrected accordingly. The profile of the LRB at nighttime was

similar to that of daytime (Philipona et al., 2013). In this regard, the decrease of the effective irradiance in the stratosphere observed at daytime is highly likely due to the negative LRB of the sensors as observed in the nighttime soundings.

## 6 Uncertainty evaluation and intercomparison

### 6.1 Uncertainty budget on radiation measurement by DTR

According to the radiation measurement formula by the DTR (Eq. (10)), the factors for the uncertainty of radiation

measurement $U(S)$ are $T_{\mathrm{W\_on}}$, $P$, $v$, $S_0$ and fitting residuals in Fig. 6(f). These factors contribute to $U(S)$ as follows:

$$\frac{\partial S}{\partial T_{\mathrm{W\_on}}} \cdot U\left(T_{\mathrm{W\_on}}\right), \tag{20}$$

$$\frac{\partial S}{\partial P} \cdot U(P), \tag{21}$$

$$\frac{\partial S}{\partial v} \cdot U(v), \tag{22}$$

$$\frac{\partial S}{\partial S_0} \cdot U(S_0), \tag{23}$$

$$\frac{S}{100} \cdot U(Fitting). \tag{24}$$

Here, $U(\mathrm{parameter})$ represents the expanded uncertainty of each parameter at $k = 2$, and the partial differential terms represent the sensitivity coefficients. The sensitivity coefficient of the uncertainty due to the fitting error $U(Fitting)$ is $S/100$ because it is provided as a percentage in Fig. 6(f). Then, $U(S)$ is obtained by combining the contributions from these factors based on the uncertainty propagation law:

$$U(S) = \sqrt{\left(\frac{\partial S}{\partial T_{T_{\mathrm{W\_on}}}}\right)^2 \cdot U\left(T_{\mathrm{W\_on}}\right)^2 + \left(\frac{\partial S}{\partial P}\right)^2 \cdot U(P)^2 + \left(\frac{\partial S}{\partial v}\right)^2 \cdot U(v)^2 + \left(\frac{\partial S}{\partial S_0}\right)^2 \cdot U(S_0)^2 + \left(\frac{S}{100}\right)^2 \cdot U(Fitting)^2} \tag{25}$$

Figures 9(a) and (b) show the average of the effective irradiance measured by DTR with the expanded uncertainty ($k = 2$) calculated using Eq. (25) at daytime and nighttime, respectively. Radiation measurements by DTR from $N = 12$ and $N = 6$ are averaged at daytime and nighttime, respectively. The negative effective irradiance at an altitude above 25 km at nighttime is statistically significant. The long-wave radiation imbalance (cooling) of sensors at stratosphere should not be the

375 issue of the DTR only because the same phenomenon was observed in the previous work based on the measurement of long-wave radiations. These findings suggest that an application of radiation correction is needed even at nighttime. Examples of

the uncertainty budget for the radiation measurement by the DTR at an altitude of 30 km are summarised in **Table 3** and **Table 4** for daytime and nighttime, respectively.

**Table 3.** Daytime uncertainty budget on radiation measurement of $S = 1141$ W·m$^{-2}$ by DTR at an altitude of 30 km.

| Uncertainty factor | Condition at 30 km | Unit | Uncertainty ($k = 2$) | Contribution to uncertainty of radiation measurement ($k = 2$) |
|---|---|---|---|---|
| $T_{W\_on}$ | −41.5 | °C | 0.23 | 106 W·m$^{-2}$ |
| $P$ | 12.6 | hPa | 0.3 | 5 W·m$^{-2}$ |
| $v$ | 6.1 | m·s$^{-1}$ | 0.12 | 5 W·m$^{-2}$ |
| $S_0$ | 960 | W·m$^{-2}$ | 61 | 73 W·m$^{-2}$ |
| Fitting error | - | % | 23.4 | 268 W·m$^{-2}$ |
| $U(S)$, Expanded uncertainty for radiation measurement of 1141 W·m$^{-2}$ ($k = 2$) | | | | 297 W·m$^{-2}$ |

**Table 4.** Nighttime uncertainty budget on radiation measurement of $S = -198$ W·m$^{-2}$ by DTR at an altitude of 30 km.

| Uncertainty factor | Condition at 30 km | Unit | Uncertainty ($k = 2$) | Contribution to uncertainty of radiation measurement ($k = 2$) |
|---|---|---|---|---|
| $T_{W\_on}$ | −44.3 | °C | 0.24 | 110 W·m$^{-2}$ |
| $P$ | 12.3 | hPa | 0.3 | 1 W·m$^{-2}$ |
| $v$ | 6.3 | m·s$^{-1}$ | 0.12 | 1 W·m$^{-2}$ |
| $S_0$ | 960 | W·m$^{-2}$ | 61 | 12.5 W·m$^{-2}$ |
| Fitting error | - | % | 23.4 | 46 W·m$^{-2}$ |
| $U(S)$, Expanded uncertainty for radiation measurement of −198 W·m$^{-2}$ ($k = 2$) | | | | 120 W·m$^{-2}$ |

**6.2 Uncertainty on radiation correction by DTR**

The radiation-corrected temperature ($T_{W\_cor}$) of DTR is obtained by subtracting the radiation correction value calculated using Eq. (19) from the raw temperature ($T_{W\_raw}$) of the white sensor:

$$T_{W\_cor} = T_{W\_raw} - S \cdot S_0^{-1} \cdot [(T_{W\_on} - T_{W\_off})_{UAS} - 0.1 \cdot (v - v_0)] \tag{26}$$

Then, the uncertainty of the corrected temperature $U(T_{W\_cor})$ is calculated as follows:

$$U(T_{W\_cor})^2 = U(T_{W\_raw})^2 + U(S \cdot S_0^{-1} \cdot [(T_{W\_on} - T_{W\_off})_{UAS} - 0.1 \cdot (v - v_0)])^2 \tag{27}$$

where $U(parameter)$ is the expanded uncertainty ($k = 2$). $U(T_{W\_raw})^2$ is the uncertainty of raw temperature that is related to the uncertainty due to the calibration $U(T_{S\_cal})^2$ of thermistors in the climate chamber (Fig. 3) and the uncertainty due to the temperature effect on the radiosonde board $U(T_{Board\_temp})^2$ (Fig. 4). The second term in the right is the uncertainty of the

radiation correction value that is obtained using Eq. (19), comprising uncertainty factors $T_{W\_on}$, $P$, $v$ and $S_0$ and fitting residuals. The fitting residuals include the uncertainty due to RRT $U(T_{RRT})^2$ (Fig. 7). Consequently, the expanded uncertainty of the corrected temperature of the DTR is as follows:

$$U(T_{W\_cor}) = \sqrt{\begin{array}{c}\left(\frac{\partial T_{W\_cor}}{\partial T_{W\_on}}\right)^2 \cdot U(T_{W\_on})^2 + \left(\frac{\partial T_{W\_cor}}{\partial P}\right)^2 \cdot U(P)^2 + \left(\frac{\partial T_{W\_cor}}{\partial v}\right)^2 \cdot U(v)^2 + \left(\frac{\partial T_{W\_cor}}{\partial S_0}\right)^2 \cdot U(S_0)^2 \\ + 1^2 \cdot U(T_{RRT})^2 + 1^2 \cdot U(T_{S\_cal})^2 + 1^2 \cdot U(T_{Board\_temp})^2\end{array}} \tag{28}$$

Figures 9(c) and (d) show $U(T_{W\_cor})$ and its uncertainty components at daytime and nighttime, respectively. At daytime, the DTR uncertainty gradually increases up to about 0.35 °C at the tropopause and is maintained in the stratosphere (0.33 °C at 30 km). However, at nighttime, the uncertainty slightly is decreased to 0.3 °C at the tropopause and 0.25 °C at 30 km as the uncertainty of effective irradiance is decreased. Examples of the uncertainty budget on the radiation-corrected temperature of the DTR ($T_{W\_cor}$) at an altitude of 30 km are summarised in **Table 5** and **Table 6** for daytime and nighttime, respectively.

**Table 5.** Uncertainty budget on radiation-corrected temperature by DTR at daytime at an altitude of 30 km.

| Uncertainty factor | Condition at 30 km | Unit | Uncertainty ($k = 2$) | Contribution to uncertainty of radiation-corrected temperature ($k = 2$) |
|---|---|---|---|---|
| $T_{W\_on}$ | −41.5 | °C | 0.23 | 0.000 °C |
| $P$ | 12.6 | hPa | 0.3 | 0.004 °C |
| $v$ | 6.1 | m·s$^{-1}$ | 0.12 | 0.01 °C |
| $S_0$ | 960 | W·m$^{-2}$ | 61 | 0.06 °C |
| Fitting error (or $T_{RRT}$) | - | °C | 0.216 | 0.23 °C |
| $T_{S\_cal}$ | −41.5 | °C | 0.227 | 0.23 °C |
| $T_{Board\_temp}$ | −41.5 | °C | 0.08 | 0.08 °C |
| $U(T_{W\_cor})$, Expanded uncertainty for radiation-corrected temperature ($k = 2$) | | | | 0.33 °C |

**Table 6.** Uncertainty budget on radiation-corrected temperature by DTR at nighttime at an altitude of 30 km.

| Uncertainty factor | Condition at 30 km | Unit | Uncertainty ($k = 2$) | Contribution to uncertainty of radiation-corrected temperature ($k = 2$) |
|---|---|---|---|---|
| $T_{W\_on}$ | −44.3 | °C | 0.24 | 0.000 °C |
| $P$ | 12.3 | hPa | 0.3 | 0.001 °C |
| $v$ | 6.3 | m·s$^{-1}$ | 0.12 | 0.002 °C |
| $S_0$ | 960 | W·m$^{-2}$ | 61 | 0.01 °C |
| Fitting error (or $T_{RRT}$) | - | °C | 0.216 | 0.04 °C |
| $T_{S\_cal}$ | −44.3 | °C | 0.227 | 0.24 °C |

| $T_{\text{Board\_temp}}$ | −44.3 | °C | 0.08 | 0.08 °C |
|---|---|---|---|---|
| $U(T_{\text{W\_cor}})$, Expanded uncertainty for radiation-corrected temperature ($k = 2$) | | | | 0.25 °C |

Altitude-dependent $U(T_{\text{W\_cor}})$ of DTR ($k = 2$) at both daytime and nighttime is summarised in **Table 7**. The uncertainty at the tropopause (~15 km) is higher than other regions mainly because the calibration uncertainty of the thermistors increases as the temperature is lowered (Fig. 3(e)). This means that a reduction of the calibration uncertainty of massive amount of thermistors is needed to improve the uncertainty of radiation-corrected temperature of the DTR.

**Table 7.** $U(T_{\text{W\_cor}})$ of DTR ($k = 2$) at daytime and nighttime.

| Altitude | $U(T_{\text{W\_cor}})$ / $T_{\text{W\_cor}}$ at daytime | $U(T_{\text{W\_cor}})$ / $T_{\text{W\_cor}}$ at nighttime |
|---|---|---|
| 0 km | 0.14 °C / 24.6 °C | 0.14 °C / 22.8 °C |
| 5 km | 0.17 °C / 0.1 °C | 0.15 °C / 0.4 °C |
| 10 km | 0.24 °C / −29.3 °C | 0.22 °C / −32.0 °C |
| 15 km | 0.34 °C / −68.0 °C | 0.31 °C / −66.8 °C |
| 20 km | 0.35 °C / −63.2 °C | 0.30 °C / −62.5 °C |
| 25 km | 0.34 °C / −50.8 °C | 0.27 °C / −51.5 °C |
| 30 km | 0.33 °C / −42.5 °C | 0.25 °C / −44.1 °C |

**6.3 Intercomparison of DTR with Vaisala RS41**

The radiation-corrected temperature of DTR ($T_{\text{W\_cor}} = T_{\text{DTR}}$) is compared to that of a commercial radiosonde (Vaisala, RS41) via parallel sounding. Figures 9(e) and (f) display the difference between the DTR and RS41 temperatures ($T_{\text{DTR}} − T_{\text{RS41}}$) with the DTR uncertainty ($k = 2$) as error bars during the daytime and nighttime, respectively. Generally, the two temperatures are within the DTR uncertainty during both daytime and nighttime. The manufacturer specifies that the uncertainty of RS41 is 0.3 °C in 0−16 km in altitude and 0.4 °C above 16 km (Vaisala). Then, the combined uncertainty of the RS41 (0.4 °C) and the DTR (0.33−35 °C) is 0.52−0.53 °C ($k = 2$) at 16 km and higher up. Thus, the observed differences between the RS41 and the DTR are within their combined uncertainty at daytime. Nevertheless, the radiation-corrected temperature of DTR is about 0.4 °C higher than that of RS41 around 30 km at daytime. A similar trend is observed in the radiation correction of the RS41 radiosonde by the GRUAN using the SISTER setup (Von Rohden et al., 2022). The radiation-corrected temperature of the RS41 obtained by the GRUAN is 0.35 °C warmer than that provided by Vaisala at 35 km although the difference of temperature between the GRUAN and Vaisala is within their combined uncertainty.

Recently, we have obtained a radiation correction formula of RS41 under a well-defined irradiance in the UAS (Lee et al., 2021). However, the correction formula cannot be applied to RS41 because the irradiance and its uncertainty in soundings are unknown. In this regard, the GRUAN uses a simulated irradiance calculated by the average of clear and cloudy sky cases for their radiation correction of RS41 (Von Rohden et al., 2022). The maximum uncertainty of RS41 by the GRUAN is about 0.3 °C at $k = 2$ which is larger than our previous work on RS41 (0.17 °C at $k = 2$). This is because the irradiance in our work is assumed to be 1360 W·m$^{-2}$ at stratosphere with a small uncertainty obtained by the laboratory experiments corresponding to the irradiance. Therefore, one of the prerequisites to the uncertainty evaluation on the radiation correction is to know the irradiance and its uncertainty in soundings. This work may contribute to improving the measurement of the irradiance and the estimation of its uncertainty using dual thermistor radiosondes.

## 7 Conclusions

The performance and uncertainty of DTR were evaluated via a series of laboratory setups and intercomparison sounding with a commercial radiosonde (Vaisala, RS41). The DTR comprises two temperature sensors (white and black) with different emissivities; their temperature difference can be used for the *in-situ* measurement of the effective irradiance and the correction of the radiation-induced bias of the white sensor. The thermistors were individually calibrated in the range of −70–30 °C in a climate chamber, and the uncertainty due to the calibration was evaluated. Moreover, the effect of temperature on resistance reading by radiosonde boards was investigated from −70 to 20 °C in the climate chamber, and the corresponding uncertainty was evaluated. RRT was individually performed on the thermistors to compensate for the unit difference. Parameterisation of the radiation measurements and correction formulas of DTR was performed via UAS experiments with varying temperature, pressure and ventilation speed. The fitting residual of the five DTRs selected from RRT was within 0.2 °C. The radiation measurement and correction formulas obtained by UAS were applied to the sounding test of DTR conducted in July, 2021. The method of obtaining the radiation-correction value of DTR using the effective irradiance measured by the temperature difference between dual sensors during sounding was discussed. Then, the contributing uncertainty factors on the corrected temperature of DTR were summarised for both daytime and nighttime. Generally, the uncertainty of the radiation-corrected temperature of DTR was about 0.35 °C at daytime and 0.3 °C at nighttime with the coverage factor $k = 2$. The corrected temperature of the DTR was about 0.4 °C higher than that of RS41 around 30 km at daytime although the difference is within the combined uncertainty (~0.5 °C at $k = 2$) of the RS41 and the DTR. The DTR methodology aims at enhancing the accuracy of the temperature measurement in the upper air based on *in-situ* radiation measurements. Future works may include more parallel sounding tests in various conditions including cloudy and windy weather to better characterise the performance of the DTR. Especially, the radiation correction of the DTR is expected to be different from others while/after passing through clouds because the DTR responds to an *in-situ* radiation flux. Moreover, an extension of the environmental ranges, such as temperature and pressure, is desirable to cover the upper air environments of global areas.

*Acknowledgement*

This work was supported by the Korea Research Institute of Standards and Science (Grant no. GP2021-0005-02).

*Author contribution*

SL analysed the experimental data and wrote the manuscript. SK, YL, JY and JS conducted the experiments. CB revised the experimental setup. SL and SK developed the measurement software. YK designed the experiments.

*Competing interests*

The authors declare that they have no conflicts of interest.

**Referencesre**

Bojinski, S., Verstraete, M., Peterson, T. C., Richter, C., Simmons, A., and Zemp, M.: The concept of essential climate variables in support of climate research, applications, and policy, Bulletin of the American Meteorological Society, 95, 1431-1443, https://doi.org/10.1175/BAMS-D-13-00047.1, 2014.

Cuccaro, R., Rosso, L., Smorgon, D., Beltramino, G., Tabandeh, S., and Fernicola, V.: Development of a low frost-point generator operating at sub-atmospheric pressure, Measurement Science and Technology, 29, 054002, 10.1088/1361-6501/aaa785, 2018.

de Podesta, M., Bell, S., and Underwood, R.: Air temperature sensors: dependence of radiative errors on sensor diameter in precision metrology and meteorology, Metrologia, 55, 229, https://doi.org/10.1088/1681-7575/aaaa52, 2018.

Dirksen, R., Sommer, M., Immler, F., Hurst, D., Kivi, R., and Vömel, H.: Reference quality upper-air measurements: GRUAN data processing for the Vaisala RS92 radiosonde, Atmospheric Measurement Techniques, 7, 4463-4490, https://doi.org/10.5194/amt-7-4463-2014, 2014.

GCOS: GCOS Reference Upper-Air Network (GRUAN): Justification, requirements, siting and instrumentation options, Available at: https://library.wmo.int/doc_num.php?explnum_id=3821 [Accessed 5th August 2021], 2007.

Key, J. R. and Schweiger, A. J.: Tools for atmospheric radiative transfer: Streamer and FluxNet, Computers & Geosciences, 24, 443-451, https://doi.org/10.1016/S0098-3004(97)00130-1, 1998.

Lee, S.-W., Kim, S., Lee, Y.-S., Choi, B. I., Kang, W., Oh, Y. K., Park, S., Yoo, J.-K., Lee, J., Lee, S., Kwon, S., and Kim, Y.-G.: Radiation correction and uncertainty evaluation of RS41 temperature sensors by using an upper-air simulator, Atmos. Meas. Tech. Discuss. [preprint], https://doi.org/10.5194/amt-2021-246, in review, 2021.

Lee, S. W., Park, E. U., Choi, B. I., Kim, J. C., Woo, S. B., Park, S., Yang, S. G., and Kim, Y. G.: Dual temperature sensors with different emissivities in radiosondes for the compensation of solar irradiation effects with varying air pressure, Meteorological Applications, 25, 49-55, https://doi.org/10.1002/met.1668, 2018a.

Lee, S. W., Park, E. U., Choi, B. I., Kim, J. C., Woo, S. B., Kang, W., Park, S., Yang, S. G., and Kim, Y. G.: Compensation of solar radiation and ventilation effects on the temperature measurement of radiosondes using dual thermistors, 490 Meteorological Applications, 25, 209-216, https://doi.org/10.1002/met.1683, 2018b.

Lee, S. W., Yang, I., Choi, B. I., Kim, S., Woo, S. B., Kang, W., Oh, Y. K., Park, S., Yoo, J. K., and Kim, J. C.: Development of upper air simulator for the calibration of solar radiation effects on radiosonde temperature sensors, Meteorological Applications, 27, e1855, https://doi.org/10.1002/met.1855, 2020.

Luers, J. K.: Estimating the temperature error of the radiosonde rod thermistor under different environments, Journal of Atmospheric and Oceanic Technology, 7, 882-895, https://doi.org/10.1175/1520-0426(1990)007<0882:ETTEOT>2.0.CO;2, 1990.

McMillin, L., Uddstrom, M., and Coletti, A.: A procedure for correcting radiosonde reports for radiation errors, Journal of Atmospheric and Oceanic Technology, 9, 801-811, https://doi.org/10.1175/1520-0426(1992)009<0801:APFCRR>2.0.CO;2, 1992.

Merlone, A., Lopardo, G., Sanna, F., Bell, S., Benyon, R., Bergerud, R. A., Bertiglia, F., Bojkovski, J., Böse, N., and Brunet, M.: The MeteoMet project–metrology for meteorology: challenges and results, Meteorological Applications, 22, 820-829, https://doi.org/10.1002/met.1528, 2015.

Merlone, A., Sanna, F., Beges, G., Bell, S., Beltramino, G., Bojkovski, J., Brunet, M., Del Campo, D., Castrillo, A., and Chiodo, N.: The MeteoMet2 project—highlights and results, Measurement Science and Technology, 29, 025802, 10.1088/1361-6501/aa99fc, 2018.

Nash, J., Oakley, T., Vömel, H., and Li, W.: WMO intercomparisons of high quality radiosonde systems., WMO/TD-1580, Available at: https://library.wmo.int/doc_num.php?explnum_id=9467, 2011.

Philipona, R., Kräuchi, A., Romanens, G., Levrat, G., Ruppert, P., Brocard, E., Jeannet, P., Ruffieux, D., and Calpini, B.: Solar and thermal radiation errors on upper-air radiosonde temperature measurements, Journal of Atmospheric and Oceanic Technology, 30, 2382-2393, https://doi.org/10.1175/JTECH-D-13-00047.1, 2013.

Sairanen, H., Heinonen, M., and Högström, R.: Validation of a calibration set-up for radiosondes to fulfil GRUAN requirements, Measurement Science and Technology, 26, 105901, https://doi.org/10.1088/0957-0233/26/10/105901, 2015.

Schmidlin, F. J., Luers, J. K., and Huffman, P.: Preliminary estimates of radiosonde thermistor errors, Available at: https://ntrs.nasa.gov/api/citations/19870002653/downloads/19870002653.pdf [Accessed 6th August 2021]. 1986.

Vaisala: Vaisala Radiosonde RS41 Measurement Performance, Available at: file:///C:/Users/%EC%9D%B4%EC%83%81%EC%9A%B1/Downloads/WEA-MET-RS41-Performance-White-paper-B211356EN-B-LOW-v3%20(1).pdf [Accessed 3rd August 2021].

von Rohden, C., Sommer, M., Naebert, T., Motuz, V., and Dirksen, R. J.: Laboratory characterisation of the radiation temperature error of radiosondes and its application to the GRUAN data processing for the Vaisala RS41, Atmospheric Measurement Techniques, 15, 383-405, https://doi.org/10.5194/amt-15-383-2022, 2022.

White, D.: Interpolation errors in thermistor calibration equations, International Journal of Thermophysics, 38, 59, 10.1007/s10765-017-2194-x, 2017.

Yang, I., Kim, S., Lee, Y. H., and Kim, Y.-G.: Simplified calibration process and uncertainty assessment for sampling large numbers of single-use thermistors for upper-air temperature measurement, Measurement Science and Technology, 32, 045002, 2021.

**Figure captions**

**Figure 1.** (a) Dual thermistor radiosonde (DTR) with a white and black sensor and (b) operation principle of DTR for irradiance measurement and correction of radiation effect based on the measured irradiance. The temperature difference between the dual thermistors ($T_{B\_raw} - T_{W\_raw}$) is linearly proportional to the irradiance, and the radiation-induced heating of the white sensor ($T_{W\_raw} - T_{W\_cor}$) is corrected based on the irradiance measured by ($T_{B\_raw} - T_{W\_raw}$).

**Figure 2.** Characterisations of DTR. (a) Individual calibration of thermistors in a climate chamber, (b) test of the effect of temperature on the resistance reading using the radiosonde boards in the climate chamber, (c) radiation test on individual thermistors, (d) parameterisation of radiation measurement and correction formulae using an upper air simulator and (e) sounding test by applying laboratory characterisation results.

**Figure 3**. Calibration of individual thermistors in a climate chamber. (a) Calibration setup showing thermistors on booms (left), a digital multimeter to read the sensor resistance (top-right) and a data acquisition computer (bottom-right). (b) Maximum temperature deviations ($T_{Ref\_devi}$) with respect to the average of five reference thermometers ($T_{Ref\_aver}$) as a function of $T_{Ref\_aver}$. Distribution of the residuals of the (c) white and (d) black sensors by individually applying the calibration curves. (e) The uncertainty budget on the radiosonde thermistor calibration, $U(T_{S\_cal})$, with a coverage factor $k = 2$. Uncertainty factors including reference temperature deviations $U(T_{Ref\_devi})$, stability $U(T_{Ref\_stab})$ and calibration $U(T_{Ref\_cal})$, radiosonde sensor stability $U(T_{S\_stab})$ and fitting residual $U(T_{S\_fit\_resid})$ are considered for $U(T_{S\_cal})$.

**Figure 4**. Test of the temperature effect on resistance reading by radiosonde boards. (a) Test setup showing the radiosonde boards in a climate chamber (left), reference resistors (top-right) and a data acquisition computer (bottom-right). (b) Difference between the reference resistance and radiosonde reading as a function of the reference resistance. (c) Residual after conversion of resistance to temperature as a function of temperature.

**Figure 5.** Rotational radiation test (RRT) on radiosonde thermistors individually. (a) RRT setup showing the radiosonde thermistors in a chamber, solar simulator, and vacuum pump (left), a digital multimeter (top-right) and a data acquisition computer (bottom-right). (b) Temperature measured by a white ($T_W$) and black ($T_B$) sensor with/without light irradiation by the solar simulator. (c) Distribution of the temperature difference between the paired white and black sensors. (d) Distribution of the temperature increase of white sensors by the irradiation. Five pairs of a white and black sensor were selected for radiation correction experiments using an upper air simulator (UAS), as indicated by black arrows in c and d.

**Figure 6.** Temperature difference between paired white and black sensors ($T_{B\_on} - T_{W\_on}$) investigated using UAS. (a–e) ($T_{B\_on} - T_{W\_on}$) of the five paired radiosonde thermistors as a function of air pressure with varying temperature. (f) Residual of irradiance calculated on the basis of ($T_{B\_on} - T_{W\_on}$) obtained in UAS and the rotational radiation test.

**Figure 7**. Radiation correction value of white sensors ($T_{\text{W\_on}} - T_{\text{W\_off}}$) investigated using UAS. (a–e) ($T_{\text{W\_on}} - T_{\text{W\_off}}$) of the five radiosonde white sensors as a function of air pressure with varying temperature. (f) Residual of correction value calculated on the basis of ($T_{\text{W\_on}} - T_{\text{W\_off}}$) in UAS and the rotational radiation test.

**Figure 8.** Sounding test of dual thermistor radiosondes. (a) Raw temperature difference between the white and black sensors ($T_{\text{B\_raw}} - T_{\text{W\_raw}}$), (b) effective irradiance based on ($T_{\text{B\_raw}} - T_{\text{W\_raw}}$) calculated by Eq. (10) and (c) radiation correction value of the white sensor at daytime calculated by Eq. (19). (d) Raw temperature difference between the white and black sensors ($T_{\text{B\_raw}} - T_{\text{W\_raw}}$), (e) calculated effective irradiance ($T_{\text{B\_raw}} - T_{\text{W\_raw}}$) and (f) radiation correction value of the white sensor at nighttime.

**Figure 9.** Uncertainty analysis on the DTR and intercomparison with Vaisala RS41. (a) Daytime and (b) nighttime effective irradiance measured by DTR with uncertainty ($k = 2$). Uncertainty factors contributing to the uncertainty of the corrected temperature $U(T_{\text{W\_cor}})$ of DTR at (c) daytime and (d) nighttime. Temperature difference between DTR and RS41 with DTR uncertainty ($k = 2$) at (e) daytime and (f) nighttime.

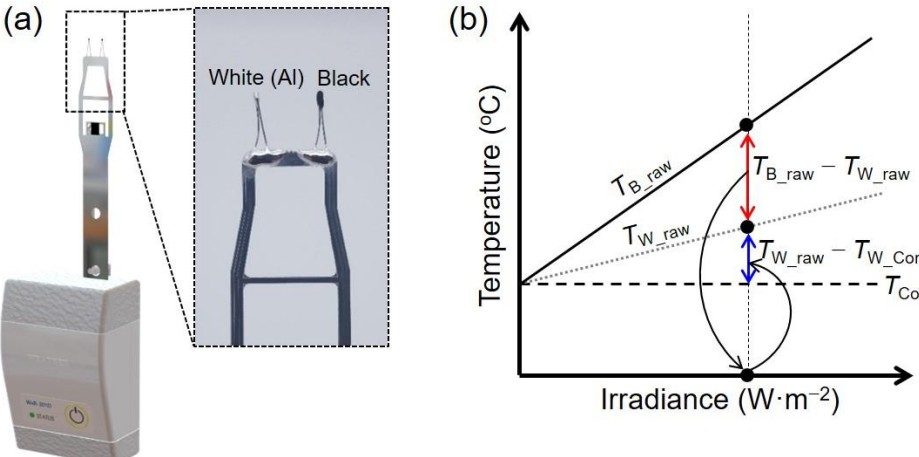

**Figure 1.** (a) Dual thermistor radiosonde (DTR) with a white and black sensor and (b) operation principle of DTR for irradiance measurement and correction of radiation effect based on the measured irradiance. The temperature difference between the dual thermistors ($T_{B\_raw} - T_{W\_raw}$) is linearly proportional to the irradiance, and the radiation-induced heating of the white sensor ($T_{W\_raw} - T_{W\_cor}$) is corrected based on the irradiance measured by ($T_{B\_raw} - T_{W\_raw}$).

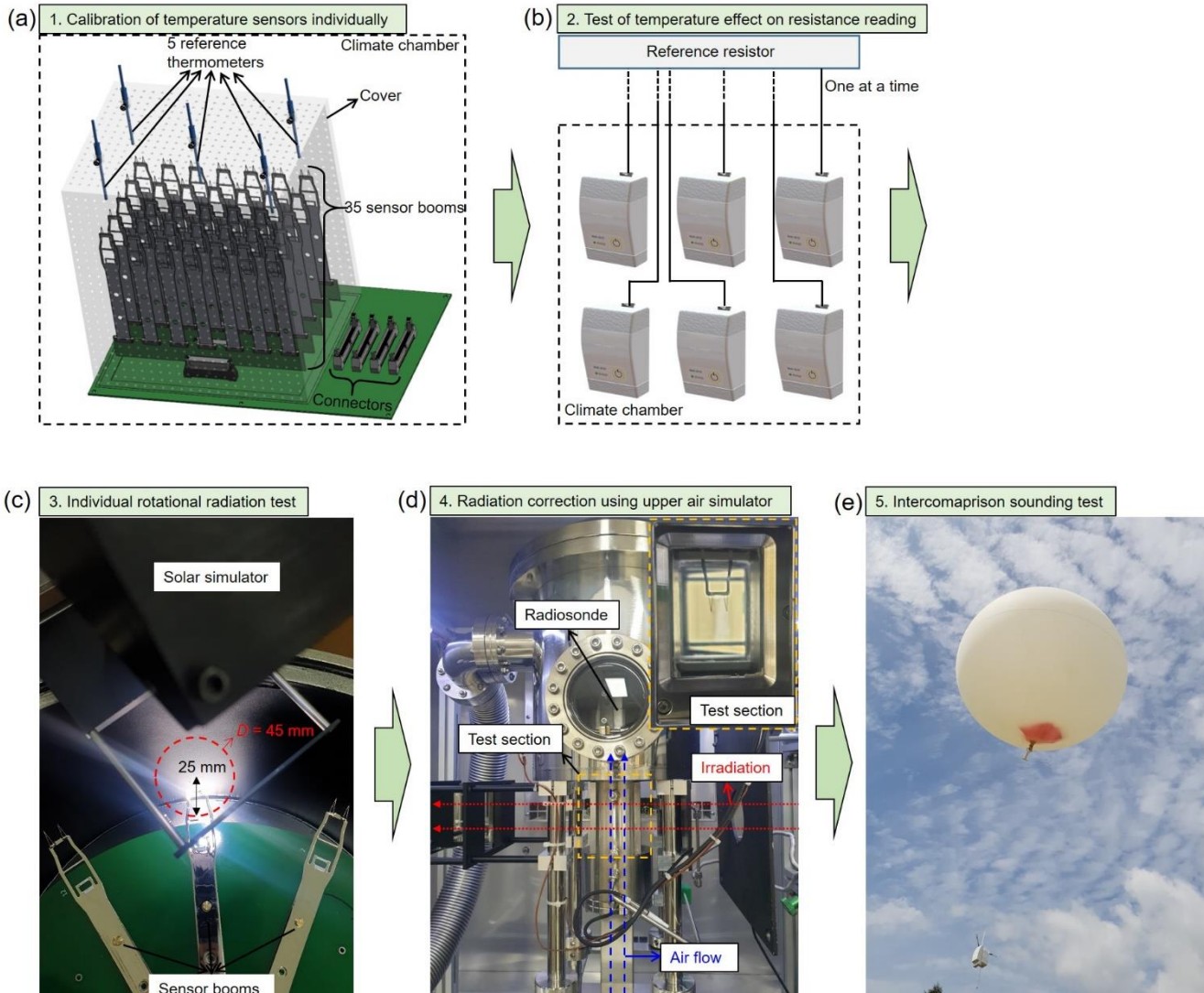

**Figure 2.** Characterisations of DTR. (a) Individual calibration of thermistors in a climate chamber, (b) test of the effect of temperature on the resistance reading using the radiosonde boards in the climate chamber, (c) radiation test on individual thermistors, (d) parameterisation of radiation measurement and correction formulae using an upper air simulator and (e) sounding test by applying laboratory characterisation results.

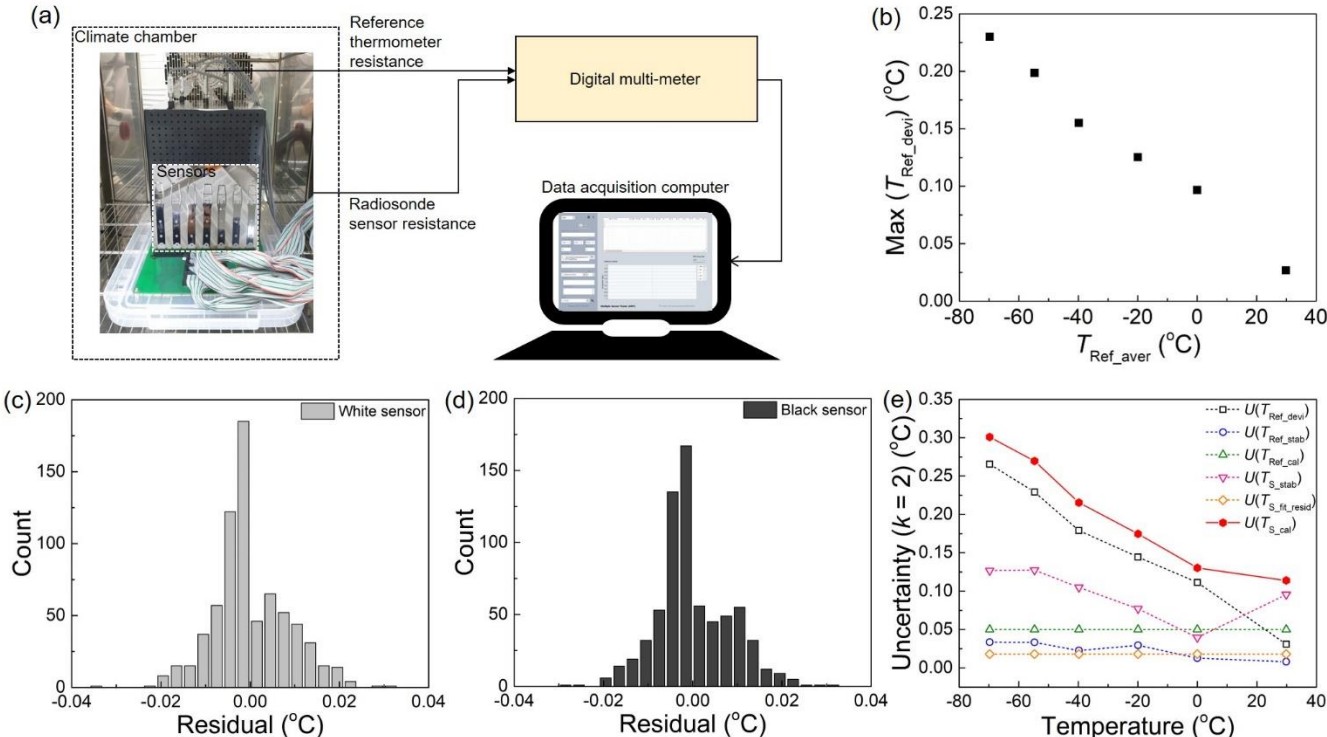

**Figure 3**. Calibration of individual thermistors in a climate chamber. (a) Calibration setup showing thermistors on booms (left), a digital multimeter to read the sensor resistance (top-right) and a data acquisition computer (bottom-right). (b) Maximum temperature deviations ($T_{\text{Ref\_devi}}$) with respect to the average of five reference thermometers ($T_{\text{Ref\_aver}}$) as a function of $T_{\text{Ref\_aver}}$. Distribution of the residuals of the (c) white and (d) black sensors by individually applying the calibration curves. (e) The uncertainty budget on the radiosonde thermistor calibration, $U(T_{\text{S\_cal}})$, with a coverage factor $k = 2$. Uncertainty factors including reference temperature deviations $U(T_{\text{Ref\_devi}})$, stability $U(T_{\text{Ref\_stab}})$ and calibration $U(T_{\text{Ref\_cal}})$, radiosonde sensor stability $U(T_{\text{S\_stab}})$ and fitting residual $U(T_{\text{S\_fit\_resid}})$ are considered for $U(T_{\text{S\_cal}})$.

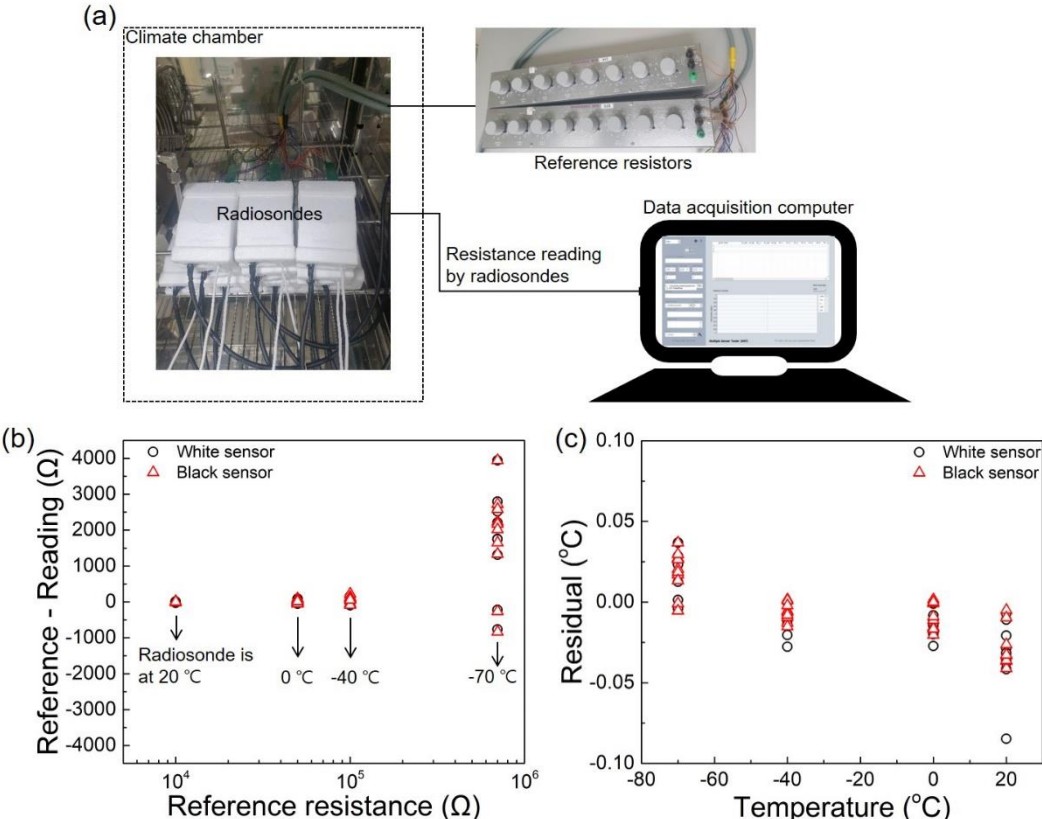

**Figure 4**. Test of the temperature effect on resistance reading by radiosonde boards. (a) Test setup showing the radiosonde boards in a climate chamber (left), reference resistors (top-right) and a data acquisition computer (bottom-right). (b) Difference between the reference resistance and radiosonde reading as a function of the reference resistance. (c) Residual after conversion of resistance to temperature as a function of temperature.

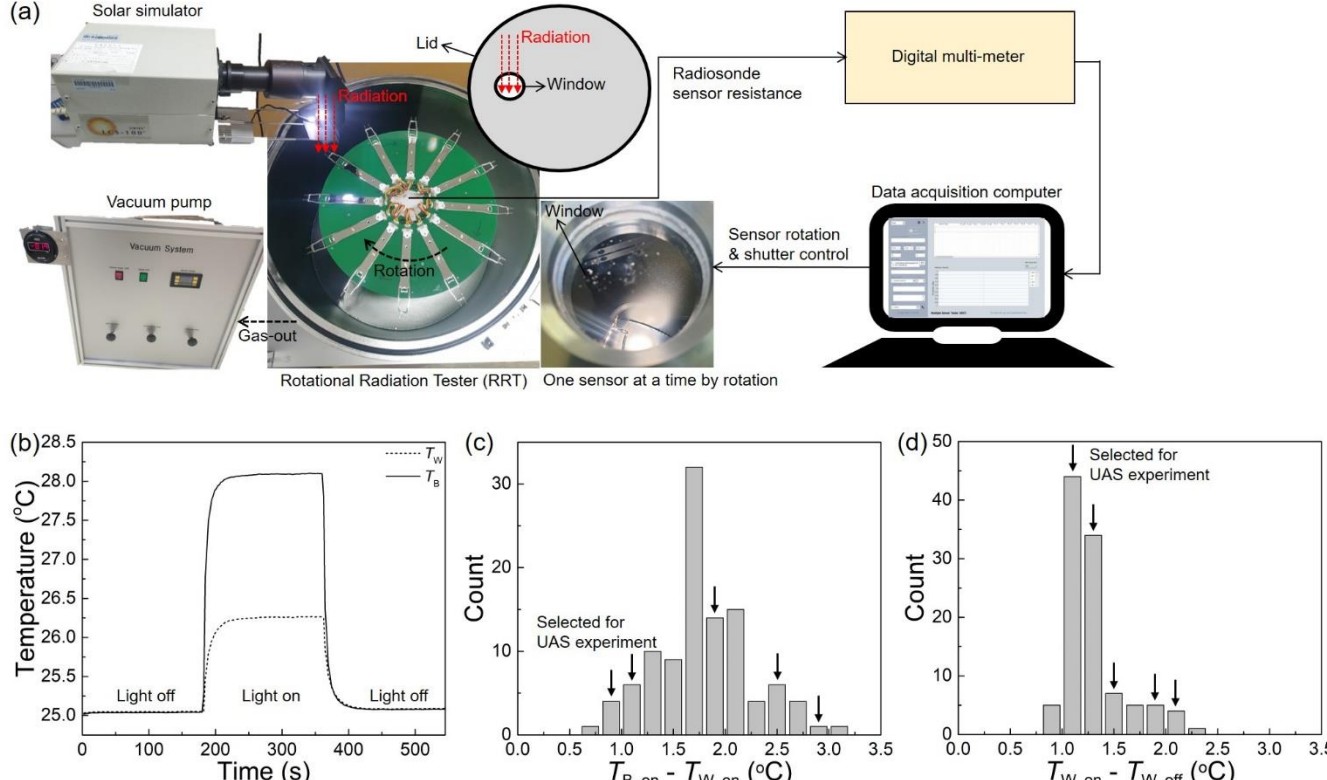

**Figure 5.** Rotational radiation test (RRT) on radiosonde thermistors individually. (a) RRT setup showing the radiosonde thermistors in a chamber, solar simulator, and vacuum pump (left), a digital multimeter (top-right) and a data acquisition computer (bottom-right). (b) Temperature measured by a white ($T_W$) and black ($T_B$) sensor with/without light irradiation by the solar simulator. (c) Distribution of the temperature difference between the paired white and black sensors. (d) Distribution of the temperature increase of white sensors by the irradiation. Five pairs of a white and black sensor were selected for radiation correction experiments using an upper air simulator (UAS), as indicated by black arrows in c and d.

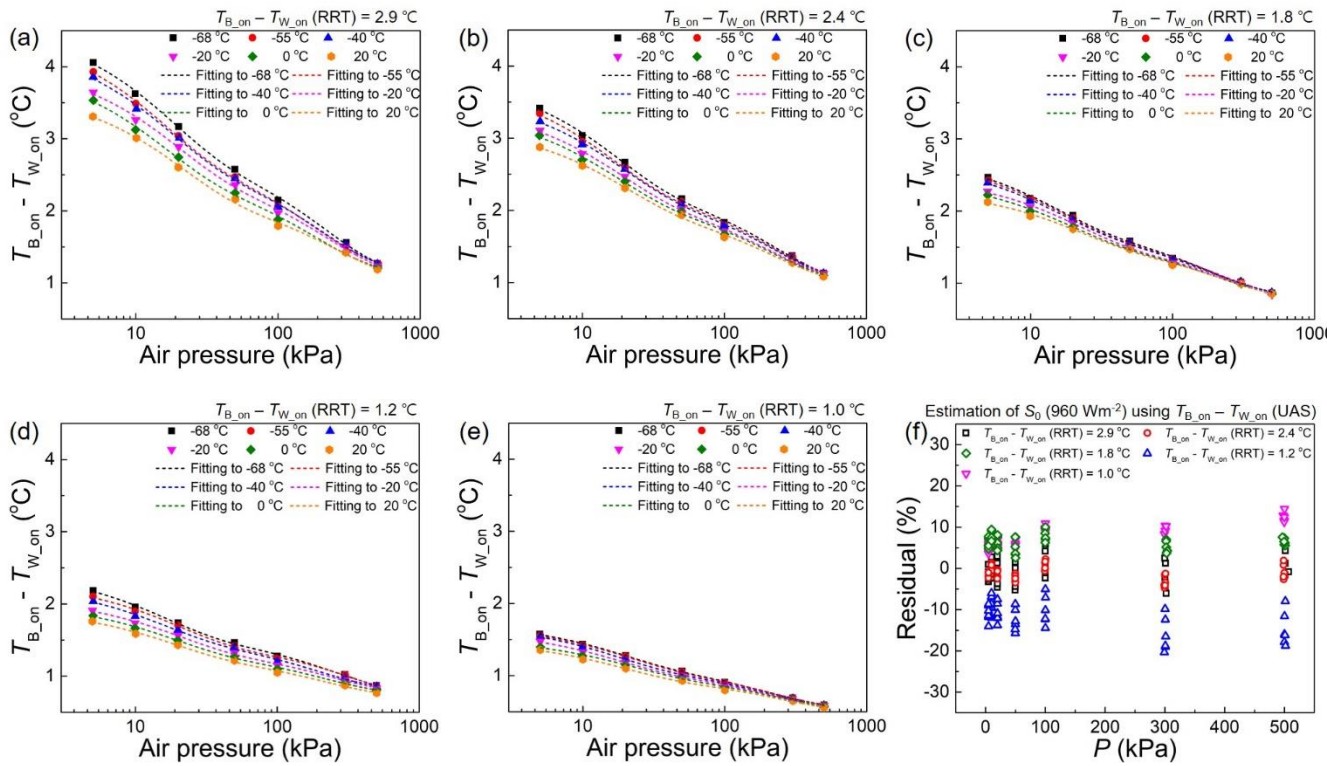

**Figure 6.** Temperature difference between paired white and black sensors ($T_{B\_on} - T_{W\_on}$) investigated using UAS. (a–e) ($T_{B\_on} - T_{W\_on}$) of the five paired radiosonde thermistors as a function of air pressure with varying temperature. (f) Residual of irradiance calculated on the basis of ($T_{B\_on} - T_{W\_on}$) obtained in UAS and the rotational radiation test.

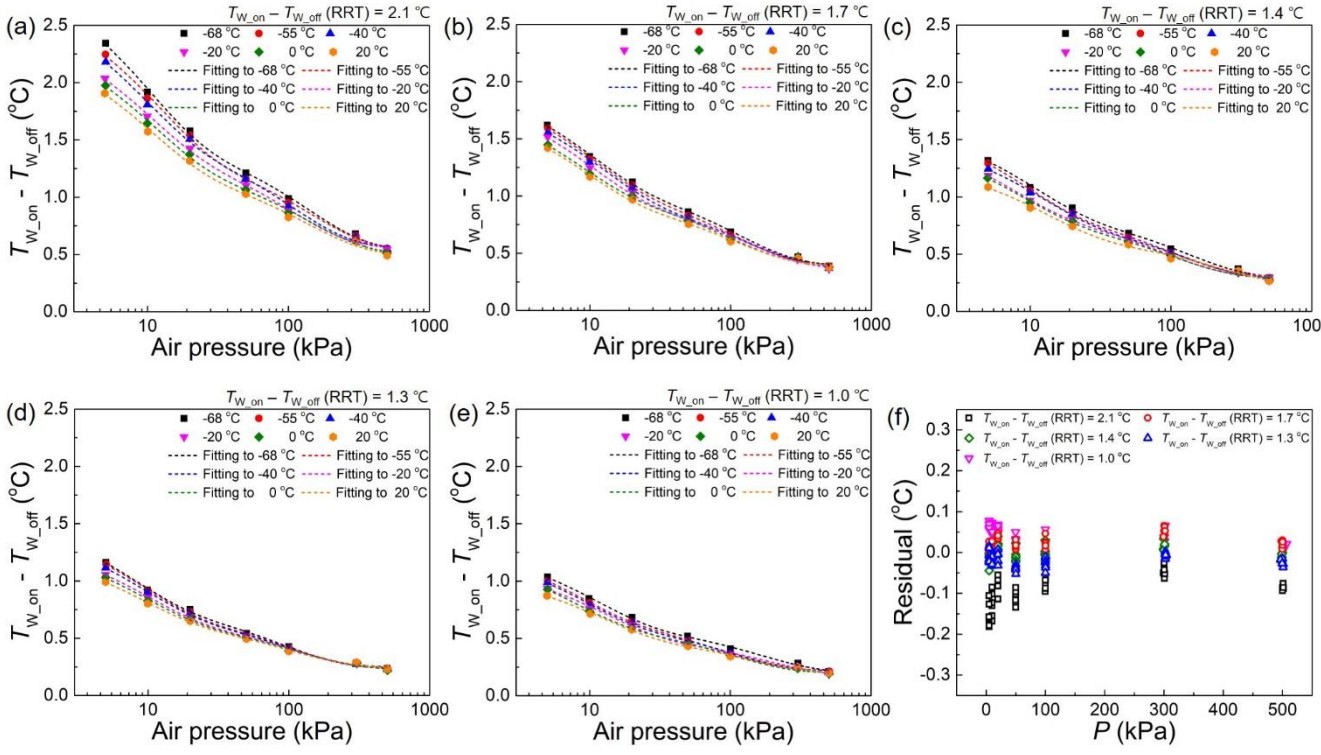

**Figure 7**. Radiation correction value of white sensors ($T_{W\_on} - T_{W\_off}$) investigated using UAS. (a–e) ($T_{W\_on} - T_{W\_off}$) of the five radiosonde white sensors as a function of air pressure with varying temperature. (f) Residual of correction value calculated on the basis of ($T_{W\_on} - T_{W\_off}$) in UAS and the rotational radiation test.

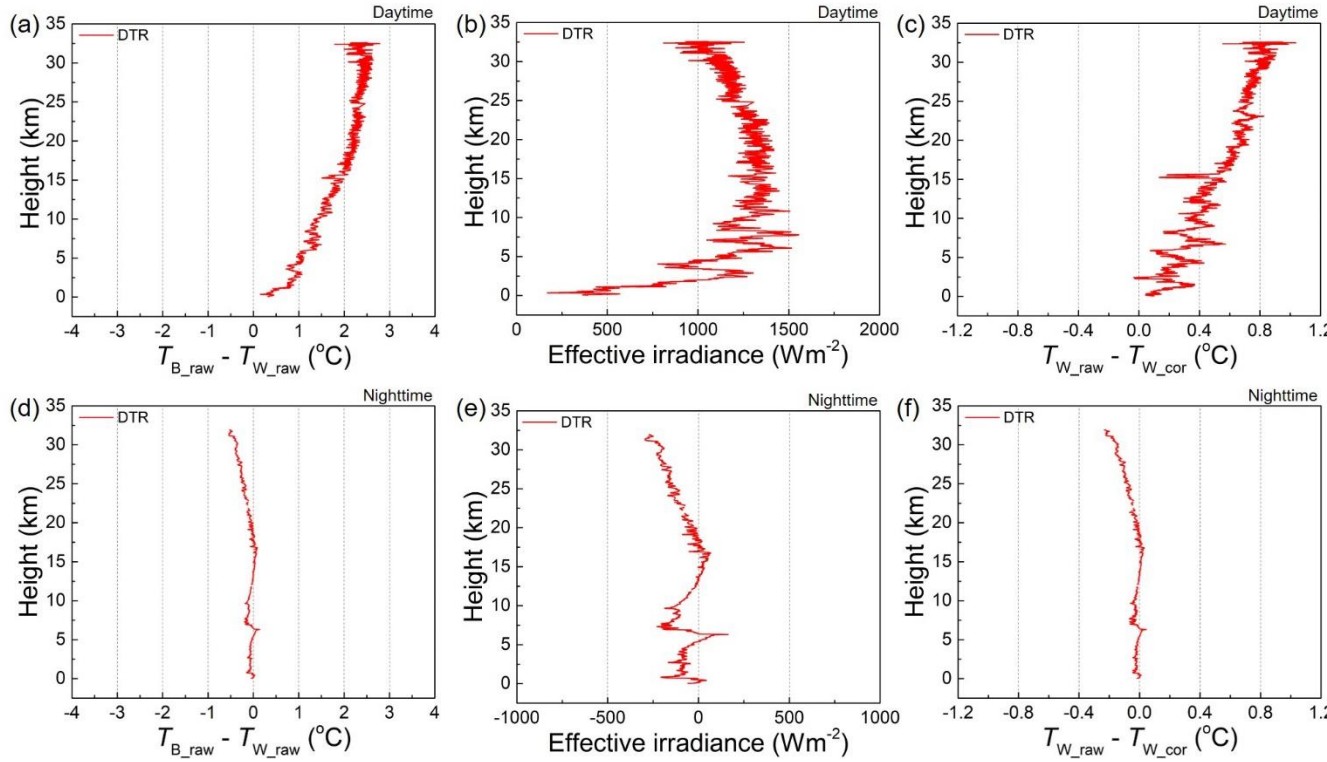

**Figure 8**. Sounding test of dual thermistor radiosondes. (a) Raw temperature difference between the white and black sensors ($T_{\text{B\_raw}} - T_{\text{W\_raw}}$), (b) effective irradiance based on ($T_{\text{B\_raw}} - T_{\text{W\_raw}}$) calculated by Eq. (10) and (c) radiation correction value of the white sensor at daytime calculated by Eq. (19). (d) Raw temperature difference between the white and black sensors ($T_{\text{B\_raw}} - T_{\text{W\_raw}}$), (e) calculated effective irradiance ($T_{\text{B\_raw}} - T_{\text{W\_raw}}$) and (f) radiation correction value of the white sensor at

620 nighttime.

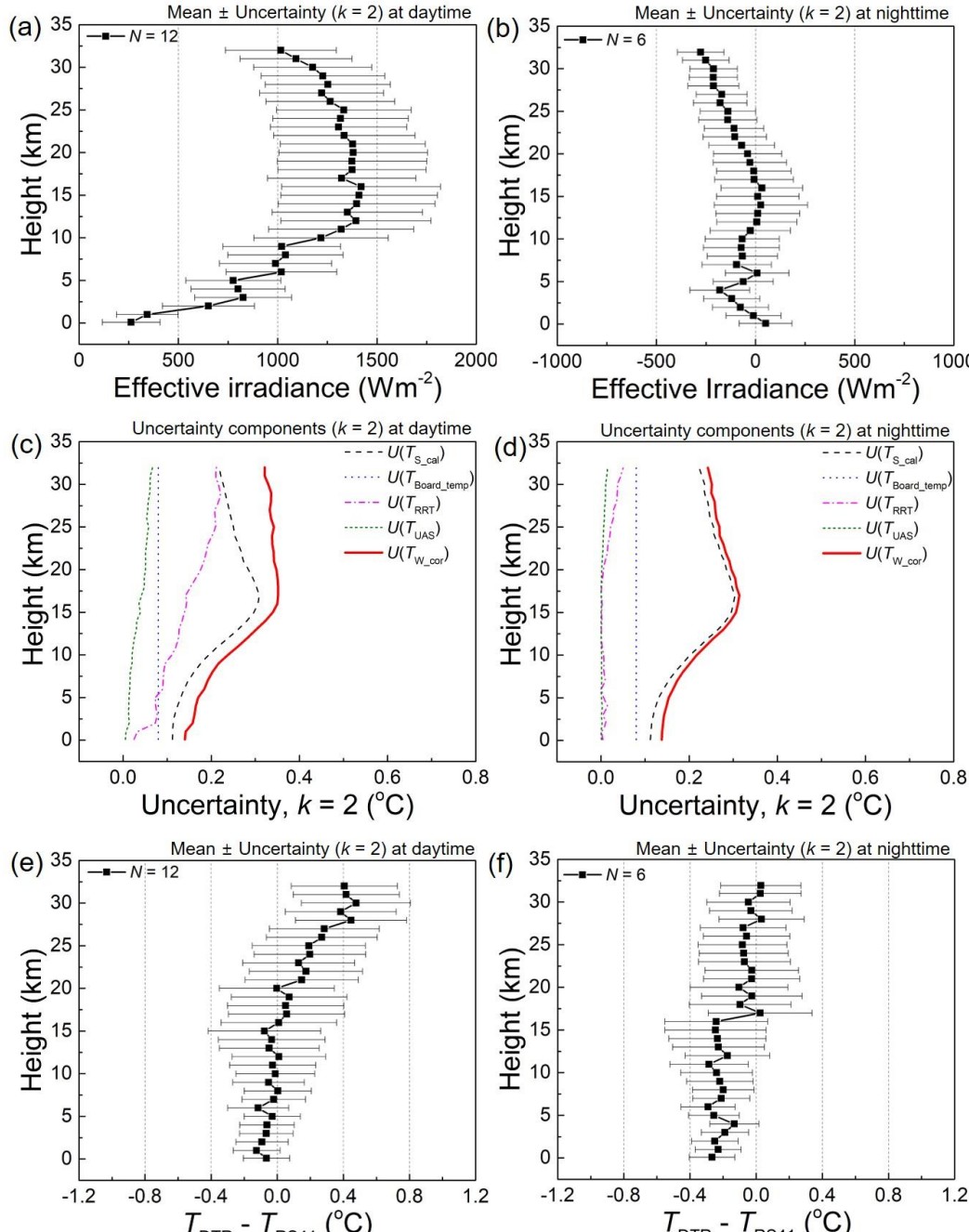

**Figure 9.** Uncertainty analysis on the DTR and intercomparison with Vaisala RS41. (a) Daytime and (b) nighttime effective irradiance measured by DTR with uncertainty ($k = 2$). Uncertainty factors contributing to the uncertainty of the corrected temperature $U(T_{W\_cor})$ of DTR at (c) daytime and (d) nighttime. Temperature difference between DTR and RS41 with DTR uncertainty ($k = 2$) at (e) daytime and (f) nighttime.