# Peer review of "Laboratory characterisations and intercomparison sounding test of dual thermistor radiosondes for radiation correction"

_Atmospheric Measurement Techniques, 2021_

## Referee Comment (RC1)

Review of manuscript "Laboratory characterisations and intercomparison sounding test of dual thermistor radiosondes for radiation correction" by Sang-Wook Lee, et al., AMT-2021-343

**General**

The manuscript presents a method to estimate and correct the solar radiation error of radiosonde temperature measurements, using a dual thermistor radiosonde. Accurate in situ measurements of temperature profiles by radiosondes constitute a highly relevant topic, for example for climate monitoring, and reference-quality temperature measurements by radiosondes are of high demand.

The method presented in this paper relies on the temperature difference between a black-coated and an aluminum-coated thermistor to derive the effective radiation field which is used in the temperature correction of the Al-coated temperature sensor. The advantage of this method is that it does not rely on modeled assumptions on the radiation field or sensor properties, but aims to measure it directly. With this approach, the authors continue earlier pioneering work by e.g. Schmidlin 1986.

The novel aspect is that the approach relies on purely experimental sensor characterisation in terms of sensitivity to radiation. This is different from previous studies using dual or multiple sensor techniques which are based on solving multiple heat balance equations and therefore require a number of assumptions or estimates with regard to sensor dimensions, material properties and other parameters. Although there are not many easily accessible publications on multi-sensor radiosondes (Schmidlin, Luers, and references herein), the authors should refer to these in their study.

The manuscript is clearly structured. However, it reads as a straightforward and rather technical description with a strong emphasis on the metrological aspects, in particular uncertainties. Although this should of course make up a significant part, more motivation, explanation or interpretation would be appropriate or even required in several places with regard to the methods and results (see detailed comments) in the light of the physical processes taking place. This would not only 'loosen up' the text but may help understanding the effects and improve the potential impact in the radiosonde data user community.

**Detailed comments**

Abstract:

L9:     The white sensor is in fact coated with Aluminum, and should be referred to as such here. The classification "white" can be used later in the manuscript.

The abstract should mention the ratio of the heating rates of the white and the black sensors (which is 1:3), see e.g. Schmidlin (1986)

L12-15:      Think of a different phrasing: more motivating instead of just list a number of facts.

Introduction:

L32:    It may be referred to the co-location issue and flight trajectories of balloon soundings.

L57:    Remove "…as previously reported (Lee et al.)"

L72:    "freezing" -> "climate"

Section 2.2:

The authors should reference to the work of Francis Schmidlin (NASA Tech. Paper 2637, 1986) on the multiple thermistor radiosonde.

Section 3.1:

L114-115: The exact procedure for the calibration and characterisation measurements is unclear at this point. Add a sentence that these will be discussed in detail in the following sections.

L115: "via" -> "due to"

L116: "…to include the differences in the sensitivities of the individual thermistors in the radiation correction."

Fig. 2 (c): Which parts of the sensor boom beyond the thermistors are irradiated?

Section 3.2:

Obviously, spatial temperature inhomogeneities within the calibration "box" dominate the calibration uncertainty (Fig. 3 (b)), which to a wide extent dominates the overall uncertainty of the corrected temperature (Fig. 9 (c) and (d)). Could this be reduced, e.g. through suitable ventilation?

L127: "…by the five…"

L129: "gradient" -> "differences" or "deviations"

L131: Between "polynomial equation" and "yields" you may insert ", i.e. the inclusion of a quadratic term, which is not present in the Steinhart-Hart equation, "

L132: "…, the Steinhart-Hart equation is modified…"

Section 3.3:

L144-146: Make more clear that the effect of the temperature of the radiosonde electronics board on the thermistor resistance (or temperature) measurement is investigated here.

Fig. 4(b): Use symbol for unit; don't use "k" and "M" for x-axis labels of resistance; Caption: "… (c) Residual after conversion of resistance to temperature as function of temperature.

L155: What is meant by "roughly distributed"?

Section 3.4:

The first paragraph should be worded more clearly and more precisely. E.g., L158: "… the unit difference in terms of the correction value.": Does that mean something like "sensitivity to irradiation and therefore the amount of radiation correction may vary for individual radiosondes, presumably related to the production process of the thermistors..."?;

L161: "Irregularities in the construction of the leads connecting sensor and boom…" It is interesting that these variations in the properties of the sensors, such as e.g. its diameter, have such a big influence. The authors should discuss this in more detail. A helpful reference may be de Podesta et al. (2018) (DOI: 10.1088/1681-7575/aaaa52).

Second paragraph:

What irradiance is applied? What are the conditions with regard to air pressure and ventilation in the RRT? Is the pump used to vary the pressure, or to create an airflow, or both? If there is no significant ventilation I would expect a certain variation of the results from that, because the cooling efficiency should strongly vary with air flow at low or vanishing flow rates. That might at least partially explain the distributions in Figs. 5 (c) and (d). If the ventilation is controlled, is it adjusted similar to what is used in the UAS ($\sim$5 m$\cdot$s$^{-1}$)?

Fig. 5 (b):     Rad. warming of Al-coated is more than one third of that for the black thermistor in Fig. 5 (b), and the absolute value of $\sim$1.2 K in the example seems unexpectedly large at a first glance (is that a typical example?). Does that mean that the reflectivity of the Al-coating is not that close to one, but say $\sim$0.7 or so?

Quantitative information on irradiance (is it the 960 W$\cdot$m$^{-2}$?), pressure and ventilation for the RRT tests would be helpful to better assess or classify the results.

Is there a test to see if the two thermistors influence each other (e.g. via heat conduction)? This could be assessed by selective irradiation.

How are the T-differences extracted/evaluated from the data in Fig. 5 (b)?

Section 4.1:

First paragraph: (How) Is the angle of the sensor boom, i.e. the irradiation angle and boom orientation, and the angle relative to the air flow taken into account in the UAS measurements?

L190:  Better: "… with the fitting coefficients being functions of T_W_on…"

L193/194:     Isn't the point here that the effective long-wave cooling is *different* for the two thermistors, according to the different emissivities, whereas the SW-absorption does not depend on T?

Eq. (2):        What motivated the exponential functions as fitting model?

L216: The radiation flux of 960 W$\cdot$m$^{-2}$ is known. Is the (re)fitting done for the purpose of estimating uncertainties in terms of the residuals?

Eq. (10):       Is the equation valid only for 7-100 hPa and 4-6.5 m$\cdot$s$^{-1}$?

Fig. 6 (also Fig. 7):   The trend of the data points is difficult to see at low p; consider using a logarithmic scale.

Section 4.2:

Eq. (19):       Validity range with regard to p and v?

L234f: Please discuss the influence of the temperature dependence of the efficiency of convective cooling, i.e. at low temperatures the thermal conductivity of air decreases leading to an increase of the radiative heating of the temperature sensor.

Section 5:

At what time was the daytime sounding performed? What was the cloud situation?

Add more discussion on what is observed in the plots in Fig. 8. The reconstructed solar (ir)radiance decreases with altitude in the stratosphere. This is opposite to what is expected. Please compare the reconstructed radiation profile to RTM calculations, and discuss the differences. See for example Philipona et al 2020 (doi: 10.1127/metz/2020/1044) for in situ measurements of the radiation profile.

In particular, it could be discussed more how the different long-wave backgrounds in the experiments and in the situations during the real soundings may influence the results, or what potential systematic errors of the radiation correction may be in connection to this.

Can it be assumed without reservation that the sensitivity of the thermistors is the same with regard to long-wave and short-wave radiation (wavelength-independent emissivity/absorptivity)?

Mention in the caption of Fig. 8 that the effective irradiance is calculated using Eq. (9) (or 10).

Section 6.2

L340/341: "enhance" -> "improve" or "reduce"; It should be discussed here (or in 3.2) whether and how the calibration uncertainty, which is obviously due to air temperature inhomogeneities within the calibration volume, can be reduced.

Section 6.3

Fig. 9 (f): The >0.2 K offset at ~16 km is striking, please comment on this.

---

## Referee Comment (RC2)

[referee-annotated manuscript omitted]

---

## Referee Comment (RC3)

[referee-annotated manuscript omitted]

---

## Referee Comment (RC4)

Review of manuscript "Laboratory characterisations and intercomparison sounding test of dual thermistor radiosondes for radiation correction" by Sang-Wook Lee, et al., AMT-2021-343

Synopsis: The authors describe a detailed metrological characterization of a dual thermistor radiosonde and how the difference between thermistors can be used for the radiation error affecting both. The result is a quite simple multilinear model for correction, based just on the differences between thermistors. The characterization methods appear quite solid. Only the comparison with the de facto standard (RS41) has been performed using too small samples.

Major comments:

- I was a bit surprised to learn that the temperature was varied only between -70 and -20 deg, since in the tropics and over Antarctica temperatures below -90 degrees are not uncommon. This seems important since in Fig. 4, there are substantial variations of resistance reading at -70 deg. Does the instrument fail at even lower temperatures? Perhaps this is a wrong impression because of the y axis being linear, not logarithmic.
- The comparison with the current de fact standard RS41 should be more comprehensive. In Fig. 9 it is not clear how many radiosondes were launched in parallel. This is very important to have a robust estimate of differences. Somewhere in the text it is written N=12 for daytime and N=6 for nighttime. That should be also in the caption. The same applies to the sounding test described in section 8. It appears it was only one ascent?
- There is a lot of redundancy in the formulae. Why do you specify $S_0 = 960$ W/m^2 all the time in formulae 2-19. It is given in the text and does never change. The same is true for $v_0$. Personally I would also recommend writing fractions as with — as divisor, not / in numbered formulae.
- Formula (25) holds only if errors are independent

Minor comments:

L69: different emissivities

Fig. 6, 7: A logarithmic y axis would be very helpful, and is also more suitable to the parameterization you give in Formula (2), which consists of exponential functions.

---

## Author Comment (AC1)

**Referee #1**

**General Comment:** The manuscript presents a method to estimate and correct the solar radiation error of radiosonde temperature measurements, using a dual thermistor radiosonde. Accurate in situ measurements of temperature profiles by radiosondes constitute a highly relevant topic, for example for climate monitoring, and reference-quality temperature measurements by radiosondes are of high demand.

The method presented in this paper relies on the temperature difference between a black-coated and an aluminum-coated thermistor to derive the effective radiation field which is used in the temperature correction of the Al-coated temperature sensor. The advantage of this method is that it does not rely on modeled assumptions on the radiation field or sensor properties, but aims to measure it directly. With this approach, the authors continue earlier pioneering work by e.g. Schmidlin 1986.

The novel aspect is that the approach relies on purely experimental sensor characterisation in terms of sensitivity to radiation. This is different from previous studies using dual or multiple sensor techniques which are based on solving multiple heat balance equations and therefore require a number of assumptions or estimates with regard to sensor dimensions, material properties and other parameters. Although there are not many easily accessible publications on multi-sensor radiosondes (Schmidlin, Luers, and references herein), the authors should refer to these in their study.

The manuscript is clearly structured. However, it reads as a straightforward and rather technical description with a strong emphasis on the metrological aspects, in particular uncertainties. Although this should of course make up a significant part, more motivation, explanation or interpretation would be appropriate or even required in several places with regard to the methods and results (see detailed comments) in the light of the physical processes taking place. This would not only 'loosen up' the text but may help understanding the effects and improve the potential impact in the radiosonde data user community.

→ We thank the Reviewer for valuable comments to improve the quality of the manuscript. As the Reviewer mentioned, the manuscript was oriented to metrological aspects rather than meteorological and physical understanding. We have reinforced the latter part by addressing detailed comments as below.

**Detailed comments**

**Abstract:**

**L9:** The white sensor is in fact coated with Aluminum, and should be referred to as such here. The classification "white" can be used later in the manuscript.

→ The word "white sensor" is replaced by "aluminium-coated sensor" in the Abstract because it is not classified as "white sensor" yet.

**Before:** white

**After (Line 9, 17, 21, & 22):** aluminium-coated

The abstract should mention the ratio of the heating rates of the white and the black sensors (which is 1:3), see e.g. Schmidlin (1986)

→ The ratio of the degree of heating between white and black sensors is presented (1:2.4).

**Added statement (Line 16-17):** to investigate the degree of heating of aluminium-coated and black sensors (the average ratio = 1:2.4)

**L12-15:** Think of a different phrasing: more motivating instead of just list a number of facts.

→ Motivations of each characterization procedure are added.

**Before:** individually calibrate the temperature of the thermistors in a climate chamber; test the effect of temperature on the resistance reading using radiosonde boards in the climate chamber; individually perform radiation tests on thermistors; and perform parameterisation of the radiation measurement and correction formulas using an upper air simulator with varying temperature, pressure and ventilation speed.

**After (Line 13-20):** individually calibrate the temperature of the thermistors in a climate chamber from −70 to 30 °C to evaluate the uncertainty of raw temperature measurement before radiation correction; test the effect of temperature on the resistance reading using radiosonde boards in the climate chamber from −70 to 20 °C to identify a potential source of errors owing to the boards, especially at cold temperatures; individually perform radiation tests on thermistors at room temperature to investigate the degree of heating of aluminium-coated and black sensors (the average ratio = 1:2.4) and use the result for obtaining unit-specific radiation correction formulas; and perform parameterisation of the radiation measurement and correction formulas with five representative pairs of sensors in terms of temperature, pressure, ventilation speed, and irradiance using an upper air simulator.

**Introduction:**

**L32:** It may be referred to the co-location issue and flight trajectories of balloon soundings.

→ The co-location issue is mentioned.

**Before:** Thus, radiosonde measurements are often used as reference to correct other measurement data.

**After (Line 34-36):** Radiosonde observations are often co-located with global navigation satellite system radio occultation and used as reference for validating their one-dimensional interpolation which follows the flight trajectories of balloon soundings.

Before: Thus, radiosonde measurements are often used as reference to correct other measurement data

**L57:** Remove "…as previously reported (Lee et al.)"

→ The phrase is removed

**Removed phrase:** as previously reported (Lee et al., 2018a).

**L72:** "freezing" -> "climate"

→ The word "freezing" is replaced by "climate"

**Before:** freezing

**After (Line 78):** climate

**Section 2.2:**

The authors should reference to the work of Francis Schmidlin (NASA Tech. Paper 2637, 1986) on the multiple thermistor radiosonde.

→ The work of Schmidlin *et al.* is referred as a pioneering research of radiation correction.

**Added statement (Line 103-105):** Previously, a pioneer work using multiple thermistors with different spectral responses (emissivity and absorptivity) was conducted for the radiation correction. In the work, however, complete knowledge on material properties of air and sensors and sensor geometry is required to solve multiple heat balance equations (Schmidlin et al., 1986).

**Before:** DTR utilises the temperature difference

**After (Line 105-106):** DTR utilises the purely experimental temperature difference

**Section 3.1:**

**L114-115:** The exact procedure for the calibration and characterisation measurements is unclear at this point. Add a sentence that these will be discussed in detail in the following sections.

→ Brief explanations are added for each procedure as in the case of Abstract. A sentence that these will be discussed in detail in the following sections is added as suggested.

**Before:** First, the thermistors on the sensor boom are individually calibrated using a climate chamber (Fig. 2(a)). Then, the temperature effect on the resistance reading by radiosonde boards is tested in the climate chamber (Fig. 2(b)). The temperature increase of all thermistors due to irradiation is individually recorded at room temperature (Fig. 2(c)) to compensate for the difference among units for radiation correction. The radiation measurement and correction formulas of DTR are obtained using the UAS with varying temperature, pressure, ventilation speed and irradiance (Fig. 2(d)). The laboratory experimental results are combined and applied to the DTR sounding system. Then, the sounding results of DTR are compared with those of a commercial radiosonde through dual soundings (Fig. 2(e)).

**After (Line 121-130):** First, the thermistors on the sensor boom are individually calibrated using a climate chamber from −70 to 30 °C to evaluate the uncertainty of raw temperature measurement before radiation correction (Fig. 2(a)). Then, the temperature effect on the resistance reading by radiosonde boards is tested in the climate chamber from −70 to 20 °C to identify a potential source of errors owing to the boards, especially at cold temperatures (Fig. 2(b)). The temperature increase of all thermistors due to irradiation is individually recorded at room temperature (Fig. 2(c)) to include the differences in the sensitivities of the individual thermistors in the radiation correction. The radiation measurement and correction formulas of DTR are obtained in terms of temperature, pressure, ventilation speed and irradiance using the UAS (Fig. 2(d)). The laboratory experimental results are combined and applied to the DTR sounding system. Then, the sounding results of DTR are compared with those of a commercial radiosonde through dual soundings (Fig. 2(e)). Each characterization procedure will be discussed in detail in the following sections.

**L115:** "via" -> "due to"

→ It is changed.

**Before:** via

**After (Line 125):** due to

**L116:** "…to include the differences in the sensitivities of the individual thermistors in the radiation correction."

→ The suggested statement is used.

**Before:** to compensate for the difference among units for radiation correction.

**After (Line 125-126):** to include the differences in the sensitivities of the individual thermistors in the radiation correction.

**Fig. 2 (c):** Which parts of the sensor boom beyond the thermistors are irradiated?

→ The diameter ($D$) of the beam spot on the sensor is 45 mm and the distance between the sensor bead and the beam boundary is 25 mm. These are marked in Figure 2(c).

**Modified Figure (Figure 2(c)):** The diameter of the beam spot and the distance between the sensor bead and the beam boundary are marked.

**Added statement (Line 197-199):** The diameter ($D$) of the beam spot on the sensor is 45 mm and the distance between the sensor bead and the beam boundary is 25 mm.

**Section 3.2:**

Obviously, spatial temperature inhomogeneities within the calibration "box" dominate the calibration uncertainty (Fig. 3 (b)), which to a wide extent dominates the overall uncertainty of the corrected temperature (Fig. 9 (c) and (d)). Could this be reduced, e.g. through suitable ventilation?

→ The deviations are due to the temperature difference between the front door side and the rear fan side of the chamber. If the effect of ventilation is increased, the uncertainty due to the stability of sensor temperature is increased. One of the practical ways to reduce the calibration uncertainty is to conduct another round of calibration with the thermistor set (35 pairs) rotated 180° in the chamber to average out the effect of temperature deviations. Another way is to find a different location with smaller temperature deviations inside the chamber. Recently, we have found that the maximum temperature deviations are reduced to 0.1 °C when the thermistor set is moved below the horizontal level of the fan.

**Added statement (Line 157-162):** The uncertainty due to spatial temperature deviations $U(T_{Ref\_devi})$ in the chamber dominates the calibration uncertainty. The deviations are due to the temperature difference between the front door side and the rear fan side of the chamber. One of the practical ways to reduce the calibration uncertainty is to conduct another round of calibration with the thermistor set (35 pairs) rotated 180° in the chamber and average out the effect of temperature deviations. Another way is to find other locations with smaller temperature deviations. The temperature deviations can be affected by the thermal insulation of the door and the aisles for data cables as well as the ventilation by the fan in the chamber.

**L127:** "…by the five…"

→ "The" is added.

**Before:** by five reference PRTs

**After (Line 138):** by the five reference PRTs

**L129:** "gradient" -> "differences" or "deviations"

→ The word "gradient" is changed to "deviations".

**Before:** gradient

**After (throughout manuscript):** deviations

**Before:** $T_{Ref\_grad}$

**After (throughout manuscript):** $T_{Ref\_devi}$

**Modified Figure (Figure 3 and Caption):** $T_{Ref\_grad}$ is changed to $T_{Ref\_devi}$

**L131:** Between "polynomial equation" and "yields" you may insert ", i.e. the inclusion of a quadratic term, which is not present in the Steinhart-Hart equation, "

→ The statement is added as suggested.

**Added statement (Line 144-145):** i.e. the inclusion of a quadratic term, which is not present in the Steinhart-Hart equation

**L132:** "…, the Steinhart-Hart equation is modified…"

→ The sentence is changed as suggested.

**Before:** Therefore, the former equation is adopted

**After (Line 145-146):** Therefore, the Steinhart-Hart equation is modified

**Section 3.3:**

**L144-146:** Make more clear that the effect of the temperature of the radiosonde electronics board on the thermistor resistance (or temperature) measurement is investigated here.

→ The purpose of the experiment in Section 3.3 is explained more clearly.

**Before:** To measure the temperature using the thermistors via Eq. (1), the measurement of the sensor resistance by the radiosonde boards should be evaluated in the temperature range of the sensor calibration.

**After (Line 164-166):** To properly measure the temperature using the thermistors via Eq. (1), the effect of the temperature of the radiosonde electronics board on the thermistor resistance measurement should be investigated in the same temperature range of the thermistor calibration.

**Fig. 4(b):** Use symbol for unit; don't use "k" and "M" for x-axis labels of resistance; Caption: "… (c) Residual after conversion of resistance to temperature as function of temperature.

→ The resistance unit, $\Omega$ is used in Fig. 4(b); The x-axis labels are changed to $10^3$, $10^4$, and $10^5$; Caption of Fig. 4(c) is changed as suggested.

**Modified Figure (Figure 4 and Caption):** The unit of resistance, x-axis labels, and the caption of (c) is changed.

**L155:** What is meant by "roughly distributed"?

→ The statement is deleted.

**Before:** Since the temperature error is roughly distributed within ±0.05 °C, the standard deviation of all data points is obtained (0.04 °C) and is used for uncertainty ($k = 1$) due to the influence of temperature on the resistance reading by radiosonde boards.

**After (Line 175-178):** Assuming that the probability distribution is a normal distribution function, the standard deviation (SD) of all data points (0.04 °C) is used for standard uncertainty due to the influence of the temperature of radiosonde electronics boards on the resistance (or temperature) measurement.

**Section 3.4:**

The first paragraph should be worded more clearly and more precisely. E.g., L158: "… the unit difference in terms of the correction value.": Does that mean something like "sensitivity to irradiation and therefore the amount of radiation correction may vary for individual radiosondes, presumably related to the production process of the thermistors..."?;

→ The first paragraph is worded more clearly.

**Before:** A difficulty faced during the radiation correction of these thermistors is the unit difference in terms of the correction value.

**After (Line 180-183):** The purpose of the calibration of thermistors and the investigation of the temperature effect on radiosonde electronics boards is to assess the accuracy (or uncertainty) of raw temperature measurement before radiation correction. The next step is to investigate the sensitivity of individual thermistors to irradiation because the amount of radiation correction varies for individual radiosondes, presumably related to the production process of the thermistors.

**L161:** "Irregularities in the construction of the leads connecting sensor and boom…" It is interesting that these variations in the properties of the sensors, such as e.g. its diameter, have such a big influence. The authors should discuss this in more detail. A helpful reference may be de Podesta et al. (2018) (DOI: 10.1088/1681-7575/aaaa52).

→ The size of glass beads is irregular according to the manufacturer specification. The distribution of the size is specified in the revised manuscript. We also suspect the connection part between the sensor leads and the boom for one of the reasons of the observed difference of individual radiation test in Fig. 5(d) because the soldering and the coating of epoxy resin were conducted manually. Radiative heating of the glass beads, leads, and connection parts should be affected by their size as in the case of de Podesta *et al*.

**Before:** ellipsoidal shape with 0.55 mm diameter and 1.1 mm length.

**After (Line 96-97):** ellipsoidal shape with $0.55 \pm 0.1$ mm diameter and $1.1 \pm 0.3$ mm length

**Before:** The irregular connection between the sensor and boom may affect the thermal conduction between them.

**After (Line 186-188):** The connection between the sensor leads and the boom may be irregular because the soldering and the coating of epoxy resin were conducted manually. Radiative heating of glass beads, leads, and connection parts between the sensor leads and the boom should be affected by their size as previously reported (De Podesta et al., 2018).

**Second paragraph:**

What irradiance is applied? What are the conditions with regard to air pressure and ventilation in the RRT? Is the pump used to vary the pressure, or to create an airflow, or both? If there is no significant ventilation I would expect a certain variation of the results from that, because the cooling efficiency should strongly vary with air flow at low or vanishing flow rates. That might at least partially explain the distributions in Figs. 5 (c) and (d). If the ventilation is controlled, is it adjusted similar to what is used in the UAS (~5 m·s−1 )?

→ The RRT irradiance at the sensor position is 800 $W \cdot m^{-2}$ with 0.8% standard deviation for each irradiation. The ventilation and air pressure depends on the performance of the vacuum pump and the sealing of the chamber lid using an O-ring. Unfortunately, these factors were not

monitored. Thus, there can be variations in the air flow and the pressure that may vary the cooling efficiency. We agree that the RRT system needs an improvement such as the measurement of air pressure at least. This point is explained in the revised manuscript.

**Added statement (Line 192-195):** The RRT irradiance at the sensor position is 800 W·m$^{-2}$ with 0.8% standard deviation for each irradiation. The ventilation and the pressure in the chamber are not measured. Since they depend on the performance of the vacuum pump and the sealing of the chamber lid using an O-ring, there can be slight variations in the ventilation and the pressure.

**Added statement (Line 210-213):** Although the irradiance is constant for each sensor, the cooling efficiency of the sensors may vary depending on the bead size of thermistors, air flow, and the pressure. Slight variations of air flow and/or pressure in the RRT chamber (not monitored) may partly be responsible for the observed distributions of radiative heating of the sensors in Figs. 5(c) and (d).

**Fig. 5 (b):** Rad. warming of Al-coated is more than one third of that for the black thermistor in Fig. 5 (b), and the absolute value of ~1.2 K in the example seems unexpectedly large at a first glance (is that a typical example?). Does that mean that the reflectivity of the Al-coating is not that close to one, but say ~0.7 or so?

→ The average ratio of the radiative heating of aluminium-coated and black sensors is 1:2.4 in the RRT experiment. We think that the reflectivity of Al-coating cannot be simply estimated using these measurements because the radiative heating of sensors is affected by various parameters. Previously, when we measured the reflectance of Al coating of the DTR from a different batch (not this work), the reflectance was 0.8-0.9 below 1000 nm in wavelength and higher than 0.9 above 1000 nm in wavelength.

**Added statement (Line 206-207):** The average ratio of the radiative heating of aluminium-coated and black sensors is 1:2.4 in the RRT experiment.

Quantitative information on irradiance (is it the 960 W·m−2?), pressure and ventilation for the RRT tests would be helpful to better assess or classify the results.

→ The RRT irradiance at the sensor position is 800 W·m$^{-2}$ with 0.8% standard deviation for each irradiation. The ventilation and the pressure are not measured but they rely on the performance of the vacuum pump with the sealing of the lid using O-ring. This point is mentioned in the earlier comment.

Is there a test to see if the two thermistors influence each other (e.g. via heat conduction)? This could be assessed by selective irradiation.

→ We have not thought about the idea nor conducted such experiment.

How are the T-differences extracted/evaluated from the data in Fig. 5 (b)?

→ The temperature rise by the irradiation in Fig. 5(b) is determined by the difference of the average temperature for the last 30 seconds (30 data) before the shutter is opened and closed. The mean temperature rise of the three repeated measurements is assigned as the RRT value for each pair of thermistors in Fig. 5(c) and (d).

**Before:** Then, the shutter is opened and a pair of thermistors are illuminated for 180 s. Temperatures of the white ($T_W$) and black ($T_B$) sensors are recorded (Fig. 5(b)), and 107 pairs of dual thermistors are tested in total.

**After (Line 202-206):** Then, the shutter is opened and closed for 180 s each and this process is repeated three times for the illumination on each pair of thermistors. The temperatures of the white ($T_W$) and black ($T_B$) sensors are recorded (Fig. 5(b)), and 107 pairs of dual thermistors are tested in total. The temperature rise by the irradiation is determined by the difference of the average temperature for the last 30 seconds (30 data) before the shutter is opened and closed. The mean temperature rise of the three repeated measurements is assigned as the RRT value for each pair of thermistors.

**Section 4.1:**

**First paragraph:** (How) Is the angle of the sensor boom, i.e. the irradiation angle and boom orientation, and the angle relative to the air flow taken into account in the UAS measurements?

→ The sensor boom is installed upside down in parallel with the air flow and perpendicular to the irradiation direction.

**Added statement (Line 218-219):** The DTR is installed upside down in the test chamber of the UAS with the thermistors and the sensor boom in parallel with the air flow but perpendicular to the irradiation.

**L190:** Better: "… with the fitting coefficients being functions of T_W_on…"

→ The phrase is changed as suggested.

**Before:** are the fitting coefficients with a $T_{W\_on}$ function

**After (Line 228):** are the fitting coefficients being functions of $T_{W\_on}$

**L193/194:** Isn't the point here that the effective long-wave cooling is different for the two thermistors, according to the different emissivities, whereas the SW-absorption does not depend on T?

→ The effect of temperature on the degree of the temperature difference between two thermistors is observed. This is because the convective heat transfer between the sensor and air is reduced as the thermal conductivity of the air is decreased at cold temperatures as studied in Lee *et al.* Atmos. Meas. Tech. Discuss. *in press*.

**Before:** Interestingly, ($T_{B\_on} - T_{W\_on}$) gradually increases with decreasing sensor temperature in the UAS. A similar phenomenon was previously observed in a chamber with no apparent air ventilation (Lee et al., 2018a). A possible reason might be that the long-wave radiation from thermistors decreases with the environmental temperature even though the irradiance is maintained at cold temperatures.

**After (Line 230-233):** Interestingly, the level of ($T_{B\_on} - T_{W\_on}$) gradually increases as the temperature decreases especially for low pressures. A similar phenomenon was previously observed in a chamber with no apparent air ventilation (Lee et al., 2018a). The observed effect of temperature on ($T_{B\_on} - T_{W\_on}$) is because the convective heat transfer between the sensor and air is reduced as the thermal conductivity of the air is decreased at cold temperatures (Lee et al., 2021).

**Eq. (2):** What motivated the exponential functions as fitting model?

→ The exponential functions are purely empirical because they agree well with the experimental data.

**L216:** The radiation flux of 960 W·m−2 is known. Is the (re)fitting done for the purpose of estimating uncertainties in terms of the residuals?

→ Eq. (9) and Eq. (10) are corrected to deliver the meaning clearly. The backbone of Eq. (9) and (10) is Eq. (2). Since Eq. (2) is obtained when the radiation flux is 960 W·m$^{-2}$, an unknown radiation flux can be measured by the temperature difference of two thermistors using the linear relationship with the ($T_{B\_on} - T_{W\_on}$)$_{UAS}$ at 960 W·m$^{-2}$ (the denominator in Eq. (9)). Therefore, the ($T_{B\_on} - T_{W\_on}$)$_{UAS}$ in Eq. (9) and (10) should be replaced by ($T_{B\_raw} - T_{W\_raw}$) where $T_{B\_raw}$ and $T_{W\_raw}$ are raw temperatures of the black and white sensors, respectively.

**Before:** Hence, Eq. (2) is employed to measure the *in-situ* irradiance based on the fact that ($T_{B\_on} - T_{W\_on}$)$_{UAS}$ is linearly proportional to S.

**After (Line 250-252):** Hence, Eq. (2) is employed to measure the *in-situ* irradiance using ($T_{B\_raw} - T_{W\_raw}$), where $T_{B\_raw}$ and $T_{W\_raw}$ are raw temperatures of the black and white sensors,

respectively, based on the fact that the temperature difference between two sensors is linearly proportional to $S$.

**Before:** $S = S_0 \times (T_{B\_on} - T_{W\_on})_{UAS} \,/\, [T_0(T_{W\_on}) + A_0(T_{W\_on})\cdot\exp(-P/P_0(T_{W\_on})) + A_1(T_{W\_on})\cdot\exp(-P/P_1(T_{W\_on}))]$ , $S_0 = 960$ W·m$^{-2}$

**After (Eq. (9)):** $S = S_0 \times (T_{B\_raw} - T_{W\_raw}) \cdot (T_{B\_on} - T_{W\_on})_{UAS}^{-1}$ ,

**Before:** $S = S_0 \times (T_{B\_on} - T_{W\_on})_{UAS} \,/\, [T_0(T_{W\_on}) + A_0(T_{W\_on})\cdot\exp(-P/P_0(T_{W\_on})) + A_1(T_{W\_on})\cdot\exp(-P/P_1(T_{W\_on})) - 0.08\cdot(v - v_0)]$, $S_0 = 960$ W·m$^{-2}$ and $v_0 = 5$ m·s$^{-1}$

**After (Eq. (10)):** $S = S_0 \times (T_{B\_raw} - T_{W\_raw}) \cdot [(T_{B\_on} - T_{W\_on})_{UAS} - 0.08\cdot(v - v_0)]^{-1}$,

**Eq. (10):** Is the equation valid only for 7-100 hPa and 4-6.5 m·s$^{-1}$?

→ The effect of ventilation speed is investigated in the range of $v = 4\text{-}6.5$ m·s$^{-1}$ and $P = 7\text{-}100$ hPa. The sensitivity coefficient against the ventilation is -0.08 °C / (m·s$^{-1}$) averaged over the pressure range. The coefficient will be significantly bigger when $v$ is lower than 4 m·s$^{-1}$ while it will be a bit smaller when $P$ is higher than 100 hPa.

**Added statement (Line 264-265):** The absolute value of the sensitivity coefficient (-0.08 °C / (m·s$^{-1}$)) against the ventilation speed will be significantly bigger when $v$ is lower than 4 m·s$^{-1}$ while it will be a bit smaller when $P$ is higher than 100 hPa.

**Fig. 6 (also Fig. 7):** The trend of the data points is difficult to see at low p; consider using a logarithmic scale.

→ Logarithmic scale is used for Figs. 6 and 7.

**Modified Figures (Figures 6 & 7):** Logarithmic scale is used for x-axis of Figures 6 and 7.

**Section 4.2:**

**Eq. (19):** Validity range with regard to p and v?

→ The effect of ventilation speed is investigated in the range of $v = 4\text{-}6.5$ m·s$^{-1}$ and $P = 7\text{-}100$ hPa. The sensitivity coefficient against the ventilation is -0.1 °C / (m·s$^{-1}$) averaged over the pressure range. The coefficient will be significantly bigger when $v$ is lower than 4 m·s$^{-1}$ while it will be a bit smaller when $P$ is higher than 100 hPa.

**Added statement (Line 312-314):** The absolute value of the sensitivity coefficient (-0.1 °C / (m·s$^{-1}$)) against the ventilation speed will be significantly bigger when $v$ is lower than 4 m·s$^{-1}$ while it will be a bit smaller when $P$ is higher than 100 hPa.

**L234f:** Please discuss the influence of the temperature dependence of the efficiency of convective cooling, i.e. at low temperatures the thermal conductivity of air decreases leading to an increase of the radiative heating of the temperature sensor.

→ This manuscript was written before the revision of the previous UAS paper (amt-2021-246). The influence of the temperature is discussed in line with the previous paper.

**Before:** $(T_{W\_on} - T_{W\_off})_{UAS}$ gradually increases with decreasing sensor temperature ($T_{W\_on}$) in UAS. This is attributed to the decrease of the long-wave radiation from thermistors at cold temperatures, despite the constant irradiation.

**After (Line 278-281):** $(T_{W\_on} - T_{W\_off})_{UAS}$ shows a temperature dependency. $(T_{W\_on} - T_{W\_off})_{UAS}$ at −68 °C is 118.9 ± 3.5% (mean ± SD of five units) of that at 20 °C, when $P = 5$ hPa. In the previous study, the ratio for RS41 investigated by the same manner is 119% (Lee et al., 2021). This is attributed to the decrease of the thermal conductivity of air at cold temperatures, which reduces the heat transfer from the sensor to air despite the constant irradiation.

**Section 5:**

At what time was the daytime sounding performed? What was the cloud situation?

→ The daytime sounding was performed from 11:00 am to 5 pm local time while the nighttime sounding was from 12:00 am to 4 am. is cloudy

**Added statement (Line 321-322):** The daytime sounding was performed from 11:00 am to 5 pm local time while the nighttime sounding was from 12:00 am to 4 am. The sky was normally cloudy.

Add more discussion on what is observed in the plots in Fig. 8. The reconstructed solar (ir)radiance decreases with altitude in the stratosphere. This is opposite to what is expected. Please compare the reconstructed radiation profile to RTM calculations, and discuss the differences. See for example Philipona et al 2020 (doi: 10.1127/metz/2020/1044) for in situ measurements of the radiation profile.

→ The negative irradiance in the stratosphere at nighttime is very clear because the temperature of the black sensor is distinctively lower than the white sensor above 20 km at nighttime (Fig. 8d). Previously, the same phenomenon was observed in the Figure 4b of Rolf Philipona *et al.* (2013) in Journal of Atmospheric and Oceanic Technology (DOI: 10.1175/JTECH-D-13-00047.1). The authors explained that "The sum of the absorbed and emitted fluxes result in the longwave radiation balance of the sensor during the night (LRB_n), which is negative in the lower troposphere, and then becomes positive and again negative further up in the stratosphere.

Hence, in the lower troposphere, LRB_n cools the temperature sensor, higher up LRB_n warms the temperature sensor, and above 25 km it again cools the temperature sensor." The effect of long-wave radiation was also applied to the daytime as they mentioned that "The longwave radiative impact on the sensor from above and from below is very similar during the day and during the night (Fig. 4c)." Indeed, the pattern of LRB_d at daytime shown in Fig. 4c of Philipona *et al.* is similar to that of the LRB_n at nighttime in Fig. 4b. Therefore, the decrease of the effective irradiance in the stratosphere at daytime is highly likely due to the negative longwave radiation balance of the sensors.

**Added statement (Line 346-356):** The negative net irradiance at nighttime was also observed in the previous work for the radiation correction of radiosondes based on the measurement of radiative flux profiles using two pyranometers for measuring downward and upward solar short-wave radiation, and two pyrgeometers for measuring upward and downward thermal long-wave radiation (Philipona et al., 2013). The long-wave radiation balance (LRB) of the sensor defined by the sum of the absorbed and emitted fluxes corresponds to the effective irradiance at nighttime in this work. Both the LRB in the work of Philipona *et al.* and the effective irradiance at nighttime in this work are negative in the lower troposphere, and then become positive and again negative further up in the stratosphere. This means that temperature sensors of radiosondes are cooled in lower troposphere, warmed in higher up, and again cooled further up in the stratosphere and thus should be corrected accordingly. The profile of the LRB at nighttime was similar to that of daytime (Philipona et al., 2013). In this regard, the decrease of the effective irradiance in the stratosphere observed at daytime is highly likely due to the negative LRB of the sensors as observed in the nighttime soundings.

In particular, it could be discussed more how the different long-wave backgrounds in the experiments and in the situations during the real soundings may influence the results, or what potential systematic errors of the radiation correction may be in connection to this. Can it be assumed without reservation that the sensitivity of the thermistors is the same with regard to long-wave and short-wave radiation (wavelength-independent emissivity/absorptivity)?

→ The radiation correction formula of the DTR is obtained based on the portion of the long-wave and the short-wave radiation from a solar simulator used as a radiation source in the UAS experiments. As the reviewer pointed out, the emissivity/absorptivity is dependent on the radiation wavelength. In this regard, the temperature rise of the DTR upon irradiation can be affected by the actual ratio of the long-wave and the short-wave radiation in soundings. For aluminum coating, the reflectance is 0.8−0.9 below 1000 nm in wavelength and 0.9 above 1000 nm. This means that the influence of the ratio between the long-wave and the short-wave radiation will be within a few percent of the radiative heating of the DTR even when the portion below 1000 nm is drastically different between the laboratory experiments and soundings.

**Added statement (Line 328-335):** The radiation correction formula of the DTR is obtained based on the portion of the long-wave and the short-wave radiation from the solar simulator used as a radiation source in the UAS experiments. The emissivity and absorptivity are

dependent on the wavelength. In this regard, the radiative heating of the DTR in soundings can be affected by the actual ratio of the long-wave and the short-wave radiation. For aluminium coating, the reflectance was 0.8−0.9 below 1000 nm and 0.9 above 1000 nm in wavelength. This means that the influence of the ratio between the long- and short-wave radiation would be a few percent of the radiative heating of the DTR even when the portion below 1000 nm is drastically different between the laboratory experiments and soundings.

Mention in the caption of Fig. 8 that the effective irradiance is calculated using Eq. (9) (or 10).

→ The caption of Fig. 8 is modified.

**Before:** (b) calculated effective irradiance based on ($T_{B\_raw} - T_{W\_raw}$) and (c) radiation correction value of the white sensor at daytime.

**After (Figure 8 Caption):** (b) effective irradiance based on ($T_{B\_raw} - T_{W\_raw}$) calculated by Eq. (10) and (c) radiation correction value of the white sensor at daytime calculated by Eq. (19).

**Section 6.2**

**L340/341:** "enhance" -> "improve" or "reduce"; It should be discussed here (or in 3.2) whether and how the calibration uncertainty, which is obviously due to air temperature inhomogeneities within the calibration volume, can be reduced.

→ The word is changed as suggested. Reduction of calibration uncertainty is discussed in earlier comments.

**Before:** enhance

**After (Line 410):** improve

**Section 6.3**

**Fig. 9 (f):** The >0.2 K offset at ~16 km is striking, please comment on this.

→ We think that the number of nighttime soundings is not enough to smooth out the averaged profiles.

---

## Author Comment (AC2)

**Referee #2**

**General Comment:** The manuscript by Lee et al. provides the description and the metrological characterisation of a dual thermistor radiosonde (DTR) comprising two sensors with different emissivities. The work described in the manuscript is done and continuity with previous efforts of scientific community and of the authors themselves. The use of DTR has the main objective to improve the adjustment of the daytime solar radiation effect on the temperature sensors which is a critical issue for all radiosonde types.

The manuscript is well structured and, beyond the introduction, sufficiently well written. The metrological characterisation of the DTR is detailed and comprises of the major steps needed to fully characterise investigated sensor. Nevertheless I have major concerns about the scientific quality of the manuscript which are detailed below.

There is a strong imbalance between the metrological characterisation of the DTR and the scientific discussion related to the assessment of the DTR performance in the atmosphere.

→ We thank the Reviewer for valuable comments. We have added more discussions on the performance of the DTR to address the Reviewer's comments. The specific changes are made as below.

**1.** The metrological characterisation of the solar radiation correction and the quantification of the uncertainty budget for the DTR are not supported by sufficient validation of results: the comparison with the RS41, which can be assumed in the context of the manuscript as a "community standard" (although the only reference is known to be the cryogenic frostpoint hygrometer), is quickly presented and lacks of discussion for the differences shown in Fig. 9. This a very important aspect to show the performances of the DTR measurements. A broader discussion on the consistency with the RS41 should be included.

→ More discussion is added as responses to this comment and the later comment in Section 6.3. Please see the response to the comment of Line 349-353 in original manuscript or Line 416-432 in revised manuscript.

**2.** The experimental set up discussed in the manuscript looks sufficiently robust, although this could be compared with previous experiments available in literature in order to show what are the pros and cons of the metrological characterization carried out by the authors. A few assumptions in the manuscript must be well justified by means of references or quantitative results. For example, when the term $(T_{B\_raw} - T_{W\_raw})$ is used instead of $(T_{B\_on} - T_{W\_on})_{UAS}$ to obtain the irradiance using Eq. (10) in the adjustement of the DTR measured profile, this is done without justifying their effect on the measurements compared to the results obtained in the laboratory experiments. Also the uncertainty due to the solar radiation correction is smaller

than for other radiosondes (according to the literature) and a comparative discussion could be provided in the benefit of the reader.

→ Although both $(T_{B\_raw} - T_{W\_raw})$ and $(T_{B\_on} - T_{W\_on})_{UAS}$ are linearly proportional to the total radiation flux, the portion of the long-wave and the short-wave radiation may affect them because the emissivity and absorptivity of thermistors are dependent on the radiation wavelength. For aluminium coating, the reflectance is 0.8−0.9 below 1000 nm in wavelength and 0.9 above 1000 nm. This implies that the influence of the portion of the long- and short-wave radiation would be a few percent of the radiative heating of the DTR even when the portion below 1000 nm is drastically different between the laboratory experiments and soundings.

**Added statement (Line 328-335):** The radiation correction formula of the DTR is obtained based on the portion of the long-wave and the short-wave radiation from the solar simulator used as a radiation source in the UAS experiments. The emissivity and absorptivity are dependent on the wavelength. In this regard, the radiative heating of the DTR in soundings can be affected by the actual ratio of the long-wave and the short-wave radiation. For aluminium coating, the reflectance was 0.8−0.9 below 1000 nm and 0.9 above 1000 nm in wavelength. This means that the influence of the ratio between the long- and short-wave radiation would be a few percent of the radiative heating of the DTR even when the portion below 1000 nm is drastically different between the laboratory experiments and soundings.

→ The uncertainty due to the solar radiation correction is smaller than other radiosondes because the DTR measures the effective irradiance while others use the simulated irradiance constructed from the average of clear and cloudy sky cases. This results in the increase of the radiation correction uncertainty in troposphere.

**Line 72-74: (Before)** the SI traceability of the simulated irradiance is incomplete and may increase the radiation correction uncertainty → **(After)** the SI traceability of the simulated irradiance is incomplete if the sky is clear or cloudy because the simulated irradiance is constructed from the average of clear and cloudy sky cases (Von Rohden et al., 2022). This results in the increase of the radiation correction uncertainty in troposphere.

**3.** The manuscript introduction is not well written and lacks of fairness and accuracy. I strongly recommend the authors to pay great care in writing inaccurate or wrong statements. Specific comments on the introduction are provided in a revised pdf version of the manuscript along with several specific comments, attached to this review.

→ The specific comments in the pdf version are addressed as below.

**Line 10:** probes?

**Line 10:** sensors → probes

**Line 12:** radiosounding

**Line 12:** sounding → radiosounding

**Line 26:** If you refer to Bojiinski please use the term "Essential Climate Variables".

**Line 30:** environmental parameters → essential climate variables

**Line 29:** repetition

**Line 33:** environmental parameters → meteorological parameters

**Line 29:** collected measuements

**Line 33:** measurement → collected measurement

**Line 30-31:** This sentence is quite generic, shoudl refer to many other aspect like time integration, representativeness, different atmospehric regions. I recommend to remove.

**Removed sentence:** Radiosonde measurements are known to be more accurate than measurement methods based on remote sensing techniques, such as LIDAR and satellite.

**Line 32:** "for calibrating" or validating. This paragraph is a bit uncorrect. Must be rephrased.

**Line 34-36: (Before)** Thus, radiosonde measurements are often used as reference to correct other measurement data. → **(After)** Radiosonde observations are often co-located with global navigation satellite system radio occultation and used as reference for validating their one-dimensional interpolation which follows the flight trajectories of balloon soundings.

**Line 32:** This is inpendent on the needs ot callibrating /validating other techniques.

**Removed word:** Hence,

**Line 37-38:** if GCOS is referring to the "meteorolgical community" this is evidently wrong because this is a climate initiative by WMO

**Removed phrase:** , such as the Global Climate Observing System (GCOS)

**Line 39:** Replace with "in the upper troposphere/lower stratosphere".

**Line 42-43: (Before)** when the project deliverables were limited to the field of upper air measurement by radiosondes → **(After)** in the environments imitating the upper troposphere/lower stratosphere

**Line 40:** investigate or detect.

**Line 43:** resolve → investigate

**Line 40:** remove

**Removed phrase:** long term

**Line 40-41:** unclear, rephase

**Line 43-44: (Before)** a certain amount of measurement uncertainty of radiosonde sensors is required for actual soundings in addition to the ground-based facilities → **(After)** a certain level of measurement uncertainty in radiosoundings should be secured.

**Line 42-43:** replace with "a dataset of traceable measurements with quantified uncertainties"

**Line 45-46: (Before)** the highest level of quality control in the upper air measurement in terms of uncertainty → **(After)** a dataset of traceable measurements with quantified uncertainties

**Line 43-44:** Please refer in the numbers and citation to Dirksen et al 2014

**Line 55-57 (Added statement):** The uncertainty of the GDP of RS92 during daytime was gradually increased from 0.2 °C at the surface to 0.6 °C at 30 km with the coverage factor $k = 2$ (Dirksen et al., 2014).

**Line 59-61 (Added statement):** Using the setup, the uncertainty of the GDP of RS41 is evaluated to be about 0.3 °C ($k$ = 2) at 35 km. It is also found that the daytime GRUAN profile is 0.35 °C warmer than the manufacturer's at 35 km (Von Rohden et al., 2022).

**Line 45:** replace with "adjustment" or "correction".

**Line 48:** compensation → correction

**Line 50-51:** this sentence must be rephrases to say way otherwise remove, the following sentence is sufficient.

**Removed sentence:** Thus, the uncertainty evaluation on the radiation correction by manufacturers cannot be fully accepted.

**Line 54:** remove

**Removed word:** algorithm

**Line 61:** remove this is already clear from the context

**Removed phrase:** long term

**Line 64:** replace with "measuring on the same payload of radiosondes"

**Line 69-70:** during soundings → measuring on the same payload of radiosondes

**Line 67-68:** one or more references here may help

**Line 72-74: (Before)** the SI traceability of the simulated irradiance is incomplete and may increase the radiation correction uncertainty → **(After)** the SI traceability of the simulated irradiance is incomplete if the sky is clear or cloudy because the simulated irradiance is constructed from the average of clear and cloudy sky cases (Von Rohden et al., 2022). This results in the increase of the radiation correction uncertainty in troposphere.

**Line 71:** put a comma

**Line 77-78:** air ventilation and temperature → air ventilation, and temperature

**Line 73:** replace with "showed to be"

**Line 79:** was → showed to be

**Line 79:** remove

**Removed word:** formulas

**Line 83:** be consistent in using the tense of verbs

→ Present tense is used in general unless conducted earlier than this work.

**Line 85-86: (Before)** The obtained formulas were used in 80 an intercomparison sounding test that was performed in July, 2021. → **(After)** The obtained formulas are used in an intercomparison sounding test performed in July, 2021.

**Line 85:** add "from parallel soundings"

**Line 91 (Added phrase):** from parallel soundings

**Line 85:** repeated verb

**Line 90-92: (Before)** Finally, the difference between the corrected temperatures of DTR and the RS41 radiosonde is presented with the DTR uncertainty → **(After)** Finally, the corrected temperatures of the DTR and the RS41 from parallel soundings are compared and the difference between them is discussed in terms of the uncertainty.

**Line 106-107:** May be this sentence could be removed, need too many details which are discussed later in the text.

**Removed sentence:** Then, the effective irradiance and other environmental factors are parameterised into a single equation to calculate the radiation correction of the white sensor $(T_{W\_raw} - T_{W\_cor})$.

**Line 131-132:** It would be interesting to show the difference between the second-order and third order polynomial. I recomment to add these resutls to the mansuscript.

→ A separate paper from our group which clearly shows the difference is added as a reference.

**Line 145:** (not shown here) → (Yang et al., 2021)

**Line 141:** use a different way to indicate the thermistors to avoid confusion with the other sensors in the chamber.

**Line 153 & 155:** sensors → sensors (thermistors)

**Line 156:** why k=1 and not k=2? Please clarify

→ The probability distribution is assumed to be a normal distribution function and the standard deviation is used for standard uncertainty ($k = 1$).

**Line 175-178: (Before)** Since the temperature error is roughly distributed within ±0.05 °C, the standard deviation of all data points is obtained (0.04 °C) and is used for uncertainty (k = 1) due to the influence of temperature on the resistance reading by radiosonde boards. → **(After)** Assuming that the probability distribution is a normal distribution function, the standard deviation (SD) of all data points (0.04 °C) is used for standard uncertainty due to the influence of the temperature of radiosonde electronics boards on the resistance (or temperature) measurement.

**Line 158:** not 100% clear, please expand the sentence. The following sentence clarifies but the reading of this unit is not fully clear.

**Line 180-183: (Before)** A difficulty faced during the radiation correction of these thermistors is the unit difference in terms of the correction value. → **(After)** The purpose of the calibration of thermistors and the investigation of the temperature effect on radiosonde electronics boards is to assess the accuracy (or uncertainty) of raw temperature measurement before radiation correction. The next step is to investigate the sensitivity of individual thermistors to irradiation because the amount of radiation correction varies for individual radiosondes, presumably related to the production process of the thermistors.

**Line 176:** The count distributions shown in Figure 5 are not discussed here at all: no statistical indicators are reported, commented, for example nothing is said why the distribution in Figure 5d is skewed while not that in Figure 5c. Are there potential hysteresis or stabilization effects? Then you must describe in the text your setup in comparison to other available in literature and describe pros and cons. As it stands the descritpion is very minimal.

→ Various factors of irregularity in the production process of the thermistors and the environments of the RRT chamber can be responsible for the count distributions in Fig. 5. The size of glass beads is irregular according to the manufacturer specification. The distribution of the size is specified in the revised manuscript. We also suspect the connection part between the sensor leads and the boom for one of the reasons of the observed difference of individual radiation test in Fig. 5(d) because the soldering and the coating of epoxy resin were conducted

manually. The ventilation and air pressure depends on the performance of the vacuum pump and the sealing of the RRT chamber lid using an O-ring. Unfortunately, these factors were not monitored. Thus, there can be variations in the air flow and the pressure that may vary the cooling efficiency.

**Line 96-97: (Before)** ellipsoidal shape with 0.55 mm diameter and 1.1 mm length → **(After)** ellipsoidal shape with 0.55 ± 0.1 mm diameter and 1.1 ± 0.3 mm length

**Line 186-188: (Before)** The irregular connection between the sensor and boom may affect the thermal conduction between them. → **(After)** The connection between the sensor leads and the boom may be irregular because the soldering and the coating of epoxy resin were conducted manually. Radiative heating of glass beads, leads, and connection parts between the sensor leads and the boom should be affected by their size as previously reported (de Podesta et al., 2018).

**Line 210-213 (Added statement):** Although the irradiance is constant for each sensor, the cooling efficiency of the sensors may vary depending on the bead size of thermistors, air flow, and the pressure. Slight variations of air flow and/or pressure in the RRT chamber (not monitored) may partly be responsible for the observed distributions of radiative heating of the sensors in Figs. 5(c) and (d).

**Line 197-201:** I suggest to use a generic notation for all the firve equations

→ We would like to use the current form of equations because they are straightforward with coefficients in Table 1.

**Line 211:** soundings

**Line 250:** sounding → soundings

**Line 214:** better to use a different way to indicate equation, this one may generate confusion

→ Eq. (9) and Eq. (10) are simplified to deliver the meaning clearly. The backbone of Eq. (9) and (10) is Eq. (2). Since Eq. (2) is obtained when the radiation flux is 960 $W \cdot m^{-2}$, an unknown radiation flux can be measured by the temperature difference of two thermistors using the linear relationship with the $(T_{B\_on} - T_{W\_on})_{UAS}$ at 960 $W \cdot m^{-2}$ (the denominator in Eq. (9)). Therefore, the $(T_{B\_on} - T_{W\_on})_{UAS}$ in Eq. (9) and (10) should be replaced by $(T_{B\_raw} - T_{W\_raw})$ where $T_{B\_raw}$ and $T_{W\_raw}$ are raw temperatures of the black and white sensors, respectively.

**Line 250-252: (Before)** Hence, Eq. (2) is employed to measure the *in-situ* irradiance based on the fact that $(T_{B\_on} - T_{W\_on})_{UAS}$ is linearly proportional to *S*. → **(After)** Hence, Eq. (2) is employed to measure the *in-situ* irradiance using $(T_{B\_raw} - T_{W\_raw})$, where $T_{B\_raw}$ and $T_{W\_raw}$ are

raw temperatures of the black and white sensors, respectively, based on the fact that the temperature difference between two sensors is linearly proportional to $S$.

**Eq. (9): (Before)** $S = S_0 \times (T_{B\_on} - T_{W\_on})_{UAS} / [T_0(T_{W\_on}) + A_0(T_{W\_on}) \cdot \exp(-P/P_0(T_{W\_on})) + A_1(T_{W\_on}) \cdot \exp(-P/P_1(T_{W\_on}))]$ , $S_0 = 960$ W·m$^{-2}$

→ **(After)** $S = S_0 \times (T_{B\_raw} - T_{W\_raw}) \cdot (T_{B\_on} - T_{W\_on})_{UAS}^{-1}$ ,

**Eq. (10): (Before)** $S = S_0 \times (T_{B\_on} - T_{W\_on})_{UAS} / [T_0(T_{W\_on}) + A_0(T_{W\_on}) \cdot \exp(-P/P_0(T_{W\_on})) + A_1(T_{W\_on}) \cdot \exp(-P/P_1(T_{W\_on})) - 0.08 \cdot (v - v_0)]$, $S_0 = 960$ W·m$^{-2}$ and $v_0 = 5$ m·s$^{-1}$

→ **(After)** $S = S_0 \times (T_{B\_raw} - T_{W\_raw}) \cdot [(T_{B\_on} - T_{W\_on})_{UAS} - 0.08 \cdot (v - v_0)]^{-1}$,

**Line 216-217:** redundant, remove

**Removed phrase:** by Eq. (9) to estimate the irradiance (960 W·m$^{-2}$) by using $(T_{B\_on} - T_{W\_on})_{UAS}$, $(T_{B\_on} - T_{W\_on})_{RRT}$ and other environmental parameters

**Line 222-223:** Do the residuals reduce after including the effect of air ventilation? Please provide details and maybe amend the plot in Figure 6.

→ The ventilation speed for five units in Figure 6 is fixed at 5 m·s$^{-1}$ and thus the effect of air ventilation cannot be identified. The effect of air ventilation is studied with a separate pair of thermistors. The standard deviation of the residual for the pair of thermistors is 4.1% with Eq. (9) while it is reduced to 3.4% with Eq. (10) when the air ventilation is actually changed (4–6.5 m·s$^{-1}$).

**Line 258-259 (Added statement):** and thus the effect of air ventilation cannot be identified in Fig. 6(f).

**Line 259 (Added statement):** by a separate pair of thermistors

**Line 263-264 (Added statement):** The standard deviation of the residual for a pair of thermistors is 4.1% with Eq. (9) while it is reduced to 3.4% with Eq. (10) when the air ventilation is actually changed (4–6.5 m·s$^{-1}$).

**Line 245:** same comment as above

→ We would like to use the current form of equations because they are straightforward with coefficients in Table 2.

**Line 265:** Same comment as at line 222 and not clear why residuals are not evaluated after the inclusion of wind speed. Please motivate.

→ As mentioned, the effect of air ventilation is studied with a separate pair of thermistors. The ventilation speed for five units in Figure 7 is fixed at 5 m·s$^{-1}$ and thus the effect of air ventilation cannot be identified. The standard deviation of the residual for a pair of thermistors is 0.10 °C with Eq. (18) while it is reduced to 0.04 °C with Eq. (19) when the air ventilation is actually changed (4–6.5 m·s$^{-1}$).

**Line 306-307 (Added statement):** and thus the effect of air ventilation cannot be identified in Fig. 7(f).

**Line 307 (Added statement):** by a separate pair of thermistors

**Line 311-314 (Added statement):** When the air ventilation is actually changed (4–6.5 m·s$^{-1}$), the standard deviation of the residual for a pair of thermistors is 0.10 °C with Eq. (18) while it is reduced to 0.04 °C with Eq. (19).

**Line 272:** Please say ho many sounding were performed during the intercomparison.

→ More information on soundings is added.

**Line 319-322 (Added statement):** One, two, or three DTRs were tested in parallel with a RS41 in a single flight. The number of comparison (N) was $N = 12$ at daytime and $N = 6$ at nighttime from 7 and 3 soundings, respectively. The daytime sounding was performed from 11:00 am to 5 pm local time while the nighttime sounding was from 12:00 am to 4 am. The sky was normally cloudy.

**Line 274-275:** What's the effect of this assumption compared to the lab experiements? Do you have results to show? Otherwise, please comment.

→ → Although both ($T_{B\_raw} - T_{W\_raw}$) and ($T_{B\_on} - T_{W\_on}$)$_{UAS}$ are linearly proportional to the total radiation flux, the portion of the long-wave and the short-wave radiation may affect them because the emissivity and absorptivity of thermistors are dependent on the radiation wavelength. For aluminum coating, the reflectance is 0.8−0.9 below 1000 nm in wavelength and 0.9 above 1000 nm. This implies that the influence of the portion of the long- and short-wave radiation would be a few percent of the radiative heating of the DTR even when the portion below 1000 nm is drastically different between the laboratory experiments and soundings.

**Added statement (Line 328-335):** The radiation correction formula of the DTR is obtained based on the portion of the long-wave and the short-wave radiation from the solar simulator used as a radiation source in the UAS experiments. The emissivity and absorptivity are dependent on the wavelength. In this regard, the radiative heating of the DTR in soundings can be affected by the actual ratio of the long-wave and the short-wave radiation. For aluminium coating, the reflectance was 0.8−0.9 below 1000 nm and 0.9 above 1000 nm in wavelength. This means that the influence of the ratio between the long- and short-wave radiation would be

a few percent of the radiative heating of the DTR even when the portion below 1000 nm is drastically different between the laboratory experiments and soundings.

**Line 275:** Please report the time of the launch and more details on the payload configuration. These are important elements to evaluate the quality of the collected profiles.

→ More information on soundings is added in the earlier comment.

**Line 289-290:** Should this correction, event to a certain exent, be needed also during daytime? Please comment in clear way.

→ The negative irradiance in the stratosphere at nighttime is very clear because the temperature of the black sensor is distinctively lower than the white sensor above 20 km at nighttime (Fig. 8d). Previously, the same phenomenon was observed in the Figure 4b of Rolf Philipona *et al.* (2013) in Journal of Atmospheric and Oceanic Technology (DOI: 10.1175/JTECH-D-13-00047.1). The authors explained that "The sum of the absorbed and emitted fluxes result in the longwave radiation balance of the sensor during the night (LRB_n), which is negative in the lower troposphere, and then becomes positive and again negative further up in the stratosphere. Hence, in the lower troposphere, LRB_n cools the temperature sensor, higher up LRB_n warms the temperature sensor, and above 25 km it again cools the temperature sensor." The effect of long-wave radiation was also applied to the daytime as they mentioned that "The longwave radiative impact on the sensor from above and from below is very similar during the day and during the night (Fig. 4c)." Indeed, the pattern of LRB_d at daytime shown in Fig. 4c of Philipona *et al.* is similar to that of the LRB_n at nighttime in Fig. 4b. Therefore, the decrease of the effective irradiance in the stratosphere at daytime is highly likely due to the negative longwave radiation balance of the sensors.

**Added statement (Line 346-356):** The negative net irradiance at nighttime was also observed in the previous work for the radiation correction of radiosondes based on the measurement of radiative flux profiles using two pyranometers for measuring downward and upward solar short-wave radiation, and two pyrgeometers for measuring upward and downward thermal long-wave radiation (Philipona et al., 2013). The long-wave radiation balance (LRB) of the sensor defined by the sum of the absorbed and emitted fluxes corresponds to the effective irradiance at nighttime in this work. Both the LRB in the work of Philipona *et al.* and the effective irradiance at nighttime in this work are negative in the lower troposphere, and then become positive and again negative further up in the stratosphere. This means that temperature sensors of radiosondes are cooled in lower troposphere, warmed in higher up, and again cooled further up in the stratosphere and thus should be corrected accordingly. The profile of the LRB at nighttime was similar to that of daytime (Philipona et al., 2013). In this regard, the decrease of the effective irradiance in the stratosphere observed at daytime is highly likely due to the negative LRB of the sensors as observed in the nighttime soundings.

**Line 306:** The nighttime residual adjustment in the upper atmosphere due to long wave emission of the black thermistor at night is estimated on 6 soundings only. This increases the DTR uncertainty. And moreover, please, provide results above significance of the DTR at night time.

→ We agree that the data is not enough in this work and thus we are preparing for more intercomparison soundings this year to provide more decisive and diverse results of the DTR. As mentioned in the earlier comment, the long-wave radiation imbalance (cooling) of sensors at stratosphere should not be the issue of the DTR only because the same phenomenon was observed in the previous work based on the measurement of long-wave radiations. These findings suggest that an application of radiation correction is needed even at nighttime (RS41 applies no radiation correction at nighttime).

**Added statement (Line 374-376):** The long-wave radiation imbalance (cooling) of sensors at stratosphere should not be the issue of the DTR only because the same phenomenon was observed in the previous work based on the measurement of long-wave radiations. These findings suggest that an application of radiation correction is needed even at nighttime.

**Line 319:** In the formula a variance is reported, make the text and the formulas coherent.

**Line 398-400: (Before)** DTR uncertainty gradually increases up to about 0.35 °C in the troposphere and is maintained in the stratosphere. However, at nighttime, the uncertainty slightly is decreased to about 0.3 °C as the uncertainty of effective irradiance is decreased. → **(After)** DTR uncertainty gradually increases up to about 0.35 °C at the tropopause and is maintained in the stratosphere (0.33 °C at 30 km). However, at nighttime, the uncertainty slightly is decreased to 0.3 °C at the tropopause and 0.25 °C at 30 km as the uncertainty of effective irradiance is decreased.

**Line 338:** Again, sometimes the authors uses k=1, some other k=2, please ensure consistency.

→ The expanded uncertainty ($k = 2$) is used throughout manuscript.

**Line 346:** Again, how many soundings? This is a crucial elementn for the radiosonde intercomparison.

→ More information on soundings is added in the earlier comment.

**Line 349-353:** Comments in this section are quite faint and not extensive as they should be. This section should indeed containt a lot of elements to demostrate the performance of you dual thermistor sonde in comparison with the most popular sensor on the market. A deeper analysis

and comments should be done with a spirit similar to the discussion provided by Dirksent et al., 2014 when comparison RS92 GDP with CFH, assuming obvioulsy the latter as the reference.

Moreover, although requiriing additional work, the authors have the unique opportunity to compare two radiosondes types with the related unceratinties, so they should do their best to include RS41 uncertainty.

→ The manufacturer specifies that the uncertainty of RS41 is 0−0.3 °C in 0−16 km and 0.4 °C at 16 km and higher up (Vaisala, white paper). However, the detailed process to obtain the uncertainty such as laboratory experiments is not disclosed. Recently, we have obtained a radiation correction formula of RS41 under a well-defined irradiance in the UAS (Lee *et al.* Atmos. Meas. Tech., *in print*). However, the correction formula cannot be applied to RS41 because the manufacturer does not provide raw temperature in soundings. Even though raw temperature would be provided, the correction formula cannot be applied because the irradiance used by RS41 is unknown. In this regard, the GRUAN uses their own simulated irradiance constructed from the average of clear and cloudy sky cases in their recent publication on RS41 (von Rohden *et al.*, Atmos. Meas. Tech., 2022). The maximum uncertainty by the GRUAN is about 0.3 °C at $k = 2$ which is larger than our previous work (0.17 °C at $k = 2$) because the irradiance in our work is assumed to be 1360 W·m$^{-2}$ at stratosphere with a small (lab scale) uncertainty. Therefore, one of the prerequisites for the evaluation of uncertainty on the radiation correction is to "know" the irradiance in soundings. Simulation of irradiation is not fully SI-traceable and costs a large uncertainty. This work aims at improving this issue by measuring the irradiance using dual thermistors. Because of the unknown irradiance and its uncertainty, the uncertainty of RS41 specified by the manufacturer should be used for the comparison with the DTR. Then, the combined uncertainty of the RS41 (0.4 °C) and the DTR (0.33−35 °C) is 0.52−0.53 °C at 16 km and higher up and thus the observed differences between the RS41 and the DTR are within their combined uncertainty at daytime. These are explained in the revised manuscript.

**Line 418-434: (Before)** However, the difference at daytime gradually increases with height above 15 km and becomes greater than the DTR uncertainty at 30 km. This implies that the radiation-corrected temperature of DTR is slightly higher than that of RS41 on average at daytime; however, the uncertainty of the RS41 radiosonde is not considered here. A similar trend was observed during the radiation correction of the RS41 radiosonde by GRUAN using the SISTER setup (Von Rohden *et al.*, 2021). The radiation-correction temperature of the RS41 radiosonde obtained by GRUAN is higher than that provided by Vaisala at daytime.

→ **(after)** The manufacturer specifies that the uncertainty of RS41 is 0.3 °C in 0−16 km in altitude and 0.4 °C above 16 km (Vaisala). Then, the combined uncertainty of the RS41 (0.4 °C) and the DTR (0.33−35 °C) is 0.52−0.53 °C ($k = 2$) at 16 km and higher up. Thus, the observed differences between the RS41 and the DTR are within their combined uncertainty at daytime. Nevertheless, the radiation-corrected temperature of DTR is about 0.4 °C higher than that of RS41 around 30 km at daytime. A similar trend is observed in the radiation correction of the RS41 radiosonde by the GRUAN using the SISTER setup (Von Rohden et al., 2022). The

radiation-corrected temperature of the RS41 obtained by the GRUAN is 0.35 °C warmer0. than that provided by Vaisala at 35 km although the difference of temperature between the GRUAN and Vaisala is within their combined uncertainty.

Recently, we have obtained a radiation correction formula of RS41 under a well-defined irradiance in the UAS (Lee et al., 2021). However, the correction formula cannot be applied to RS41 because the irradiance and its uncertainty in soundings are unknown. In this regard, the GRUAN uses a simulated irradiance calculated by the average of clear and cloudy sky cases for their radiation correction of RS41 (Von Rohden et al., 2022). The maximum uncertainty of RS41 by the GRUAN is about 0.3 °C at $k = 2$ which is larger than our previous work on RS41 (0.17 °C at $k = 2$). This is because the irradiance in our work is assumed to be 1360 W·m$^{-2}$ at stratosphere with a small uncertainty obtained by the laboratory experiments corresponding to the irradiance. Therefore, one of the prerequisites to the uncertainty evaluation on the radiation correction is to know the irradiance and its uncertainty in soundings. This work may contribute to improving the measurement of the irradiance and the estimation of its uncertainty using dual thermistor radiosondes.

**Line 366:** replace with "discussed"

**Line 447:** demonstrated → discussed

**Line 369-370:** Without considering, the RS41 uncertainty this statement is not valid.

**Line 450-452: (Before)** The corrected temperature of DTR was mostly consistent with that of RS41 within the expanded DTR uncertainty. → **(After)** The corrected temperature of the DTR was about 0.4 °C higher than that of RS41 around 30 km at daytime although the difference is within the combined uncertainty (~0.5 °C at $k = 2$) of the RS41 and the DTR.

**Line 371:** also more parallel soundings. This is a very important aspect to mention here.

We are preparing for more parallel soundings this year to provide more decisive and diverse results of the DTR. Besides the nighttime correction, one of our interests is how DTR would respond while/after passing through clouds.

**Line 453-455 (Before)** Future works may include more sounding tests in various conditions including cloudy and windy weather to better characterise the DTR performance of in-situ radiation measurements and corrections. → **(After)** Future works may include more parallel sounding tests in various conditions including cloudy and windy weather to better characterise the performance of the DTR. Especially, the radiation correction of the DTR is expected to be different from others while/after passing through clouds because the DTR responds to an *in-situ* radiation flux.

**Line 475:** Caption must report also details to understand the content of the right plot.

**Line 572-574 (Added statement):** Line The temperature difference between the dual thermistors ($T_{\text{B\_raw}} - T_{\text{W\_raw}}$) is linearly proportional to the irradiance, and the radiation-induced heating of the white sensor ($T_{\text{W\_raw}} - T_{\text{W\_cor}}$) is corrected based on the irradiance measured by ($T_{\text{B\_raw}} - T_{\text{W\_raw}}$).

---

## Author Comment (AC3)

**Referee #2**

**General Comment:** The manuscript by Lee et al. provides the description and the metrological characterisation of a dual thermistor radiosonde (DTR) comprising two sensors with different emissivities. The work described in the manuscript is done and continuity with previous efforts of scientific community and of the authors themselves. The use of DTR has the main objective to improve the adjustment of the daytime solar radiation effect on the temperature sensors which is a critical issue for all radiosonde types.

The manuscript is well structured and, beyond the introduction, sufficiently well written. The metrological characterisation of the DTR is detailed and comprises of the major steps needed to fully characterise investigated sensor. Nevertheless I have major concerns about the scientific quality of the manuscript which are detailed below.

There is a strong imbalance between the metrological characterisation of the DTR and the scientific discussion related to the assessment of the DTR performance in the atmosphere.

 $\rightarrow$  We thank the Reviewer for valuable comments. We have added more discussions on the performance of the DTR to address the Reviewer's comments. The specific changes are made as below.

**1.** The metrological characterisation of the solar radiation correction and the quantification of the uncertainty budget for the DTR are not supported by sufficient validation of results: the comparison with the RS41, which can be assumed in the context of the manuscript as a "community standard" (although the only reference is known to be the cryogenic frostpoint hygrometer), is quickly presented and lacks of discussion for the differences shown in Fig. 9. This a very important aspect to show the performances of the DTR measurements. A broader discussion on the consistency with the RS41 should be included.

→ More discussion is added as responses to this comment and the later comment in Section 6.3. Please see the response to the comment of Line 349-353 in original manuscript or Line 416-432 in revised manuscript.

2. The experimental set up discussed in the manuscript looks sufficiently robust, although this could be compared with previous experiments available in literature in order to show what are the pros and cons of the metrological characterization carried out by the authors. A few assumptions in the manuscript must be well justified by means of references or quantitative results. For example, when the term  $(T_{B_raw} - T_{W_raw})$  is used instead of  $(T_{B_on} - T_{W_on})_{UAS}$  to obtain the irradiance using Eq. (10) in the adjustement of the DTR measured profile, this is done without justifying their effect on the measurements compared to the results obtained in the laboratory experiments. Also the uncertainty due to the solar radiation correction is smaller

than for other radiosondes (according to the literature) and a comparative discussion could be provided in the benefit of the reader.

→ Although both  $(T_{B_raw} - T_{W_raw})$  and  $(T_{B_on} - T_{W_on})_{UAS}$  are linearly proportional to the total radiation flux, the portion of the long-wave and the short-wave radiation may affect them because the emissivity and absorptivity of thermistors are dependent on the radiation wavelength. For aluminium coating, the reflectance is 0.8–0.9 below 1000 nm in wavelength and 0.9 above 1000 nm. This implies that the influence of the portion of the long- and shortwave radiation would be a few percent of the radiative heating of the DTR even when the portion below 1000 nm is drastically different between the laboratory experiments and soundings.

Added statement (Line 328-335): The radiation correction formula of the DTR is obtained based on the portion of the long-wave and the short-wave radiation from the solar simulator used as a radiation source in the UAS experiments. The emissivity and absorptivity are dependent on the wavelength. In this regard, the radiative heating of the DTR in soundings can be affected by the actual ratio of the long-wave and the short-wave radiation. For aluminium coating, the reflectance was 0.8–0.9 below 1000 nm and 0.9 above 1000 nm in wavelength. This means that the influence of the ratio between the long- and short-wave radiation would be a few percent of the radiative heating of the DTR even when the portion below 1000 nm is drastically different between the laboratory experiments and soundings.

 $\rightarrow$  The uncertainty due to the solar radiation correction is smaller than other radiosondes because the DTR measures the effective irradiance while others use the simulated irradiance constructed from the average of clear and cloudy sky cases. This results in the increase of the radiation correction uncertainty in troposphere.

Line 72-74: (Before) the SI traceability of the simulated irradiance is incomplete and may increase the radiation correction uncertainty  $\rightarrow$  (After) the SI traceability of the simulated irradiance is incomplete if the sky is clear or cloudy because the simulated irradiance is constructed from the average of clear and cloudy sky cases (Von Rohden et al., 2022). This results in the increase of the radiation correction uncertainty in troposphere.

**3.** The manuscript introduction is not well written and lacks of fairness and accuracy. I strongly recommend the authors to pay great care in writing inaccurate or wrong statements. Specific comments on the introduction are provided in a revised pdf version of the manuscript along with several specific comments, attached to this review.

 $\rightarrow$  The specific comments in the pdf version are addressed as below.

Line 10: probes?

**Line 10:** sensors  $\rightarrow$  probes

Line 12: radiosounding

**Line 12:** sounding  $\rightarrow$  radiosounding

Line 26: If you refer to Bojiinski please use the term "Essential Climate Variables". Line 30: environmental parameters → essential climate variables

Line 29: repetition

Line 33: environmental parameters  $\rightarrow$  meteorological parameters

Line 29: collected measuements

Line 33: measurement  $\rightarrow$  collected measurement

**Line 30-31:** This sentence is quite generic, should refer to many other aspect like time integration, representativeness, different atmospehric regions. I recommend to remove.

**Removed sentence:** Radiosonde measurements are known to be more accurate than measurement methods based on remote sensing techniques, such as LIDAR and satellite.

Line 32: "for calibrating" or validating. This paragraph is a bit uncorrect. Must be rephrased.

Line 34-36: (Before) Thus, radiosonde measurements are often used as reference to correct other measurement data.  $\rightarrow$  (After) Radiosonde observations are often co-located with global navigation satellite system radio occultation and used as reference for validating their one-dimensional interpolation which follows the flight trajectories of balloon soundings.

Line 32: This is inpendent on the needs ot callibrating /validating other techniques.

Removed word: Hence,

**Line 37-38:** if GCOS is referring to the "meteorolgical community" this is evidently wrong because this is a climate initiative by WMO

**Removed phrase:**, such as the Global Climate Observing System (GCOS)

Line 39: Replace with "in the upper troposphere/lower stratosphere".

Line 42-43: (Before) when the project deliverables were limited to the field of upper air measurement by radiosondes  $\rightarrow$  (After) in the environments imitating the upper troposphere/lower stratosphere

Line 40: investigate or detect.

**Line 43:** resolve  $\rightarrow$  investigate

Line 40: remove

Removed phrase: long term

Line 40-41: unclear, rephase

Line 43-44: (Before) a certain amount of measurement uncertainty of radiosonde sensors is required for actual soundings in addition to the ground-based facilities  $\rightarrow$  (After) a certain level of measurement uncertainty in radiosoundings should be secured.

Line 42-43: replace with "a dataset of traceable measurements with quantified uncertainties"

Line 45-46: (Before) the highest level of quality control in the upper air measurement in terms of uncertainty  $\rightarrow$  (After) a dataset of traceable measurements with quantified uncertainties

Line 43-44: Please refer in the numbers and citation to Dirksen et al 2014

Line 55-57 (Added statement): The uncertainty of the GDP of RS92 during daytime was gradually increased from 0.2 °C at the surface to 0.6 °C at 30 km with the coverage factor k = 2 (Dirksen et al., 2014).

**Line 59-61 (Added statement):** Using the setup, the uncertainty of the GDP of RS41 is evaluated to be about 0.3 °C (k = 2) at 35 km. It is also found that the daytime GRUAN profile is 0.35 °C warmer than the manufacturer's at 35 km (Von Rohden et al., 2022).

Line 45: replace with "adjustment" or "correction".

Line 48: compensation  $\rightarrow$  correction

Line 50-51: this sentence must be rephrases to say way otherwise remove, the following sentence is sufficient.

**Removed sentence:** Thus, the uncertainty evaluation on the radiation correction by manufacturers cannot be fully accepted.

Line 54: remove

Removed word: algorithm

Line 61: remove this is already clear from the context

Removed phrase: long term

Line 64: replace with "measuring on the same payload of radiosondes"

Line 69-70: during soundings  $\rightarrow$  measuring on the same payload of radiosondes

Line 67-68: one or more references here may help

Line 72-74: (Before) the SI traceability of the simulated irradiance is incomplete and may increase the radiation correction uncertainty  $\rightarrow$  (After) the SI traceability of the simulated irradiance is incomplete if the sky is clear or cloudy because the simulated irradiance is constructed from the average of clear and cloudy sky cases (Von Rohden et al., 2022). This results in the increase of the radiation correction uncertainty in troposphere.

Line 71: put a comma

Line 77-78: air ventilation and temperature  $\rightarrow$  air ventilation, and temperature

Line 73: replace with "showed to be"

**Line 79:** was  $\rightarrow$  showed to be

Line 79: remove

**Removed word:** formulas

Line 83: be consistent in using the tense of verbs

 $\rightarrow$  Present tense is used in general unless conducted earlier than this work.

Line 85-86: (Before) The obtained formulas were used in 80 an intercomparison sounding test that was performed in July, 2021.  $\rightarrow$  (After) The obtained formulas are used in an intercomparison sounding test performed in July, 2021.

Line 85: add "from parallel soundings"

Line 91 (Added phrase): from parallel soundings

Line 85: repeated verb

Line 90-92: (Before) Finally, the difference between the corrected temperatures of DTR and the RS41 radiosonde is presented with the DTR uncertainty  $\rightarrow$  (After) Finally, the corrected temperatures of the DTR and the RS41 from parallel soundings are compared and the difference between them is discussed in terms of the uncertainty.

Line 106-107: May be this sentence could be removed, need too many details which are discussed later in the text.

**Removed sentence:** Then, the effective irradiance and other environmental factors are parameterised into a single equation to calculate the radiation correction of the white sensor  $(T_{W_raw} - T_{W_cor})$ .

Line 131-132: It would be interesting to show the difference between the second-order and third order polynomial. I recomment to add these results to the mansuscript.

 $\rightarrow$  A separate paper from our group which clearly shows the difference is added as a reference.

Line 145: (not shown here)  $\rightarrow$  (Yang et al., 2021)

Line 141: use a different way to indicate the thermistors to avoid confusion with the other sensors in the chamber.

Line 153 & 155: sensors  $\rightarrow$  sensors (thermistors)

**Line 156: why k=1 and not k=2? Please clarify**

 $\rightarrow$  The probability distribution is assumed to be a normal distribution function and the standard deviation is used for standard uncertainty (k = 1).

Line 175-178: (Before) Since the temperature error is roughly distributed within  $\pm 0.05$  °C, the standard deviation of all data points is obtained (0.04 °C) and is used for uncertainty (k = 1) due to the influence of temperature on the resistance reading by radiosonde boards.  $\rightarrow$  (After) Assuming that the probability distribution is a normal distribution function, the standard deviation (SD) of all data points (0.04 °C) is used for standard uncertainty due to the influence of the temperature of radiosonde electronics boards on the resistance (or temperature) measurement.

**Line 158:** not 100% clear, please expand the sentence. The following sentence clarifies but the reading of this unit is not fully clear.

Line 180-183: (Before) A difficulty faced during the radiation correction of these thermistors is the unit difference in terms of the correction value.  $\rightarrow$  (After) The purpose of the calibration of thermistors and the investigation of the temperature effect on radiosonde electronics boards is to assess the accuracy (or uncertainty) of raw temperature measurement before radiation correction. The next step is to investigate the sensitivity of individual thermistors to irradiation because the amount of radiation correction varies for individual radiosondes, presumably related to the production process of the thermistors.

**Line 176:** The count distributions shown in Figure 5 are not discussed here at all: no statistical indicators are reported, commented, for example nothing is said why the distribution in Figure 5d is skewed while not that in Figure 5c. Are there potential hysteresis or stabilization effects? Then you must describe in the text your setup in comparison to other available in literature and describe pros and cons. As it stands the descritpion is very minimal.

→ Various factors of irregularity in the production process of the thermistors and the environments of the RRT chamber can be responsible for the count distributions in Fig. 5. The size of glass beads is irregular according to the manufacturer specification. The distribution of the size is specified in the revised manuscript. We also suspect the connection part between the sensor leads and the boom for one of the reasons of the observed difference of individual radiation test in Fig. 5(d) because the soldering and the coating of epoxy resin were conducted

manually. The ventilation and air pressure depends on the performance of the vacuum pump and the sealing of the RRT chamber lid using an O-ring. Unfortunately, these factors were not monitored. Thus, there can be variations in the air flow and the pressure that may vary the cooling efficiency.

**Line 96-97: (Before)** ellipsoidal shape with 0.55 mm diameter and 1.1 mm length  $\rightarrow$  (After) ellipsoidal shape with 0.55 ± 0.1 mm diameter and 1.1 ± 0.3 mm length

Line 186-188: (Before) The irregular connection between the sensor and boom may affect the thermal conduction between them.  $\rightarrow$  (After) The connection between the sensor leads and the boom may be irregular because the soldering and the coating of epoxy resin were conducted manually. Radiative heating of glass beads, leads, and connection parts between the sensor leads and the boom should be affected by their size as previously reported (de Podesta et al., 2018).

**Line 210-213 (Added statement):** Although the irradiance is constant for each sensor, the cooling efficiency of the sensors may vary depending on the bead size of thermistors, air flow, and the pressure. Slight variations of air flow and/or pressure in the RRT chamber (not monitored) may partly be responsible for the observed distributions of radiative heating of the sensors in Figs. 5(c) and (d).

**Line 197-201: I suggest to use a generic notation for all the firve equations**

 $\rightarrow$  We would like to use the current form of equations because they are straightforward with coefficients in Table 1.

**Line 211: soundings**

**Line 250:** sounding $\rightarrow$ soundings**

**Line 214: better to use a different way to indicate equation, this one may generate confusion**

→ Eq. (9) and Eq. (10) are simplified to deliver the meaning clearly. The backbone of Eq. (9) and (10) is Eq. (2). Since Eq. (2) is obtained when the radiation flux is 960 W·m-2, an unknown radiation flux can be measured by the temperature difference of two thermistors using the linear relationship with the  $(T_{B_on} - T_{W_on})_{UAS}$  at 960 W·m-2 (the denominator in Eq. (9)). Therefore, the  $(T_{B_on} - T_{W_on})_{UAS}$  in Eq. (9) and (10) should be replaced by  $(T_{B_raw} - T_{W_raw})$  where  $T_{B_raw}$  and  $T_{W_raw}$  are raw temperatures of the black and white sensors, respectively.

**Line 250-252: (Before)** Hence, Eq. (2) is employed to measure the *in-situ* irradiance based on the fact that  $(T_{B_on} - T_{W_on})_{UAS}$  is linearly proportional to S.  $\rightarrow$  (After) Hence, Eq. (2) is employed to measure the *in-situ* irradiance using  $(T_{B_raw} - T_{W_raw})$ , where  $T_{B_raw}$  and  $T_{W_raw}$  are

raw temperatures of the black and white sensors, respectively, based on the fact that the temperature difference between two sensors is linearly proportional to *S*.

Eq. (9): (Before)  $S = S_0 \times (T_{B_{on}} - T_{W_{on}})_{UAS} / [T_0(T_{W_{on}}) + A_0(T_{W_{on}}) \cdot \exp(-P/P_0(T_{W_{on}})) + A_1(T_{W_{on}}) \cdot \exp(-P/P_1(T_{W_{on}}))], S_0 = 960 \text{ W} \cdot \text{m}^{-2}$

 $\rightarrow (\text{After}) S = S_0 \times (T_{\text{B}_{\text{raw}}} - T_{\text{W}_{\text{raw}}}) \cdot (T_{\text{B}_{\text{on}}} - T_{\text{W}_{\text{on}}})_{\text{UAS}}^{-1},$

Eq. (10): (Before)  $S = S_0 \times (T_{B_on} - T_{W_on})_{UAS} / [T_0(T_{W_on}) + A_0(T_{W_on}) \cdot \exp(-P/P_0(T_{W_on})) + A_1(T_{W_on}) \cdot \exp(-P/P_1(T_{W_on})) - 0.08 \cdot (v - v_0)], S_0 = 960 \text{ W} \cdot \text{m}^{-2} \text{ and } v_0 = 5 \text{ m} \cdot \text{s}^{-1}$

→ (After)  $S = S_0 \times (T_{B_{raw}} - T_{W_{raw}}) \cdot [(T_{B_{on}} - T_{W_{on}})_{UAS} - 0.08 \cdot (v - v_0)]^{-1}$ ,

Line 216-217: redundant, remove

**Removed phrase:** by Eq. (9) to estimate the irradiance (960 W·m-2) by using  $(T_{B_on} - T_{W_on})_{UAS}$ ,  $(T_{B_on} - T_{W_on})_{RRT}$  and other environmental parameters

**Line 222-223:** Do the residuals reduce after including the effect of air ventilation? Please provide details and maybe amend the plot in Figure 6.

→ The ventilation speed for five units in Figure 6 is fixed at 5 m·s-1 and thus the effect of air ventilation cannot be identified. The effect of air ventilation is studied with a separate pair of thermistors. The standard deviation of the residual for the pair of thermistors is 4.1% with Eq. (9) while it is reduced to 3.4% with Eq. (10) when the air ventilation is actually changed (4– $6.5 \text{ m} \cdot \text{s}^{-1}$ ).

**Line 258-259 (Added statement):** and thus the effect of air ventilation cannot be identified in Fig. 6(f).

Line 259 (Added statement): by a separate pair of thermistors

**Line 263-264 (Added statement):** The standard deviation of the residual for a pair of thermistors is 4.1% with Eq. (9) while it is reduced to 3.4% with Eq. (10) when the air ventilation is actually changed  $(4-6.5 \text{ m} \cdot \text{s}^{-1})$ .

**Line 245: same comment as above**

 $\rightarrow$  We would like to use the current form of equations because they are straightforward with coefficients in Table 2.

Line 265: Same comment as at line 222 and not clear why residuals are not evaluated after the inclusion of wind speed. Please motivate.

→ As mentioned, the effect of air ventilation is studied with a separate pair of thermistors. The ventilation speed for five units in Figure 7 is fixed at 5 m·s-1 and thus the effect of air ventilation cannot be identified. The standard deviation of the residual for a pair of thermistors is 0.10 °C with Eq. (18) while it is reduced to 0.04 °C with Eq. (19) when the air ventilation is actually changed (4–6.5 m·s-1).

**Line 306-307 (Added statement):** and thus the effect of air ventilation cannot be identified in Fig. 7(f).**

Line 307 (Added statement): by a separate pair of thermistors

**Line 311-314 (Added statement):** When the air ventilation is actually changed (4–6.5 m·s-1), the standard deviation of the residual for a pair of thermistors is 0.10 °C with Eq. (18) while it is reduced to 0.04 °C with Eq. (19).

Line 272: Please say ho many sounding were performed during the intercomparison.

 $\rightarrow$  More information on soundings is added.

Line 319-322 (Added statement): One, two, or three DTRs were tested in parallel with a RS41 in a single flight. The number of comparison (N) was N = 12 at daytime and N = 6 at nighttime from 7 and 3 soundings, respectively. The daytime sounding was performed from 11:00 am to 5 pm local time while the nighttime sounding was from 12:00 am to 4 am. The sky was normally cloudy.

**Line 274-275:** What's the effect of this assumption compared to the lab experiements? Do you have results to show? Otherwise, please comment.

→ → Although both  $(T_{B_raw} - T_{W_raw})$  and  $(T_{B_on} - T_{W_on})_{UAS}$  are linearly proportional to the total radiation flux, the portion of the long-wave and the short-wave radiation may affect them because the emissivity and absorptivity of thermistors are dependent on the radiation wavelength. For aluminum coating, the reflectance is 0.8–0.9 below 1000 nm in wavelength and 0.9 above 1000 nm. This implies that the influence of the portion of the long- and shortwave radiation would be a few percent of the radiative heating of the DTR even when the portion below 1000 nm is drastically different between the laboratory experiments and soundings.

Added statement (Line 328-335): The radiation correction formula of the DTR is obtained based on the portion of the long-wave and the short-wave radiation from the solar simulator used as a radiation source in the UAS experiments. The emissivity and absorptivity are dependent on the wavelength. In this regard, the radiative heating of the DTR in soundings can be affected by the actual ratio of the long-wave and the short-wave radiation. For aluminium coating, the reflectance was 0.8–0.9 below 1000 nm and 0.9 above 1000 nm in wavelength. This means that the influence of the ratio between the long- and short-wave radiation would be

a few percent of the radiative heating of the DTR even when the portion below 1000 nm is drastically different between the laboratory experiments and soundings.

**Line 275:** Please report the time of the launch and more details on the payload configuration. These are important elements to evaluate the quality of the collected profiles.

 $\rightarrow$  More information on soundings is added in the earlier comment.

**Line 289-290:** Should this correction, event to a certain exent, be needed also during daytime? Please comment in clear way.**

→ The negative irradiance in the stratosphere at nighttime is very clear because the temperature of the black sensor is distinctively lower than the white sensor above 20 km at nighttime (Fig. 8d). Previously, the same phenomenon was observed in the Figure 4b of Rolf Philipona *et al.* (2013) in Journal of Atmospheric and Oceanic Technology (DOI: 10.1175/JTECH-D-13-00047.1). The authors explained that "The sum of the absorbed and emitted fluxes result in the longwave radiation balance of the sensor during the night (LRB\_n), which is negative in the lower troposphere, and then becomes positive and again negative further up in the stratosphere. Hence, in the lower troposphere, LRB\_n cools the temperature sensor, higher up LRB\_n warms the temperature sensor, and above 25 km it again cools the temperature sensor." The effect of long-wave radiation was also applied to the daytime as they mentioned that "The longwave radiative impact on the sensor from above and from below is very similar during the day and during the night (Fig. 4c)." Indeed, the pattern of LRB\_d at daytime shown in Fig. 4c of Philipona *et al.* is similar to that of the LRB\_n at nighttime in Fig. 4b. Therefore, the decrease of the effective irradiance in the stratosphere at daytime is highly likely due to the negative longwave radiation balance of the sensors.

Added statement (Line 346-356): The negative net irradiance at nighttime was also observed in the previous work for the radiation correction of radiosondes based on the measurement of radiative flux profiles using two pyranometers for measuring downward and upward solar short-wave radiation, and two pyrgeometers for measuring upward and downward thermal long-wave radiation (Philipona et al., 2013). The long-wave radiation balance (LRB) of the sensor defined by the sum of the absorbed and emitted fluxes corresponds to the effective irradiance at nighttime in this work. Both the LRB in the work of Philipona *et al.* and the effective irradiance at nighttime in this work are negative in the lower troposphere, and then become positive and again negative further up in the stratosphere. This means that temperature sensors of radiosondes are cooled in lower troposphere, warmed in higher up, and again cooled further up in the stratosphere and thus should be corrected accordingly. The profile of the LRB at nighttime was similar to that of daytime (Philipona et al., 2013). In this regard, the decrease of the effective irradiance in the stratosphere observed at daytime is highly likely due to the negative LRB of the sensors as observed in the nighttime soundings. **Line 306:** The nighttime residual adjustment in the upper atmosphere due to long wave emission of the black thermistor at night is estimated on 6 soundings only. This increases the DTR uncertainty. And moreover, please, provide results above significance of the DTR at night time.

→ We agree that the data is not enough in this work and thus we are preparing for more intercomparison soundings this year to provide more decisive and diverse results of the DTR. As mentioned in the earlier comment, the long-wave radiation imbalance (cooling) of sensors at stratosphere should not be the issue of the DTR only because the same phenomenon was observed in the previous work based on the measurement of long-wave radiations. These findings suggest that an application of radiation correction is needed even at nighttime (RS41 applies no radiation correction at nighttime).

Added statement (Line 374-376): The long-wave radiation imbalance (cooling) of sensors at stratosphere should not be the issue of the DTR only because the same phenomenon was observed in the previous work based on the measurement of long-wave radiations. These findings suggest that an application of radiation correction is needed even at nighttime.

Line 319: In the formula a variance is reported, make the text and the formulas coherent.

Line 398-400: (Before) DTR uncertainty gradually increases up to about 0.35 °C in the troposphere and is maintained in the stratosphere. However, at nighttime, the uncertainty slightly is decreased to about 0.3 °C as the uncertainty of effective irradiance is decreased.  $\rightarrow$  (After) DTR uncertainty gradually increases up to about 0.35 °C at the tropopause and is maintained in the stratosphere (0.33 °C at 30 km). However, at nighttime, the uncertainty slightly is decreased to 0.3 °C at the tropopause and 0.25 °C at 30 km as the uncertainty of effective irradiance is decreased.

Line 338: Again, sometimes the authors uses k=1, some other k=2, please ensure consistency.

 $\rightarrow$  The expanded uncertainty (*k* = 2) is used throughout manuscript.

Line 346: Again, how many soundings? This is a crucial element for the radiosonde intercomparison.

 $\rightarrow$  More information on soundings is added in the earlier comment.

**Line 349-353:** Comments in this section are quite faint and not extensive as they should be. This section should indeed containt a lot of elements to demostrate the performance of you dual thermistor sonde in comparison with the most popular sensor on the market. A deeper analysis and comments should be done with a spirit similar to the discussion provided by Dirksent et al., 2014 when comparison RS92 GDP with CFH, assuming obvioulsy the latter as the reference.

Moreover, although requiring additional work, the authors have the unique opportunity to compare two radiosondes types with the related uncertainties, so they should do their best to include RS41 uncertainty.

 $\rightarrow$  The manufacturer specifies that the uncertainty of RS41 is 0–0.3 °C in 0–16 km and 0.4 °C at 16 km and higher up (Vaisala, white paper). However, the detailed process to obtain the uncertainty such as laboratory experiments is not disclosed. Recently, we have obtained a radiation correction formula of RS41 under a well-defined irradiance in the UAS (Lee et al. Atmos. Meas. Tech., in print). However, the correction formula cannot be applied to RS41 because the manufacturer does not provide raw temperature in soundings. Even though raw temperature would be provided, the correction formula cannot be applied because the irradiance used by RS41 is unknown. In this regard, the GRUAN uses their own simulated irradiance constructed from the average of clear and cloudy sky cases in their recent publication on RS41 (von Rohden et al., Atmos. Meas. Tech., 2022). The maximum uncertainty by the GRUAN is about 0.3 °C at k = 2 which is larger than our previous work (0.17 °C at k = 2) because the irradiance in our work is assumed to be 1360  $W \cdot m^{-2}$  at stratosphere with a small (lab scale) uncertainty. Therefore, one of the prerequisites for the evaluation of uncertainty on the radiation correction is to "know" the irradiance in soundings. Simulation of irradiation is not fully SI-traceable and costs a large uncertainty. This work aims at improving this issue by measuring the irradiance using dual thermistors. Because of the unknown irradiance and its uncertainty, the uncertainty of RS41 specified by the manufacturer should be used for the comparison with the DTR. Then, the combined uncertainty of the RS41 (0.4 °C) and the DTR (0.33–35 °C) is 0.52–0.53 °C at 16 km and higher up and thus the observed differences between the RS41 and the DTR are within their combined uncertainty at daytime. These are explained in the revised manuscript.

**Line 418-434: (Before)** However, the difference at daytime gradually increases with height above 15 km and becomes greater than the DTR uncertainty at 30 km. This implies that the radiation-corrected temperature of DTR is slightly higher than that of RS41 on average at daytime; however, the uncertainty of the RS41 radiosonde is not considered here. A similar trend was observed during the radiation correction of the RS41 radiosonde by GRUAN using the SISTER setup (Von Rohden *et al.*, 2021). The radiation-correction temperature of the RS41 radiosonde obtained by GRUAN is higher than that provided by Vaisala at daytime.

→ (after) The manufacturer specifies that the uncertainty of RS41 is 0.3 °C in 0–16 km in altitude and 0.4 °C above 16 km (Vaisala). Then, the combined uncertainty of the RS41 (0.4 °C) and the DTR (0.33–35 °C) is 0.52–0.53 °C (k = 2) at 16 km and higher up. Thus, the observed differences between the RS41 and the DTR are within their combined uncertainty at daytime. Nevertheless, the radiation-corrected temperature of DTR is about 0.4 °C higher than that of RS41 around 30 km at daytime. A similar trend is observed in the radiation correction of the RS41 radiosonde by the GRUAN using the SISTER setup (Von Rohden et al., 2022). The

radiation-corrected temperature of the RS41 obtained by the GRUAN is 0.35 °C warmer0. than that provided by Vaisala at 35 km although the difference of temperature between the GRUAN and Vaisala is within their combined uncertainty.

Recently, we have obtained a radiation correction formula of RS41 under a well-defined irradiance in the UAS (Lee et al., 2021). However, the correction formula cannot be applied to RS41 because the irradiance and its uncertainty in soundings are unknown. In this regard, the GRUAN uses a simulated irradiance calculated by the average of clear and cloudy sky cases for their radiation correction of RS41 (Von Rohden et al., 2022). The maximum uncertainty of RS41 by the GRUAN is about 0.3 °C at k = 2 which is larger than our previous work on RS41 (0.17 °C at k = 2). This is because the irradiance in our work is assumed to be 1360 W·m-2 at stratosphere with a small uncertainty obtained by the laboratory experiments corresponding to the irradiance. Therefore, one of the prerequisites to the uncertainty evaluation on the radiation correction is to know the irradiance and its uncertainty in soundings. This work may contribute to improving the measurement of the irradiance and the estimation of its uncertainty using dual thermistor radiosondes.

Line 366: replace with "discussed"

Line 447: demonstrated  $\rightarrow$  discussed

Line 369-370: Without considering, the RS41 uncertainty this statement is not valid.

**Line 450-452: (Before)** The corrected temperature of DTR was mostly consistent with that of RS41 within the expanded DTR uncertainty.  $\rightarrow$  (After) The corrected temperature of the DTR was about 0.4 °C higher than that of RS41 around 30 km at daytime although the difference is within the combined uncertainty (~0.5 °C at k = 2) of the RS41 and the DTR.

Line 371: also more parallel soundings. This is a very important aspect to mention here.

We are preparing for more parallel soundings this year to provide more decisive and diverse results of the DTR. Besides the nighttime correction, one of our interests is how DTR would respond while/after passing through clouds.

Line 453-455 (Before) Future works may include more sounding tests in various conditions including cloudy and windy weather to better characterise the DTR performance of in-situ radiation measurements and corrections.  $\rightarrow$  (After) Future works may include more parallel sounding tests in various conditions including cloudy and windy weather to better characterise the performance of the DTR. Especially, the radiation correction of the DTR is expected to be different from others while/after passing through clouds because the DTR responds to an *in-situ* radiation flux.

Line 475: Caption must report also details to understand the content of the right plot.

**Line 572-574 (Added statement):** Line The temperature difference between the dual thermistors ( $T_{B_raw} - T_{W_raw}$ ) is linearly proportional to the irradiance, and the radiation-induced heating of the white sensor ( $T_{W_raw} - T_{W_cor}$ ) is corrected based on the irradiance measured by ( $T_{B_raw} - T_{W_raw}$ ).

---

## Author Comment (AC4)

**Referee #3**

Review of manuscript "Laboratory characterisations and intercomparison sounding test of dual thermistor radiosondes for radiation correction" by Sang-Wook Lee, et al., AMT-2021-343

**Synopsis:** The authors describe a detailed metrological characterization of a dual thermistor radiosonde and how the difference between thermistors can be used for the radiation error affecting both. The result is a quite simple multilinear model for correction, based just on the differences between thermistors. The characterization methods appear quite solid. Only the comparison with the de facto standard (RS41) has been performed using too small samples.

→ We thank the Reviewer for providing valuable comments. We have addressed his/her comments as below.

**Major comments:**

I was a bit surprised to learn that the temperature was varied only between -70 and 20 deg, since in the tropics and over Antarctica temperatures below -90 degrees are not uncommon. This seems important since in Fig. 4, there are substantial variations of resistance reading at -70 deg. Does the instrument fail at even lower temperatures? Perhaps this is a wrong impression because of the y axis being linear, not logarithmic.

→ The lowest limit of the temperature of the climate chamber used for the calibration of thermistors in Figure 4 is -75 °C. This is the typical temperature limit of commercially-available climate chambers. We know the importance of the sensor calibration down to -90 °C to measure the temperature of upper air globally. Unfortunately, it is not feasible in our system at the moment. We have mentioned this point clearly in the revised manuscript.

The resistance of the negative temperature coefficient (NTC) thermistors used in this work is changed from 10 kΩ to 700 kΩ when the temperature is varied from 20 °C to -70 °C, respectively, as shown in the x-axis of Fig. 4(b). Although the absolute difference between the reference and radiosonde reading is accordingly increased at -70 °C, the residual of converted temperature is not increased as shown in Fig. 4(c). The y-axis cannot be changed into logarithmic scale because some of the data is negative.

**Added statement (Line 140-142):** Although the calibration range should be extended to −90 °C to cover temperatures over tropic and polar regions, it is not feasible using the climate chamber because the typical lowest temperature limit is more or less −80 °C.

The comparison with the current de fact standard RS41 should be more comprehensive. In Fig. 9 it is not clear how many radiosondes were launched in parallel. This is very important to have a robust estimate of differences. Somewhere in the text it is written N=12 for daytime and N=6

for nighttime. That should be also in the caption. The same applies to the sounding test described in section 8. It appears it was only one ascent?

→ More information on soundings is added.

**Added statement (Line 319-322):** One, two, or three DTRs were tested in parallel with a RS41 in a single flight. The number of comparison ($N$) was $N = 12$ at daytime and $N = 6$ at nighttime from 7 and 3 soundings, respectively. The daytime sounding was performed from 11:00 am to 5 pm local time while the nighttime sounding was from 12:00 am to 4 am. The sky was normally cloudy.

There is a lot of redundancy in the formulae. Why do you specify S_0 = 960 W/m^2 all the time in formulae 2-19. It is given in the text and does never change. The same is true for v_0. Personally I would also recommend writing fractions as with – as divisor, not / in numbered formulae.

→ Equations are simplified, for example, by removing the redundancy such as $S_0$ and $v_0$. In the fractions, '$^{-1}$' is used instead of '/' as suggested.

**Before:** $(T_{B\_on} - T_{W\_on})_{UAS} = T_0(T_{W\_on}) + A_0(T_{W\_on}) \cdot \exp(-P/P_0(T_{W\_on})) + A_1(T_{W\_on}) \cdot \exp(-P/P_1(T_{W\_on}))$, $S_0 = 960 \text{ W} \cdot \text{m}^{-2}$ ,

**After (Eq. (2)):** $(T_{B\_on} - T_{W\_on})_{UAS} = T_0(T_{W\_on}) + A_0(T_{W\_on}) \cdot \exp(-P \cdot P_0(T_{W\_on})^{-1}) + A_1(T_{W\_on}) \cdot \exp(-P \cdot P_1(T_{W\_on})^{-1})$,

**Before:** $S = S_0 \times (T_{B\_on} - T_{W\_on})_{UAS} / [T_0(T_{W\_on}) + A_0(T_{W\_on}) \cdot \exp(-P/P_0(T_{W\_on})) + A_1(T_{W\_on}) \cdot \exp(-P/P_1(T_{W\_on}))]$ , $S_0 = 960 \text{ W} \cdot \text{m}^{-2}$

**After (Eq. (9)):** $S = S_0 \times (T_{B\_raw} - T_{W\_raw}) \cdot (T_{B\_on} - T_{W\_on})_{UAS}^{-1}$ ,

**Before:** $S = S_0 \times (T_{B\_on} - T_{W\_on})_{UAS} / [T_0(T_{W\_on}) + A_0(T_{W\_on}) \cdot \exp(-P/P_0(T_{W\_on})) + A_1(T_{W\_on}) \cdot \exp(-P/P_1(T_{W\_on})) - 0.08 \cdot (v - v_0)]$, $S_0 = 960 \text{ W} \cdot \text{m}^{-2}$ and $v_0 = 5 \text{ m} \cdot \text{s}^{-1}$

**After (Eq. (10)):** $S = S_0 \times (T_{B\_raw} - T_{W\_raw}) \cdot [(T_{B\_on} - T_{W\_on})_{UAS} - 0.08 \cdot (v - v_0)]^{-1}$,

**Before:** $(T_{W\_on} - T_{W\_off})_{UAS} = T_1(T_{W\_on}) + A_2(T_{W\_on}) \cdot \exp(-P/P_2(T_{W\_on})) + A_3(T_{W\_on}) \cdot \exp(-P/P_3(T_{W\_on}))$, $S_0 = 960 \text{ W} \cdot \text{m}^{-2}$ ,

**After (Eq. (11)):** $(T_{W\_on} - T_{W\_off})_{UAS} = T_1(T_{W\_on}) + A_2(T_{W\_on}) \cdot \exp(-P \cdot P_2(T_{W\_on})^{-1}) + A_3(T_{W\_on}) \cdot \exp(-P \cdot P_3(T_{W\_on})^{-1})$,

**Before:** $(T_{W\_on} - T_{W\_off})_{UAS} = (S/S_0) \times [T_1(T_{W\_on}) + A_2(T_{W\_on}) \cdot \exp(-P/P_2(T_{W\_on})) + A_3(T_{W\_on}) \cdot \exp(-P/P_3(T_{W\_on}))]$, $S_0 = 960 \text{ W} \cdot \text{m}^{-2}$,

**After (Eq. (18)):** $(T_{W\_raw} - T_{W\_cor}) = (S \cdot S_0^{-1}) \times (T_{W\_on} - T_{W\_off})_{UAS}$,

**Before:** $(T_{W\_on} - T_{W\_off})_{UAS} = (S/S_0) \times [T_1(T_{W\_on}) + A_2(T_{W\_on}) \cdot \exp(-P/P_2(T_{W\_on})) + A_3(T_{W\_on}) \cdot \exp(-P/P_3(T_{W\_on})) - 0.1 \cdot (v - v_0)]$, $S_0 = 960$ W·m$^{-2}$ and $v_0 = 5$ m·s$^{-1}$ (19).

**After (Eq. (19)):** $(T_{W\_raw} - T_{W\_cor}) = (S \cdot S_0^{-1}) \times [(T_{W\_on} - T_{W\_off})_{UAS} - 0.1 \cdot (v - v_0)]$,

-Formula (25) holds only if errors are independent.

→ Each parameter is controlled independently while others are fixed using the upper air simulator and the corresponding uncertainty is analyzed.

**Minor comments:**

**L69:** different emissivities

→ The word is changed

**Before:** difference emissivities

**After (Line 75-76):** different emissivities

**Fig. 6, 7:** A logarithmic y axis would be very helpful, and is also more suitable to the parameterization you give in Formula (2), which consists of exponential functions.

→ Logarithmic scale is used for Figs. 6 and 7.

**Modified Figures (Figures 6 & 7):** Logarithmic scale is used for x-axis of Figures 6 and 7.

---

## Author Response (AR2)

**Referee #1**

**General Comment:** The authors have addressed all comments and remarks. With the revision, the explanations became clearer and smoother. After consideration of a few further minor comments below, the manuscript should be published.

→ We thank the Reviewer for valuable comments to improve the quality of the manuscript.

**Detailed comments**

**L192**: Explanation of the abbreviation RRT (only emerges in 4.1)

→ The abbreviation RRT is defined in Section 3.4 at its first appearance.

**Before**: Therefore, a radiation test is performed on all thermistors in a vacuum chamber at room temperature

**After (Line 192-193)**: Therefore, a rotational radiation test (RRT) is performed on all thermistors in a vacuum chamber at room temperature

**Before**: the rotational radiation test (RRT) in Fig. 5(c).

**After (Line 220)**: the RRT in Fig. 5(c).

**L205**: '30 data' ◊ '30 data points' (?)

→ 30 data points are used.

**Before**: 30 data

**After (Line 207)**: 30 data points

**L247**: For easier understanding, a sentence should be added here (or at another suitable place) saying that the applied concept of transferring the individual radiation sensitivities from the RRT tests based on the five chosen units to Eq. 2 does not necessarily rely on 'realistic' irradiation and ventilation conditions in the RRT setup, but – due to the proportionality with the UAS results - rather on the consistence of the existing conditions in the RRT over the radiation tests of all other sondes as part of the representativeness of the results from the five thermistor pairs.

→ We agree that the comment is necessary for better understanding of the RRT. The suggested sentence is added in the revised manuscript.

**Added statement (Line 247-251)**: The applied concept of transferring the individual radiation sensitivities from the RRT based on the five chosen units to Eq. (2) does not necessarily rely on 'realistic' irradiation and ventilation conditions in the RRT setup, but rather on the consistence of the existing conditions in the RRT over the radiation tests of all other sondes. The representativeness of the RRT results of the five thermistor pairs as part of all thermistors is based on the proportionality with the UAS results.

**Section 4.2**: The procedure in described section 4.2 is to a wide extent identical to that in section 4.1. With regard to the writing style, I suggest to avoid too much copy-pasting of whole sentences where possible.

→ Section 4.2 is partly rewritten to avoid repeated expressions.

**L281 (maybe already in L234)**: The heat transfer coefficient is not only dependent on the thermal conductivity of air, but also on the viscosity (approximately proportional to T) and density (inversely proportional). Wouldn't it be better to say more generally that the heat transfer therefore correlates positively with temperature overall, i.e. as a net effect?

→ As suggested, the heat transfer is explained more generally.

**Before**: because the convective heat transfer between the sensor and air is reduced as the thermal conductivity of the air is decreased at cold temperatures (Lee et al., 2021).

**After (Line 232-233)**: because the convective heat transfer between the sensor and air is reduced at cold temperatures with positive correlations between the thermal conductivity and the viscosity of air and the air temperature (Lee et al., 2021).

**Before**: This is attributed to the decrease of the thermal conductivity of air at cold temperatures, which reduces the heat transfer from the sensor to air despite the constant irradiation.

**After (Line 284-287)**: The thermal conductivity and the viscosity of air decrease as the air temperature decreases while the density of air is inversely correlated with the temperature. The net effect of these air properties is that the heat transfer from the sensor to air is positively correlated with the air temperature (Lee et al., 2022). The effect of long-wave radiation from the sensor is minor compared with that of convective heat transfer.

**Fig. 6 and Fig. 7**: The units given in the x-axis titles should be hPa (instead of kPa) to be consistent with the axis labels

→ The unit is changed to hPa.

**Modified Figure (Figures 6 & 7)**: The unit of x-axis title is changed to hPa.

**Referee #2**

**General Comment:**

In this second round of the review, the authors have improved the text of the manuscript providing several clarifications and explanations which were missing or not sufficient in the previously submitted version. Nevertheless, also on the basis of the new text added, I think there are still major revisions to apply to the paper and a few major concerns in the presented analysis.

→ We thank the Reviewer for valuable comments to improve the quality of the manuscript.

I report below my major comments.

The authors state that various factors of irregularity in the production process of the thermistors and the environments of the RRT chamber can be responsible for a portion of the total uncertainty. Several possible technical reasons are reported. Despite the large number of tested thermistor It is not clear how this can affect the DTR performances and uncertainties over long term, putting the DTR sonde still at an experimental level. Irregularities in the production process may mean that: 1. the results presented in this paper cannot be used as a general assessment of the dual-thermistor sondes performance. Instead, the presented analysis should be reported as an additional step in the optimization of DTRs and an investigation of what must be improved in the future productions process; 2. this radiosonde type cannot be easily employed on routine basis in a reference network and it's not yet ready to improve the accuracy of temperature measurement in the upper air within the framework of the traceability to the SI. These aspects must be reflected in the paper.

→ As the Reviewer mentioned, this paper is an additional step before an optimization of the DTR. Nevertheless, the paper is meaningful because it shows whole characterization processes of the DTR including the fabrication and the evaluation. This was the first time for us to finish the whole process step by step. We admit that there is much room for the improvement in the fabrication of the DTR, the evaluation using laboratory setups, and comparison soundings. Each process is being polished for an optimization. Very recently, the irregularity of the sensor fabrication is improved and the calibration uncertainty is reduced with minimizing the spatial temperature deviations in the climate chamber. Moreover, we plan to conduct more comparison soundings this year. If there is a major improvement on the DTR or a new finding worth reporting, we will continue to report. This point is added in the revised manuscript.

**Added statement (Line 425-427)**: Future works include an optimization of each process shown in this study such as the fabrication of the DTR and the evaluation using laboratory setups to improve the uncertainties due to irregularities in the production and testing of sensors.

**Before**: Future works may include more parallel sounding tests in various conditions including cloudy and windy weather to better characterise the performance of the DTR.

**After (Line 427-429)**: In addition, more parallel sounding tests in various conditions including daytime and nighttime and/or cloudy and windy weather will be conducted to better characterise the performance of the DTR.

Another question is: can the residuals in the plot of Figure 5 be related to this "production" uncertainty? The authors did not fully answer to my request for clarification about residuals after the first review stage.

→ Figure 5 shows individual radiation sensitivities (not residuals) of pairs of thermistors at a specific condition using the rotational radiation test (RRT) setup. The RRT result is transferred to the radiation correction formula obtained using the upper air simulator (UAS). Since the UAS experiment on a pair of thermistors takes a very long time, only five representative pairs are selected for the UAS testing as shown by arrows in Fig. 5(c) and (d). Then, the individual sensitivity of the RRT is incorporated in the radiation correction formula obtained by the UAS and thus the production (radiation sensitivity) irregularity in Fig. 5 is neutralized as shown in Fig. 7(f). The representativeness of the five thermistor pairs from the RRT results is based on the proportionality with the UAS results.

**Added statement (Line 247-251)**: The applied concept of transferring the individual radiation sensitivities from the RRT based on the five chosen units to Eq. (2) does not necessarily rely on 'realistic' irradiation and ventilation conditions in the RRT setup, but rather on the consistence of the existing conditions in the RRT over the radiation tests of all other sondes. The representativeness of the RRT results of the five thermistor pairs as part of all thermistors is based on the proportionality with the UAS results.

The number of parallel soundings, particularly at night, is too small to provide solid conclusions. I ask the authors to remove section 6.3 from the manuscript or at maximum report this in the appendix, although the clear need for a nighttime radiation correction, also stated by the authors themselves, suggest completely removing this part from the manuscript.

→ All contents on the nighttime soundings are removed from the paper while the daytime contents remain.

The authors state that "The ventilation speed for five units in Figure 6 is fixed at 5 m·s-1 and thus the effect of air ventilation cannot be identified. The effect of air ventilation is studied with a separate pair of thermistors". Given the irregularities in the production process can the values of the standard deviation of the residual for the pair of thermistors (reduced from 4.1% to 3.4% when the air ventilation is actually changed 4–6.5 m·s−1) be considered typical? Otherwise I'd suggest the authors to extend the assessment of the effect of air ventilation to a larger number of thermistors.

→ The effect of ventilation is not significant in the range of 4–6.5 m·s$^{-1}$ as shown in the Figure below and thus the decrease of the residuals is small when Eq. (10) is used to reflect the ventilation effect. Eq. (9) is deduced when the ventilation speed is 5 m·s$^{-1}$ while Eq. (10) is from 4–6.5 m·s$^{-1}$ using an average of the sensitivity coefficient (–0.08 °C/(m·s$^{-1}$)) over the range. The ventilation effect will become significant below 4 m·s$^{-1}$ because the convective cooling will be weakened as studied by the GRUAN. This remains to be our future study because we have learned that there are some researchers who wants to collect more data with a slow ascent of radiosondes.

[Figure]

**Added statement (Line 432-433)**: Since the radiation correction formula presented in this study is valid for the ventilation speed of 4–6.5 m·s$^{-1}$, the range should be widened to extend the applicability of the DTR.

**Specific comments**: Below also a few specific comments are also reported.

**Introduction**: the authors state that: "Radiosonde observations are often co-located with global navigation satellite system radio occultation and used as a reference for validating their one-dimensional interpolation which follows the flight trajectories of balloon soundings". I think the sentence must be more general, applications using GNSS data do not always consider interpolation along the balloon flight trajectories.

→ The sentence is modified to deliver that both techniques are mutually helpful.

**Before**: Radiosonde observations are often co-located with global navigation satellite system radio occultation and used as reference for validating their one-dimensional interpolation which follows the flight trajectories of balloon soundings.

**After (Line 34-36)**: Radiosonde observations can be co-located with global navigation satellite system radio occultation and these measurements are compared with each other to enhance the applicability and reliability of both techniques.

**Line 35**: provide a more general sentence considering that ground-based remote sensing measurements are not always less accurate than radiosoundings in all the atmospheric regions and conditions.

→ We have changed the sentence in the prior comment.

**Line 44**: It is not only needed to have "sufficiently small uncertainties" but also to well quantify them and in a traceable way. This is the motivation behind GRUAN, which must be reflected in the text. Please rephrase.

→ The sentence is modified to include the traceability as well as uncertainty.

**Before**: To investigate the climate change, a certain level of measurement uncertainty in radiosoundings should be secured.

**After (Line 43-44)**: To investigate the climate change, a certain level of measurement uncertainty in radiosoundings should be secured in a SI-traceable way.

**Line 121-122**: likely a repetition from the previous text, reword.

→ The sentence is modified to avoid a repetition.

**Before**: First, the thermistors on the sensor boom are individually calibrated using a climate chamber from −70 to 30 °C to evaluate the uncertainty of raw temperature measurement before radiation correction (Fig. 2(a)).

**After (Line 121-122)**: First, the calibration of thermistors attached to the sensor boom is conducted from −70 to 30 °C in a climate chamber (Fig. 2(a)) and the uncertainty of raw temperature measurement is evaluated.

**Line 131-132**: It would be interesting to show the differences between the second-order and third-order polynomial. I recommended adding these results to the manuscripts. Although from a separate paper under printing, a short sentence/summary of these results would be helpful for the reader.

→ A short summary of using the second-order and third-order polynomial is added.

**Added sentence (Line 145-147)**: In the work of Yang *et al.*, the maximum value of the residuals was 117 mK and 13 mK for the second-order polynomial and the third-order polynomial, respectively.

**Line 159-160**: the authors state: "One of the practical ways to reduce the calibration uncertainty is to conduct another round of calibration with the thermistor set (35 pairs) rotated 180° in the chamber and average out the effect of temperature deviations.". It is not clear to me if the authors performed the suggested additional round of calibration or not in the DTR assessment. Clarify and in the negative case, please, justify the related additional uncertainty contribution.

→ The idea is that the temperature deviations between the front and the rear can be averaged out if the thermistor set is rotated 180° in the second round of the calibration. The temperature of thermistors at the rear side is colder than the front side because wind blows from the rear side fan. However, the idea is not tried because we have found another effective way to reduce the spatial temperature deviation by moving the thermistor set (35 pairs) lower than the rear side fan to avoid the direct wind. The wind blows above the thermistor set and then the spatial temperature deviation within the thermistors set is reduced by about one fourth.

**Before**: One of the practical ways to reduce the calibration uncertainty is to conduct another round of calibration with the thermistor set (35 pairs) rotated 180° in the chamber and average out the effect of temperature deviations. Another way is to find other locations with smaller temperature deviations.

**After (Line 160-162)**: One of the practical ways to improve the calibration uncertainty is to find a location with reduced spatial temperature deviations in the climate chamber. More recently, the deviations are reduced by about one fourth of Fig. 3(b) at −70 °C by moving the thermistor set (35 pairs) lower than the rear side fan to avoid the direct wind.

**Line 181**: replace "next" with "following"

→ The word is replaced as suggested.

**Before**: next

**After (Line 183)**: following

**Line183-188**: see the general comments on irregularities in the production process.

→ This is answered in the General comment.

**Line 194**: The decrease with the temperature of (TB_on − TW_on) deserves more attention and specific measurements or more detailes discussion may be optionally added to the manuscript.

→ It is hard to figure out what this comment is about because there is no such mention in Line 194 of the previously revised manuscript. If this comment is about the temperature effect on

radiation correction, there are several factors affecting the convective heat transfer from the sensor to the air (i.e. cooling of sensor). The thermal conductivity and the viscosity of air decrease as the air temperature decreases while the density of air is inversely correlated with the temperature. The net effect of these air properties is that the convective heat transfer from the sensor to air is positively correlated with the air temperature. The detailed calculation is introduced in our previous paper (Lee *et al*. Atm. Meas. Tech. 5, 1107-1121, 2022). The effect of long-wave radiation from the sensor is minor compared with the effect of convective heat transfer. We have explained this point in the revised manuscript.

**Before**: because the convective heat transfer between the sensor and air is reduced as the thermal conductivity of the air is decreased at cold temperatures (Lee et al., 2021).

**After (Line 232-233)**: because the convective heat transfer between the sensor and air is reduced at cold temperatures with positive correlations between the thermal conductivity and the viscosity of air and the air temperature (Lee et al., 2022).

**Before**: This is attributed to the decrease of the thermal conductivity of air at cold temperatures, which reduces the heat transfer from the sensor to air despite the constant irradiation.

**After (Line 284-287)**: The thermal conductivity and the viscosity of air decrease as the air temperature decreases while the density of air is inversely correlated with the temperature. The net effect of these air properties is that the heat transfer from the sensor to air is positively correlated with the air temperature (Lee et al., 2022). The effect of long-wave radiation from the sensor is minor compared with that of convective heat transfer.

**Lines 212-213**: repetition, please remove.

→ The sentence is removed.

**Removed sentence**: Slight variations of air flow and/or pressure in the RRT chamber (not monitored) may partly be responsible for the observed distributions of radiative heating of the sensors in Figs. 5(c) and (d).

**Line 263**: The decrease in the uncertainty due to the change in ventilation speed appears to be small; can the authors provided a more detailed explanation? For example the range "4.5-6 m/s-1" means that results are an average over this range of values? If the measurements were at different values in this range, could you show a plot for this?

→ This is answered in the General comment.

**Line 307**: "…. in Eq. (18) and thus the effect of air ventilation cannot be identified in Fig. 7(f)", this is redundant, remove, please.

→ The phrase is removed.

**Removed phrase**: in Eq. (18) and thus the effect of air ventilation is not yet included in Fig. 7(f)

**Line 319-322**: beyond limitation in the number of compariaon profile, the fact that the sky was normally cloudy further decreases the value of the comparison with RS41, because of the larger uncertainties in cloud conditions tipically affecting the radiosonde measurements (e.g. radiation correction).

→ We agree that the number of comparison profile is not enough and thus we are preparing more comparison soundings this year. The Reviewer has a point on the cloud condition and the uncertainty. We plan to study how the cloud condition (reflection/screen of solar radiation) would affect the effective irradiance measurement and radiation correction of the DTR this year.

**Line 355-356**: this means that a proper correction would require a much larger number of parallel soundings than 3 only.

→ All contents on the nighttime soundings are removed from the paper.

**Line 375**: which previous work are you referring to here?

→ This part is removed.

**Line 376**: see general comments on the need for radiation correction also for nighttime DTR sonde profiles

→ This is answered in the General comment.

**Section 6.3**: In the previous review stage, I asked to improve the description of the results; the authors, instead, added a long description based on GRUAN related experiments. However, see my general comment on section 6.3.

→ All contents on the nighttime soundings are removed from the paper while the daytime contents remain. We think that there is nothing much to be added in Section 6.3 except comparing with the previous relevant works by the GRUAN.